# Synthetic X-ray-driven tracking and control of miniature medical devices

Chunxiang Wang [1,2], Wenbin Kang [3] ✉, Mengmeng Sun[1,4], Hongchuan Zhang [1], Chong Hong [1], Sinan Ozgun Demir [1], Halim Ugurlu [5], Kun Hao[6], Zemin Liu [1,2], Tianlu Wang [1,7] ✉ & Metin Sitti [1,8] ✉

The clinical translation of miniature medical devices (MMDs) for minimally invasive surgery promises transformative advances in biomedical engineering, offering enhanced precision, reduced patient trauma and faster recovery times. However, their effective deployment in complex anatomies under real-time X-ray guidance—a widely used surgical imaging modality—presents challenges such as low imaging quality and difficulties of spatial MMD control. Manual identification and operation are labour intensive and error prone. Meanwhile, deep learning-based automation is limited by the scarcity of annotated X-ray datasets of MMDs owing to costly data collection, laborious annotation and privacy constraints. Here we introduce MicroSyn-X, a framework for training computer vision models to enable robotic teleoperation of MMDs using synthesized high-fidelity, pixel-accurate, auto-labelled and domain-randomized X-ray images, eliminating manual data curation. Integrating MicroSyn-X into a teleoperated robotic system enables real-time localization and navigation of magnetic soft and magnetic liquid MMDs within both ex vivo and dynamic in vivo environments, demonstrating robustness under challenging imaging conditions of low contrast, high noise and occlusion. With these promises, we open source the X-ray MMD dataset to enable benchmarking. Addressing data scarcity and enabling real-time robotic navigation, this work advances MMD-assisted minimally invasive surgery towards next-generation precision interventions.

The integration of miniature medical devices (MMDs), ranging from miniature robots[1–3] to implantable biosensors[4–6], into minimally invasive surgical practices is transformative in biomedical engineering[7,8]. Specifically, small-scale devices actuated by external fields can navigate through enclosed spaces challenging for conventional tethered tools[7,8] and offer functionalities such as drug delivery[9–11] and physiological property sensing[12–14]. To translate these innovations towards clinical applicability, safe and effective MMD deployment is essential, requiring real-time medical imaging to continuously monitor the operation in non-transparent biological environments[15]. Among available medical imaging modalities[16–32] (Extended Data Tables 1 and 2), X-ray fluoroscopy is widely used in surgeries given its deep tissue penetration, high resolution, near real-time imaging and expansive visualization window[33,34]. However, fluoroscopic image-guided operation in dynamic, cluttered anatomical environments remains labour intensive, leading to operator fatigue and reduced precision[35–37]. Limitations persist in

[1]Physical Intelligence Department, Max Planck Institute for Intelligent Systems, Stuttgart, Germany. [2]Department of Information Technology and Electrical Engineering, ETH Zurich, Zurich, Switzerland. [3]Department of Mechanical Engineering, City University of Hong Kong, Hong Kong, China. [4]Department of Biomedical Engineering, National University of Singapore, Singapore, Singapore. [5]Zentrum für Radiologie Heilbronn, Heilbronn, Germany. [6]Department of Pharmacology, Shandong University, Jinan, China. [7]Department of Mechanical Engineering, University of Hawai'i at Mānoa, Honolulu, HI, USA. [8]School of Medicine and College of Engineering, Koç University, Istanbul, Turkey. ✉e-mail: wenbin.kang@cityu.edu.hk; tianluw@hawaii.edu; msitti@ku.edu.tr

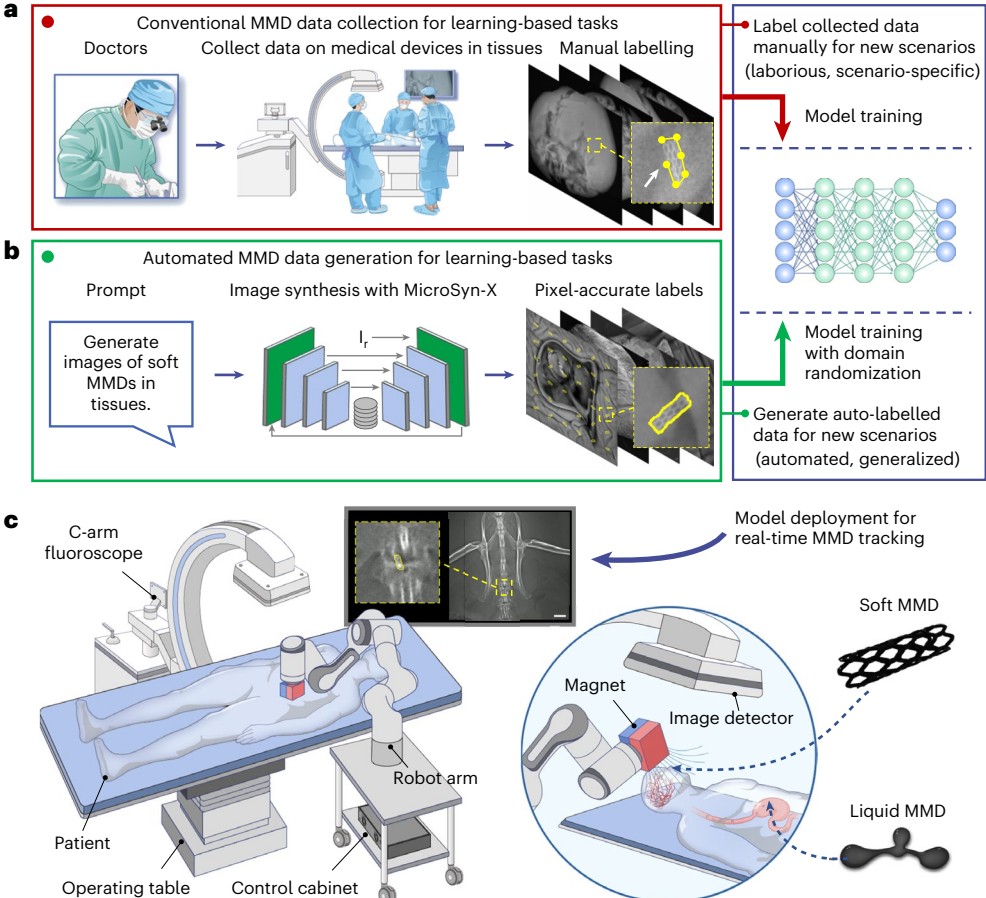

**Fig. 1 | Overall concept of MicroSyn-X. a**, The conventional data collection of MMDs for learning-based tasks. Clinicians perform labour-intensive image collection and annotation in real tissues, requiring repeated efforts for new MMDs or anatomical targets. **b**, The automated model training with MicroSyn-X. MicroSyn-X automates domain-adapted synthetic image generation, eliminating manual labelling and enabling scalable training across diverse MMD–tissue scenarios. MMD images are denoted by $I_r$. **c**, A robotic platform for magnetic MMD manipulation. A robotic arm with an actuating magnet remotely guides MMDs, including stent-structured and shape-morphable liquid MMDs, through anatomical barriers. Real-time C-arm fluoroscopy visualizes the MMD, while the machine learning model trained with MicroSyn-X provides guidance feedback. The image is from an in vivo animal experiment with a 5 mm scale bar.

manual object identification from overlapping anatomical features[38,39] and precise manual adjustments of tools[37,40–42], underscoring the imperative for robotic systems with automated object tracking to improve procedural efficiency and alleviate human effort.

Deep learning excels at medical image analysis, particularly in complex environments challenging for conventional image-processing techniques using handcrafted features[43–46]. However, its performance relies on large, high-quality annotated datasets. Insufficient data causes poor generalization, resulting in deceptively high training accuracy and performance degradation on unseen clinical settings[47]. Furthermore, data scarcity is common in medicine owing to the challenges of data collection and annotation[48,49] (Fig. 1a). Data collection is hindered by the need for specialized imaging equipment to capture MMDs within tissues and stringent ethical regulations restricting public access[50,51]. Meanwhile, annotation demands meticulous manual effort, such as selecting precise bounding points, which is laborious and prone to human error due to cognitive fatigue[48,49].

Synthetic data offer a transformative solution to data scarcity by leveraging generative models or simulations to create artificial datasets that mimic the structural properties of real-world data[50–52]. This further enables the creation of diverse datasets that mitigate class imbalance, where rare conditions or underrepresented objects hinder model generalization[48]. Current applications have demonstrated their clinical utility[49], with well-validated generation quality and enhanced downstream model performance in surgical scene synthesis for organ segmentation[53–57] and

depth estimation[58–60], privacy-preserving X-ray image generation for pathological analysis[61–64] and disease image synthesis[65–69].

Small-scale medical devices present unique challenges for image-guided deployment. Unlike macroscopic features, these tiny components appear as low-contrast, noisy entities within cluttered anatomical scenes, easily occluded by surrounding tissues and manipulation tools[70,71]. Furthermore, MMD deployment requires real-time tracking and integration into robotic systems to ensure precision and robustness under clinical constraints[72], such as poor imaging conditions and dynamic anatomical variability. The utilization of state-of-the-art computer vision (CV) models in MMD-relevant scenarios is impeded by the absence of publicly accessible datasets and generative models tailored for MMDs. While general-purpose models such as the Segment Anything Model[73,74] and vision language models[75,76] excel in broad CV tasks, they require fine-tuning with domain-specific data in specific medical contexts[77–80]. Likewise, self-supervised learning for label-free model training minimizes annotation but still demands high-quality datasets[81–83]. In addition, current robotic systems for MMD deployment exhibit limitations, such as manual object identification[3,9,14,42], experiments within simplified phantoms instead of realistic clinical environments[84–86] and focus on macroscopic devices with good visibility[51,87,88].

To address the challenges of tracking and deploying micro- and millimetre-scale MMDs in complex anatomical environments, we propose MicroSyn-X, a framework that synthesizes X-ray MMD images

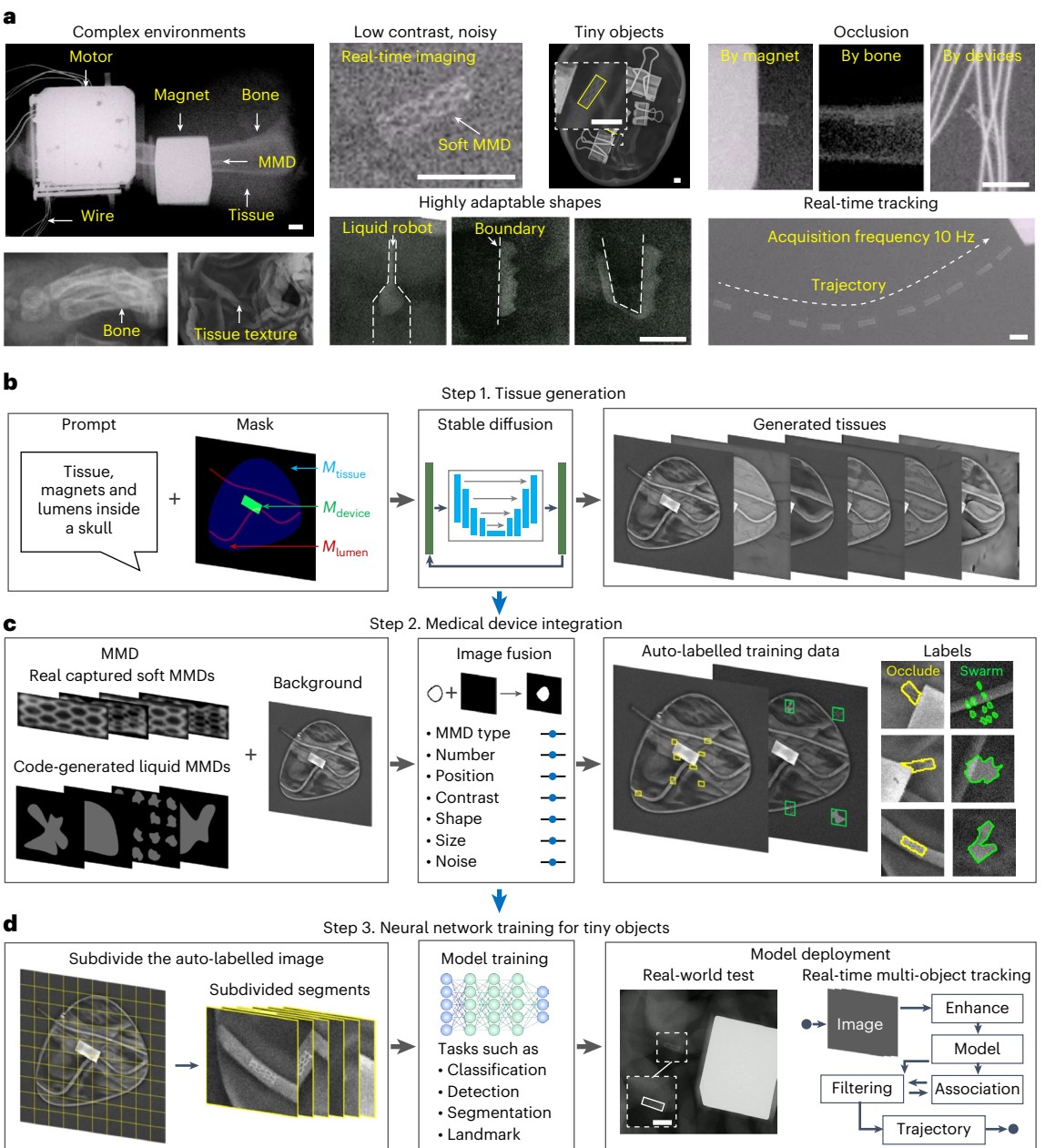

**Fig. 2 | Workflow of MicroSyn-X. a**, The clinical challenges of MMD perception under real-time X-ray imaging. **b**, The controlled tissue generation process. Stable diffusion creates high-fidelity tissue images from user-defined masks and prompts. **c**, The integration of medical devices with tissues. Captured or generated MMD images are seamlessly integrated into the background with flexible parameters, ensuring pixel-accurate labelling. **d**, A schematic of neural network training for MMD tracking. MMD–tissue images are subdivided to improve precision and efficiency in tiny object localization. The model, trained on synthetic data, is then deployed for real-world MMD tracking.

for training CV models, and develop a teleoperated robotic system achieving vision-based MMD control using these trained models. Unlike conventional approaches of recursively curating and annotating MMD data (Fig. 1a), MicroSyn-X enables end-to-end synthesis of high-fidelity, pixel-accurately labelled MMD data (Fig. 1b). Guided by user prompts, it controllably generates images of MMDs inside diverse environments to mimic clinical scenarios with domain randomization, where scene properties, such as MMD appearances and background complexity, are randomized. This approach enables rapid adaptability to new applications, addressing data scarcity while eliminating manual labelling. Furthermore, the synthesized data directly trains CV models to deploy on robotic systems, enabling precise MMD navigation despite low contrast, occlusion and imaging noise under clinical X-ray fluoroscopy (Fig. 1c), relieving users of labour-intensive operations.

We demonstrate real-time deployment and tracking of two representative MMDs, including magnetically actuated soft MMDs[3,9] and shape-adaptable liquid MMDs[89] in ex vivo and in vivo environments, validating the framework's effectiveness. Notably, we open-source the X-ray MMD dataset, facilitating benchmarking and democratizing research in MMDs. Bridging the data gap and enabling precise MMD control, this work advances the feasibility of MMD-assisted minimally invasive surgery under clinical X-ray imaging, marking a critical step towards next-generation precision interventions.

## Results

### Workflow of MicroSyn-X

The challenges of tracking MMDs under clinical fluoroscopy are summarized in Fig. 2a, including complex imaging environments, tiny objects

in low-contrast and noisy scenes, occlusion and adaptive MMD shapes. X-ray imaging leverages differential attenuation of X-rays through materials with varying density and atomic composition, presenting constraints in clinical settings[33,34]. First, all anatomical structures are captured within the field of view, creating visual distractions that obscure MMDs with inherently poor visibility, requiring operators to manually search objects. Occlusion further complicates localization, as dense tissues such as bone or metallic components can block MMDs in two-dimensional (2D) projections[70,71]. Moreover, imaging noise degrades clarity, particularly under low milliampere second (mAs) fluoroscopic settings[90]. Specifically, the shape-deformable liquid MMDs enables the navigation of confined spaces but requires CV models to handle continuously changing shapes[89,91,92].

We present MicroSyn-X, a modular, label-free end-to-end synthetic data generation pipeline for clinical CV model deployment, structured into three stages for MMD tracking. First, background generation leverages a diffusion model to synthesize surgical scenes with anatomically realistic tissue textures and operation tools. Second, magnetic MMDs, captured from real-world scenarios or algorithmic generation, are programmatically overlaid onto synthetic backgrounds. This process incorporates domain randomization, adjusting parameters such as contrast and shape, while generating pixel-accurate labels. This step eliminates manual annotation, a critical bottleneck in medical CV tasks. Finally, the synthetic dataset is used to train a downstream CV model for tasks such as detection and segmentation. The trained models are integrated into a real-time object tracking framework for soft and liquid MMDs in physiological environments. This framework offers a fast and scalable solution to train CV models for MMD navigation.

The framework firstly employs a pix2pix stable diffusion model[93] to synthesize surgical scenes utilizing user prompts and mask inputs, as shown in Fig. 2b. The input mask image, composed of three channels (tissue area $M_{tissue}$, metallic device area $M_{device}$ and contrast agent-filled lumen area $M_{lumen}$), enables precise control over anatomical structures and device placement. Users can customize the shape, position and brightness of these regions while providing prompts to specify scene composition. This hybrid approach, combining conditional diffusion with spatial guidance, generates backgrounds replicating anatomical variability, such as differences in organ geometry or tissue density, critical for training robust CV models.

MMD integration (Fig. 2c) further merges MMD images and tissue via an add-weighted strategy (see 'MMD data preparation and integration' in Methods). This approach mimics the X-ray imaging mechanism, where the MMD modulates the X-ray signals of tissues, creating overlapping attenuation in a 2D projection[33]. MMD images are obtained by capturing static devices on clean backgrounds or generating deformable liquid MMD shapes using parametric spline curves to mimic dynamic shape changes (Supplementary Fig. 1 and 'Mask generation with spline curves' in Methods). The MMD is pasted with a known mask, automatically generating labels (class and bounding points) without manual annotation. To reflect clinical realism, Poisson, Gaussian and pepper noise are injected to simulate imaging noise, while occlusion is modelled by positioning the MMD beneath the $M_{device}$ area. This workflow ensures synthetic data captures heterogeneous tissue–MMD interactions and contrast variations, offering computational efficiency and scalability for large datasets.

To train a CV model with strong generalization capabilities and minimize the sim-to-real gap, domain randomization is implemented in two approaches (Fig. 2b,c). First, background randomization introduces variability in anatomical and imaging conditions by altering tissue type, brightness, shape and position by adjusting user prompt and $M_{tissue}$; device morphology ($M_{device}$) parameters such as shape, position and brightness; lumen structures ($M_{lumen}$) with randomized shape and contrast; and controlled noise levels. Second, MMD-specific attributes are randomized, including type, quantity, position, shape (via geometrical transformation) and contrast. This strategy ensures a broad data distribution, forcing the CV model to prioritize invariant features (such as MMD contours) over spurious correlations (such as texture-specific cues), a principle validated in domain generalization studies[48,51].

For localizing MMDs in large, cluttered images, where small objects risk being drowned out by irrelevant background features, a patch-based strategy is utilized for CV model training and inference (Fig. 2d and Supplementary Fig. 2). Images are subdivided into smaller patches for model training and inference, aligning with evidence that deep neural networks excel at capturing fine-grained details when trained on region-specific data[94,95]. By prioritizing local features over global context, the method mitigates challenges of low contrast and noise and improves model generalization, and enables scalable training on memory-constrained hardware, as smaller tiles fit within graphics processing unit (GPU) memory limits while maintaining high-resolution analysis. Last, the trained models are integrated into a multi-object tracking framework, enabling continuous MMD localization.

## Tissue generation and open-source X-ray MMD dataset

Synthetic tissue background generation is critical for training robust CV models considering the heterogeneous tissue textures, dynamic occlusions and variable noise of real scenarios. It is difficult to capture all such variations with manual data collection, whereas synthetic data provide an efficient and cost-effective alternative. Diffusion models have proven effective for generating realistic medical images[49,52,65] owing to their training stability and fine detail synthesis[56,62,63,68]. We employed a pix2pix diffusion model[93] to generate X-ray images with minimal manual efforts (Supplementary Fig. 3 and 'Diffusion model training and inference' in Methods); existing X-ray datasets, such as the small-mammal anatomical dataset[96], can also be incorporated.

The generation results demonstrate that domain randomization effectively enhances tissue heterogeneity. When randomizing parameters such as diffusion steps and prompt guidance weights the framework produced high-fidelity backgrounds with a structural similarity index (SSIM)[97] ranging from 0.65 to 0.91 compared with real images (Fig. 3a,b). Programmatically varying mask shapes and prompts further enables diverse anatomical textures within controlled boundaries, alongside customized metallic devices and lumens (Fig. 3c). Domain analysis validates the expanded diversity of synthetic data: Inception V3 feature extraction[98] and principal component analysis (PCA)[99] reveal a broader distribution than real data, covering underrepresented regions (Fig. 3d). As generation quality directly influences downstream model performance, quality control procedures are detailed in the 'Diffusion model training and inference' in Methods.

This X-ray MMD dataset is open source, featuring stent-structured soft MMDs[3,9] and shape-morphing ferrofluid MMDs[89] (Fig. 3e and Supplementary Fig. 4). The dataset is composed of real and synthetic domains. The real domain comprises dynamic (D1) and static (D2) subsets: D1 captures real-time MMD locomotion across diverse tissues, while D2 contains static MMDs under varying imaging conditions (Extended Data Fig. 1 and 'Dataset preparation' in Methods). The static subset enables quantitative evaluation of imaging parameter effects on model performance, while dynamic datasets facilitate testing tracking algorithms. The synthetic dataset (D3) was created by MicroSyn-X. This publicly available dataset provides a foundation for training and benchmarking CV models on MMD data, enabling systematic comparison of different architectures and training strategies.

## Evaluation of MicroSyn-X

The evaluation aims to assess its ability to bridge the synthetic-to-real gap and robustness under unpredictable imaging conditions and anatomical variability. It is compared with baselines of conventional CV model training and clinical experts and validated across ex vivo tissues, in vivo experiments, multiple CV models and various imaging conditions. Unlike incomplete and expensive real data, synthetic data facilitate

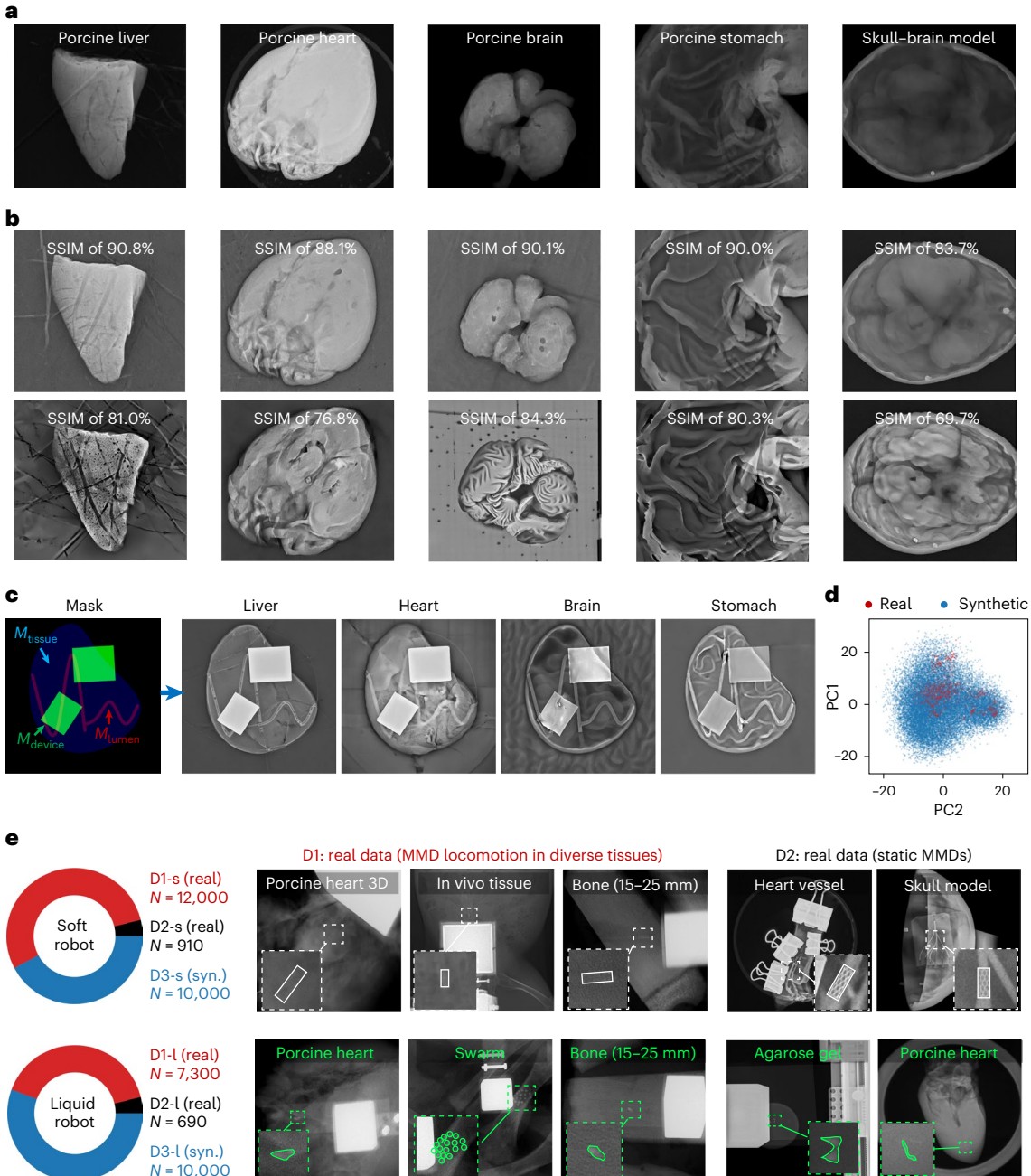

**Fig. 3 | Domain randomization of synthetic tissue images and open-sourced MMD X-ray dataset. a**, Real tissue images under X-ray imaging. **b**, The generation of precisely matched synthetic tissues with randomized and enhanced textures. **c**, Image generation with mask-guided conditioning and prompts. Randomization of masks and prompts substantially expands the dataset. **d**, A domain comparison of real and synthetic tissues. Features from 1,140 real and 24,803 synthetic tissue images, extracted using Inception V3, are visualized via PCA. **e**, An overview of the open-source MMD dataset under X-ray imaging. Dataset 1 (D1) contains real-time recordings and annotations of MMD locomotion. Dataset 2 (D2) includes static MMD images captured under various voltage and current conditions. Dataset 3 (D3) provides synthetic images with corresponding segmentation labels. The suffixes '-s' and '-l' denote the soft and liquid MMDs, respectively.

expanding data distributions and improving CV model adaptability. Three datasets are presented: D1 (real MMD locomotion) serves as the test set, while D2 (static MMDs under varied imaging conditions) and D3 (synthetic data) train models, respectively (model (syn.) and model (real)) (Extended Data Fig. 2a). Features extracted from D1–D3 via the model (syn.) backbone ($F_1, F_2$ and $F_3$) are visualized via dimensionality reduction, and used for data distribution coverage analysis (Extended Data Fig. 2b–e and Supplementary Fig. 5). For real-time MMD localization, the YOLO11-seg class was adopted for its high accuracy and speed[100] ('CV model training and inference' in Methods). Performance metrics included average precision (AP), mean AP at intersection of union (IoU) of 0.50 (mAP50) for basic localization accuracy and mAP50:95 for rigorous evaluation ('Computation of metrics' in Methods).

MicroSyn-X demonstrates generalization and robustness in realistic scenarios. For soft MMDs, model (syn.) outperforms model (real) in both mAP50 and mAP50:95, especially in low-contrast, high-noise environments such as dynamic in vivo environments (Extended Data Fig. 2d and Supplementary Fig. 6a). It is attributed to expanded data distribution through domain randomization while preserving realistic MMD appearance, covering edge cases

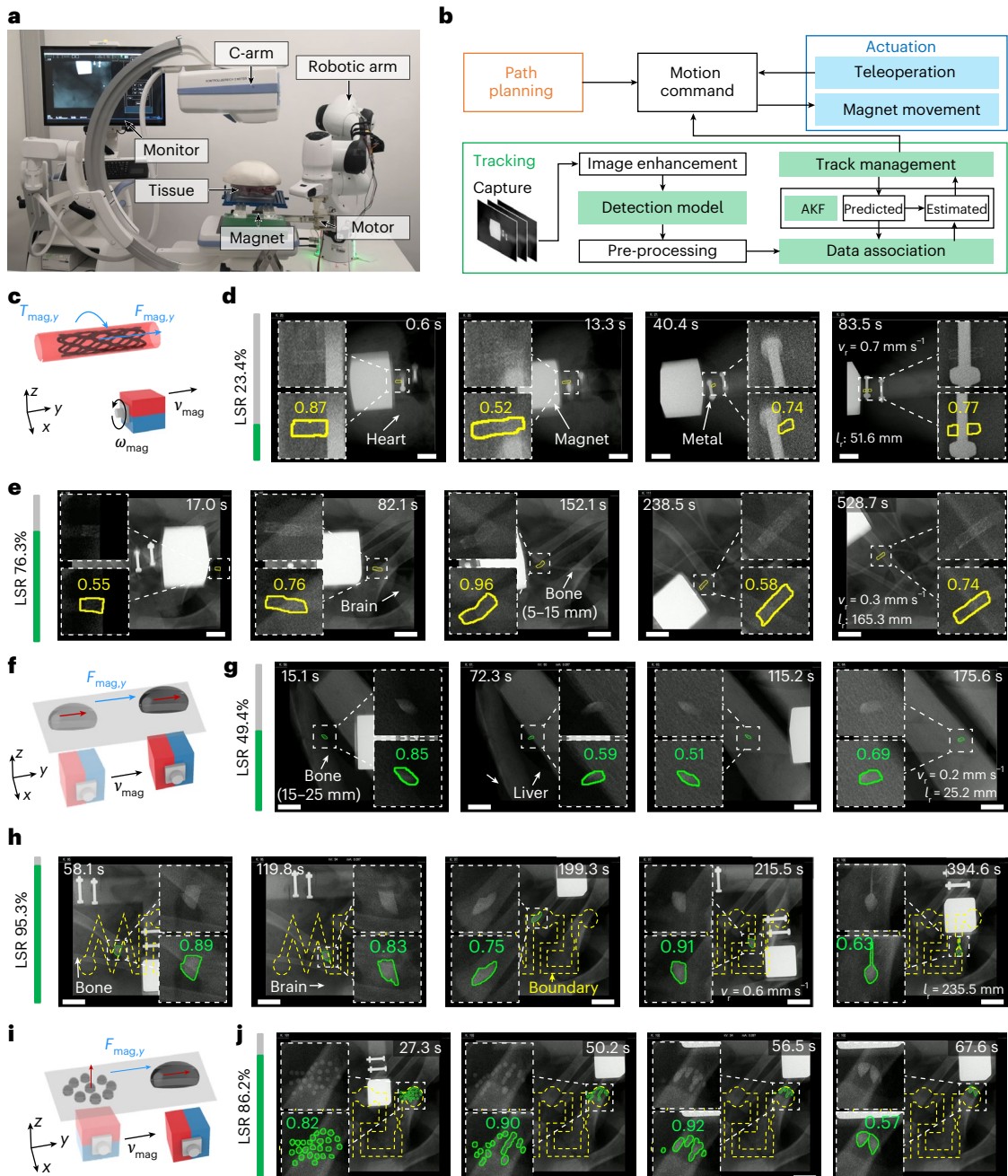

**Fig. 4 | Robotic system integrated with MicroSyn-X-trained models. a**, The X-ray-guided robotic actuation system. **b**, A schematic of the supervised autonomous robotic navigation system. **c**, The auction principle of the soft MMD utilizing magnetic torque and force. Using a rotating 30-mm N45 cubic magnet, the magnetic torque and force range from 2.0 μNm to 13.2 μNm and from 0.1 mN to 0.4 mN, respectively. Soft MMD dimensions were 1.5 mm in diameter and 5.0 mm in length. The axes $x$, $y$ and $z$ define the coordinate system. $F_{mag,y}$ and $T_{mag,y}$ denote the magnetic force and torque along the $y$-direcon, respectively, while $v_{mag}$ and $\omega_{mag}$ represent the magnet's translational velocity and rotational speed. **d**, Real-time soft MMD tracking under high-occlusion, noisy and low-contrast imaging conditions. The localization success ratio (LSR), defined as the ratio of successfully localized frames to total video frames, is displayed alongside. The top row illustrates zoomed-in MMD views, while the bottom row shows segmentation results with confidence scores. **e**, Robotic navigation and real-time tracking within tortuous lumen networks under bone occlusion conditions. **f**, The magnetic gradient-driven translation of a ferrofluidic MMD. The red arrow represents the polarization direction of the magnetic field. Using a 20-mm N45 cubic magnet, the magnetic gradient along the desired motion direction ranges from 0.06 to 0.12 T m⁻¹. The liquid MMD volume is 40–60 μl. **g**, Real-time liquid MMD tracking beneath bone structures up to 25 mm in thickness. **h**, Liquid MMD tracking in environments with abrupt spatial variations under bones in real time. The MMD morphology dynamically adapts to structural boundaries, such as the narrow-channel segment 'I' of the MPI configuration, enabling navigation through confined pathways. **i**, The liquid MMD separation and recombination via magnetic field modulation. **j**, The real-time tracking of liquid MMD swarms and recombination dynamics under bone occlusion. Scale bars represent 10 mm. The mean MMD translation speed and locomotion distance are denoted by $v_r$ and $l_r$, respectively.

impractical to collect in real-world settings. In the domain analysis of soft MMDs, synthetic data exhibits broader coverage compared with real data (Extended Data Fig. 2c and Supplementary Fig. 5), highlighting its ability to span underrepresented variations. For liquid MMDs, model (syn.) achieves comparable mAP50 to model (real) and surpasses it in stricter mAP50:95 (Extended Data Fig. 2f and Supplementary Fig. 6b). While mathematically generated spline curves introduce shape diversity, their simplified appearances lead

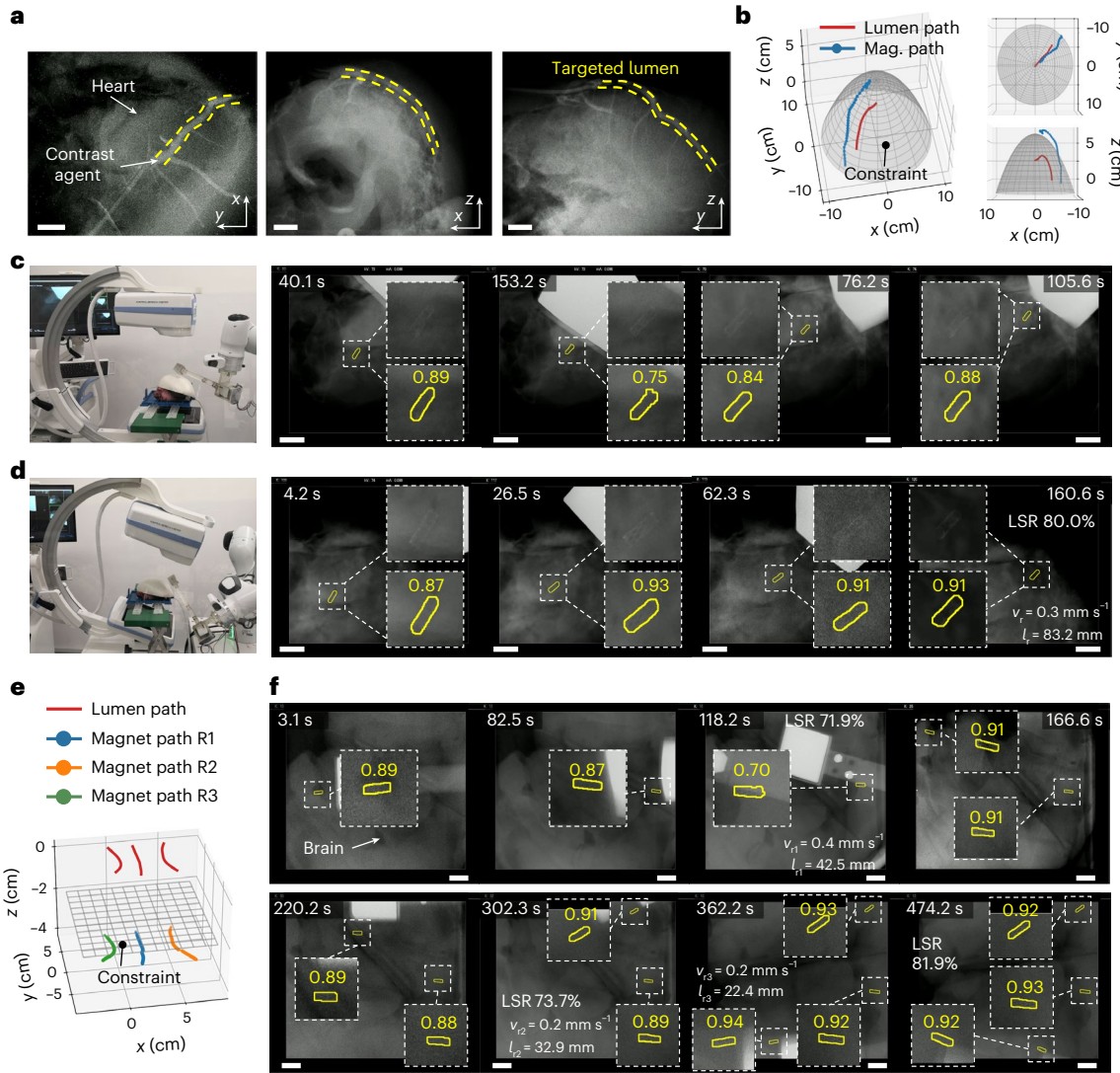

**Fig. 5 | Robotic navigation in 3D ex vivo tissues and multi-MMD deployment. a**, A 3D reconstruction of vascular pathways via intra-vascular contrast agent injection and multi-angle imaging. **b**, A reconstructed lumen path and planned magnet path for targeted robotic deployment. **c**, MMD deployment under low-contrast imaging conditions. Left: imaging-angle configuration. Top row insets: magnified MMD views. Bottom row insets: segmentation outputs with confidence scores. **d**, Real-time MMD tracking under imaging-angle misalignment. The segmentation neural network trained on fixed-angle MMD imagery demonstrates robust generalization to appearance variations induced by 3D rotational misalignment. Left: imaging-angle configuration. Top row insets: magnified MMD views. Bottom row insets: segmentation outputs with confidence scores. **e**, The path planning for multi-MMD deployment. **f**, Real-time multi-MMD deployment and tracking. Scale bars represent 10 mm. The mean MMD translation speed and locomotion distance are denoted by $v_r$ and $l_r$, respectively. Using a rotating 30-mm N45 cubic magnet, the magnetic torque and force range from 2.0 μNm to 13.2 μNm and from 0.1 mN to 0.4 mN, respectively. Soft MMD dimensions of 1.5 mm in diameter and 5.0 mm in length.

to lower data coverage (Supplementary Fig. 5) and slightly reduce mAP50 for easy detections. However, this diversity enhances performance in complex tasks such as tracking swarms under bone occlusions, particularly during dynamic shape transitions (splitting and merging). Future improvements could focus on refining MMD fidelity (Supplementary Fig. 7) and incorporating physics-based deformation models[101]. Scalability tests further validate the effectiveness of MicroSyn-X (Extended Data Fig. 2g), where models of varying sizes (2.8 M, 10.1 M, 22.4 M and 27.6 M parameters) achieve comparable high accuracy.

We also investigated the impact of synthetic background quality on downstream CV models (Extended Data Fig. 3). For MMDs with distinct features, such as stents, performance is largely unaffected by background quality, whereas for MMDs with ambiguous features, the effect is model-dependent: smaller models degrade with low-quality data, while larger models can utilize it as effective regularization. To

tackle this issue, we adopt a two-phase quality control strategy during tissue generation (diffusion model selection and artefact minimization) and prioritize the utilization of large downstream models with its robustness to noise ('Diffusion model training and inference' in Methods). A classifier can be developed to automatically select backgrounds for CV model training as a future step.

To evaluate the clinical relevance, we benchmarked its performance with experts in low-contrast and high-noise environments. Six soft MMDs were placed within a three-dimensional (3D) lumen phantom (Extended Data Fig. 4a,b) and imaged across varying X-ray voltages and currents (Supplementary Fig. 8). Both clinical experts and the CV model were tasked with counting visible MMDs: experts manually annotated images, while the model required segmentation with an IoU >0.5 for valid predictions. Quantitative analysis revealed that the model outputs matched expert consensus (Extended Data Fig. 4c). For soft MMDs in dataset D2-s, a subset

was manually annotated (Supplementary Fig. 9 and 'Computation of metrics' in Methods). The CV model outputs aligned with manual identification (Extended Data Fig. 4d), reliably detecting MMDs when contrast exceeded 0.018, a threshold challenging for operators owing to signal degradation. These results validate the clinical applicability of MicroSyn-X.

## Fluoroscopy-guided robotic deployment

Utilizing the CV model trained with MicroSyn-X, the telerobotic system translates vision-based localization into robotic deployment, integrating hardware and software to enable image-guided deployment of MMDs under X-ray imaging. In Fig. 4a, the system uses a robotic arm with ±1 mm precision, a permanent magnet (PM) mounted on a stepper motor and C-arm fluoroscopy for real-time navigation ('Latency of the teleoperated robotic system' in Methods). The software consists of four modules: planning, actuation, tracking and control (Fig. 4b). The planning module computes the MMD and PM paths[3], utilizing preoperative data (computed tomography[102] or rotational angiography[103]) and the MMD dynamic model (Extended Data Fig. 5). The actuation module drives PM translation and rotation to execute planned motions, while the tracking module localizes MMDs as feedback. The control module implements supervised autonomy: operators issue high-level commands (for example, 'advance to the next waypoint'), while the system autonomously executes manipulation and localization. This hybrid scheme maintains operator oversight, as required in clinical workflows[8], while enhancing precision and repeatability through automation.

In detection-based MMD tracking, the CV model localizes the MMD in individual frames (Supplementary Fig. 2) and the tracking algorithm links these detections into trajectories. To handle the dynamic, low-contrast and noisy imaging environment with frequent occlusions, the system mitigates false positives, missed detections and abrupt appearance changes through several strategies. Each frame is preprocessed (for example, brightness/contrast adjustment and histogram equalization) to enhance MMD visibility[97]. Detection outputs are filtered by confidence scores, geometric consistency, spatial plausibility and temporal persistence, and the adaptive Kalman filter interpolates missing data during occlusions[104] (Extended Data Fig. 6 and 'Measures for handling degradation in image quality' in Methods).

The robotic system demonstrates robust deployment of MMDs in clinically relevant scenarios. For soft MMDs, a rotating PM generates magnetic torque and force for navigating complex anatomical pathways[3] (Fig. 4c and Extended Data Fig. 5). Supplementary Movie 1 demonstrates reliable tracking across diverse tissue types despite varying tissue textures, imaging noise and partial occlusions. To validate tracking robustness under extreme conditions, stress tests were conducted in low-contrast, high-noise and severe occlusion environments. In Fig. 4d, the PM rotated at 1.3 Hz, inducing rapid occlusions and degraded visibility. In Fig. 4e, a soft MMD navigated contrast agent-filled lumens, traversing bifurcations and reversing direction under persistent bone-induced occlusions. Despite these adversities, the tracking algorithm maintained uninterrupted localization, demonstrating its capacity to handle non-detections and false positives (Supplementary Movie 2).

Ferrofluid-based liquid MMDs exhibit exceptional deformability, allowing adaptation to complex terrains but posing challenges for tracking. As shown in Fig. 4f and Extended Data Fig. 5, magnetic gradients generated by the PM drive droplet translation, while controlled magnetic field orientation induces shape deformation to navigate uneven or confined spaces. Supplementary Movie 3 and Fig. 4g demonstrate successful tracking as it traverses a 25-mm thick bone phantom, even at occluded boundaries. In Fig. 4h and Supplementary Movie 4, the MMD navigated an 'MPI'-shaped structure with randomized bone occlusions (where MPI stands for Max Planck Institute), deforming substantially to pass through narrow channels while remaining tracked. The system also supports dynamic splitting and merging of ferrofluid

droplets[91] (Fig. 4i). A strong vertical magnetic field (from PM proximity) generates internal repulsive forces exceeding surface tension, splitting the droplet into smaller units[105]. Subsequent horizontal PM polarization initiates reassembly[105]. As shown in Fig. 4j and Supplementary Movie 4, swarm droplets were constantly tracked during the merging process under persistent bone and magnet occlusion. These results underscore the pipeline's capability to track highly deformable objects in constrained, high-occlusion scenarios.

## Robotic deployment and tracking in ex vivo and in vivo tissues

To test the robotic deployment and tracking framework in realistic tissue environments, we conducted experiments in ex vivo and in vivo settings. A soft MMD was first deployed in a 3D curved porcine artery, then three MMDs were sequentially deployed beneath a skull model. In dynamic in vivo scenarios, a soft MMD navigated the rabbit femoral artery under physiological motion, while long-distance deployment in the rat abdominal aorta and iliac artery verified robust tracking despite severe bone occlusion and imaging degradation.

For robotic deployment in ex vivo tissues, the contrast agent was injected into the porcine heart artery, imaged from multiple angles and the 3D path was reconstructed by correlating the centreline of the contrast-enhanced regions (Fig. 5a). The path planning algorithm further computed PM trajectories optimized for magnetic actuation, considering the workspace constraints[3] (Fig. 5b). The robotic system then executed user commands, guided by tracking results, to steer the PM along the planned path (Fig. 5c). To maximize visibility during deployment, the C-arm angle was adjusted to align with the MMD position. Despite low-contrast and noisy imaging conditions, the MMD was continuously tracked (Fig. 5d and Supplementary Movie 5). In Fig. 5e,f, three soft MMDs were sequentially deployed and tracked simultaneously in separate lumens, with all MMDs remaining tracked despite occlusions (Supplementary Movie 6).

A hybrid robotic deployment strategy is proposed for navigation in environments in vivo, integrating a customized mechanical device with fluoroscopic guidance and magnetic actuation. The system employs a suture with tailored flexibility and biocompatible materials to enable fluid-driven locomotion while ensuring fail-safe control during clinical interventions (Supplementary Fig. 10). A soft MMD (550 μm outer diameter) is delivered into the vasculature via an artery sheath, after which blood flow is harnessed for passive advancement under real-time fluoroscopic imaging. Directional control is achieved using a static magnet to guide the MMD to navigate through bifurcations to enter desired branches. In small blood vessels with insufficient haemodynamic force, the rotating PM generates magnetic torque and force to overcome resistance, enabling active navigation in low-flow environments. This dual-mode approach, combining physiological fluid dynamics with external magnetic actuation, enhances adaptability in complex anatomical settings.

The first in vivo demonstration involved a soft MMD navigating the rabbit femoral arterial network (Fig. 6a). Utilizing rotating PM actuation, the MMD traversed complex vascular structures (Fig. 6b and Supplementary Movie 7), where it entered two branches and executed bidirectional locomotion. In the second experiment, a rat model was used (Fig. 6c), where the MMD was delivered into the abdominal aorta and propelled by blood flow to waypoint 2 before entering a bifurcation. Despite a large imaging window, low imaging resolution and continuous spinal occlusion, the tracking algorithm achieved effective localization (Fig. 6d). Under combined magnetic guidance and fluid dynamics, the MMD entered the bifurcation area, after which the rotating PM actuated it to the distal target area. Detailed evaluation results are shown in Supplementary Fig. 11. Histology and biocompatibility analysis confirm the safety, biocompatibility and haemocompatibility[9] (Supplementary Fig. 12 and 'Histological examination' in Methods). These results underscore the robustness of the tracking algorithm in clinically relevant scenarios, highlighting the potential of soft MMDs

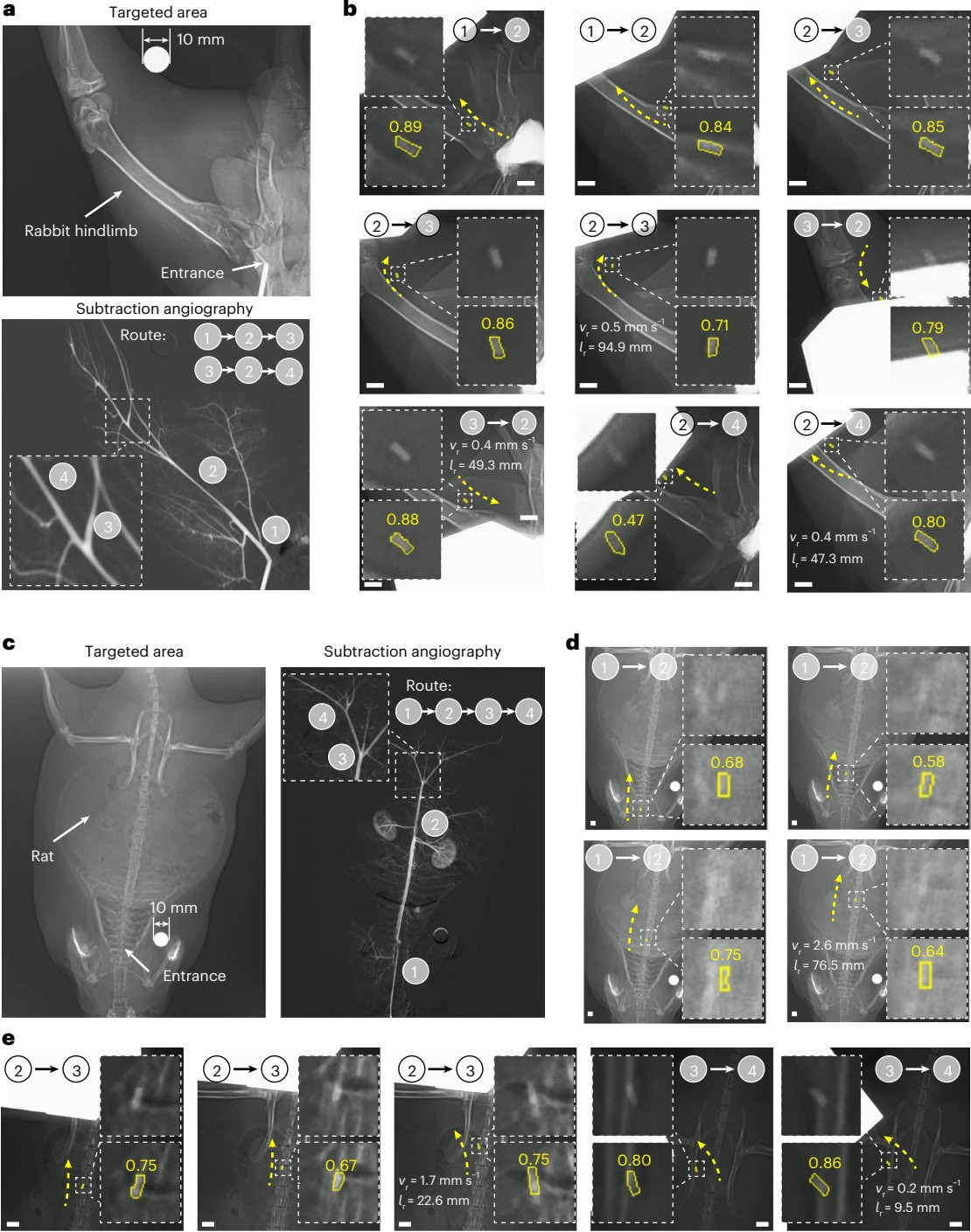

**Fig. 6 | Robotic navigation in arterial environments in vivo. a**, The targeted rabbit femoral arterial network with numbered waypoints during navigation. **b**, Navigation in dynamic in vivo conditions. Top row insets: magnified intra-arterial MMD views. Bottom row insets: segmentation outputs with confidence metrics. **c**, The targeted in vivo rat arterial region. **d**, The MMD deployment leveraging blood flow in the abdominal aorta. **e**, Magnetically actuated MMD navigation in the abdominal aorta and lilac artery. Scale bars represent 5 mm. The mean MMD translation speed and locomotion distance are denoted by $v_r$ and $l_r$, respectively. Using a rotating 60-mm N45 cubic magnet, the magnetic torque and force range from 0.46 μNm to 4.11 μNm and from 0.05 mN to 0.11 mN, respectively. Soft MMD dimensions of 0.55 mm in diameter and 1.80 mm in length.

for minimally invasive vascular interventions. All locomotion data are summarized in Supplementary Tables 1 and 2.

## Discussion

Medical imaging-guided deployment of MMDs in physiological environments faces challenges such as detecting tiny, low-contrast objects in noisy, occluded scenes, limiting real-time tracking and precise control. Thus, we have developed a synthetic data generation pipeline (MicroSyn-X) to train CV models for robotic MMD navigation. Using diffusion-based synthesis, MicroSyn-X generates realistic X-ray scenes with automatic, pixel-accurate labels. Domain randomization broadens simulated physiological conditions, improving generalization to unseen clinical settings. The framework has been validated on soft and liquid MMDs in ex vivo and in vivo tissues, achieving performance comparable to clinical experts. Integrated into a robotic system, MicroSyn-X enables multi-robot navigation in 3D lumens and

continuous tracking under bone occlusion and in vivo. In addition, we have released the open-sourced MMD dataset to foster reproducibility in medical robotics.

The proposed framework advances MMD localization and deployment under X-ray fluoroscopy by overcoming key limitations of existing methods. MicroSyn-X expands medical data synthesis to MMD-specific conditions, generating high-fidelity X-ray images that incorporate realistic noise, occlusion and low-contrast scenarios. It bridges the synthetic-to-real gap, demonstrating the feasibility of training models exclusively on synthetic data to perform robustly in clinical environments. With this framework, inexpensive high-quality data with large volume, expanded distribution and accurate labelling is obtained to train generalizable downstream models. Furthermore, the robotic system serves as a functional platform for translating these advancements into clinical applications. This work facilitates clinical translation of MMDs in minimally invasive procedures, targeted therapies and diagnostics.

The proposed system can be improved in multiple aspects. First, more advanced generative models and physics-based deformation models can be adopted to produce more realistic X-ray images that closely mimic real-world anatomical and device-specific features[106]. Moreover, integrating domain knowledge, such as biomechanical models of tissue deformation, could generate time-resolved datasets reflecting physiological motion[101,107,108]. Furthermore, advanced image fusion techniques could seamlessly embed MMDs into anatomical backgrounds with accurate 3D poses and textures[109], while multimodal fusion with ultrasound or magnetic resonance imaging could address X-ray limitations in capturing microscopic biological interactions[110,111]. Second, other downstream CV models, such as transformer-based models[37], can be utilized to enhance the tracking performance. Temporal models for video-based tracking could shift from frame-wise detection to continuous localization, improving efficiency in dynamic fluoroscopic sequences[112,113], while extending MicroSyn-X to 3D segmentation could enable navigation in volumetric X-ray data[114]. Third, reinforcement learning could be adopted for autonomous MMD control[46], along with digital twin interfaces to facilitate better visualization[115]. Last, more comprehensive in vivo validation is necessary to evaluate the system's effectiveness in rare anatomical pathologies and long-term biocompatibility.

## Methods

### Hardware of the teleoperated robotic system

The six-degree-of-freedom (DOF) magnetic actuation platform comprises a cubic permanent magnet (N45, IMPLOTEX GmbH) driven by a NEMA 17 stepper motor (RS Components) and mounted on a 7-DOF robotic arm (Panda, Franka Emika GmbH). Visual feedback was provided by a C-arm fluoroscopy system (Fluoroscan InSight FD, Hologic GmbH). System control was split between a slave computer for robotic arm control and a host computer for fluoroscopic visualization, user input and command transmission. This configuration enables precise six-degree-of-freedom magnetic field control at the end-effector. During navigation, fluoroscopy settings were maintained above 50 kV and 50 μA, and MMD translation speed was limited to <3 mm s$^{-1}$ to minimize motion blur (Extended Data Fig. 1).

Static MMD datasets were acquired using an X-ray cabinet system (XPERT 80, KUBTEC Scientific). A 20-mm cubic magnet (N45, IMPLOTEX GmbH) was actuated using two translational motorized stages (LTS300/M, Thorlabs Inc.) equipped with a stepper motor (535-0372, RS Components GmbH) and a servo motor (SKU 900-00360, Parallax Inc.). Sample height adjustments were performed using an additional LTS300/M translational stage. Together, these components constituted a 5-DOF robotic system[3].

### Diffusion model training and inference

The diffusion model was trained to generate realistic X-ray tissue backgrounds conditioned on anatomical masks and textual prompts

(Supplementary Fig. 3). The image number for each tissue category was less than 20 (Supplementary Fig. 13), and each image was automatically segmented into three channels representing tissue regions ($M_{tissue}$), metallic devices ($M_{device}$) and lumens with contrast agents ($M_{lumen}$), using thresholding-based methods optimized for each channel. To enhance dataset diversity and improve texture learning, geometric transformations and colour–space augmentations were applied during training. A programmatic prompt generation strategy was implemented to automate textual conditioning. A small, fixed vocabulary of anatomical terms (for example, 'brain', 'skull', 'vessel' and 'lumen') was curated once, and prompts were dynamically assembled through random combinations of these terms during data generation (for example, 'porcine brain within the skull' and 'lumen inside heart'). This approach eliminated per-image manual input and maintained constant human effort regardless of dataset size. Future extensions may incorporate large language models to further enrich prompt diversity[116].

A two-phase quality control strategy was applied during model training and inference. During training, candidate models were periodically evaluated using the SSIM to assess fidelity to real tissue images, Inception V3 feature distributions were visualized via PCA to confirm expanded but consistent domain coverage and qualitative screening was used to ensure realistic textures, illumination and contrast. During inference, generation parameters—including diffusion steps and classifier-free guidance scale ($\rho$)—were tuned to minimize artefacts such as blurriness, grid-like patterns, inconsistent illumination or overly smooth textures (Supplementary Fig. 14).

### Latency of the teleoperated robotic system

We quantified the latency of each processing step to assess the system's real-time performance and stability during dynamic locomotion. The end-to-end process consists of three main stages: image acquisition, image processing and actuation command execution.

**Image acquisition.** X-ray imaging was performed using a C-arm system (Fluoroscan InSight FD, Hologic GmbH) operating at adaptive frame rates depending on the imaging mode. Under a continuous high-resolution mode, the frame rate ranged from 0 to 15 frames per second (f.p.s.), corresponding to a minimum interval of 66.7 ms per frame. Under a continuous standard-resolution mode, the system operated at 0–30 f.p.s. with a minimum frame interval of 33.3 ms, as specified in the manufacturer's datasheet.

**Image processing.** Object localization was achieved using a dual-mode tracking algorithm (Supplementary Fig. 2 and Extended Data Fig. 6). In the local tracking mode—the primary operational mode focused on regions of interest (ROIs)—the average processing time was 21.6 ± 1.6 ms per object (model size of 22.4 M), measured from raw data input to result output. In the global re-initialization mode, used sparingly for comprehensive searches across the entire image, the latency was 333.8 ± 4.9 ms (model size of 22.4 M, 25 patches). All computations were performed on a workstation equipped with an NVIDIA RTX Titan GPU, Intel Xeon 5220 CPU at 2.2 GHz, and 64 GB RAM. The processing throughput is fully compatible with the X-ray acquisition rate of up to 30 f.p.s.

**Actuation command.** For clinical safety, the system incorporates a user-in-the-loop protocol, requiring operator approval before movement execution. After approval, the latency from command dispatch to robotic arm actuation was measured to be <112 ms. The robot's locomotion speed remained below 1.5 mm s$^{-1}$, and new commands were issued after the robot advanced approximately 1–2 mm. This latency is well within acceptable limits for stable dynamic locomotion.

### MMD data preparation and integration

The stent-structured soft MMDs were fabricated by moulding[9], while the oil-based ferrofluids with a density of 1.43 g cm$^{-3}$ and

dynamic viscosities of 8 mPas were from Ferrotec Corporation (Supplementary Fig. 15). As shown in Supplementary Fig. 1, the soft MMDs were imaged with a clean background under X-ray cabinet imaging (XPERT 80, KUBTEC Scientific) with varying voltages and currents. Subsequently, the MMD regions in the resulting images were automatically segmented using the automated thresholding algorithm[97]. The largest MMD contour by area was extracted and rotated horizontally to standardize orientation, with blank regions cropped to isolate the soft MMD data. Ferrofluid images were synthesized using spline curve interpolation.

MMD integration was conducted as in the following steps. First, the targeted MMD pixel value was calculated. Given the original pixel value $v_b(x,y)$ of the background at a target pixel location $(x,y)$ and predefined thresholds $v_{min}$ (lowest value) and $v_{max}$ (highest value), the target value $v_t$ was computed by sampling a random contrast multiplier $\rho \in [\rho_{low}, \rho_{high}]$. The target pixel value was calculated with $v_t(x,y) = \rho \times v_b(x,y)$, and $v_t(x,y)$ is clipped between $v_{min}$ and $v_{max}$. Then, MMD selection and geometric transformation are performed. For each insertion position $(x,y)$ in the list of desired MMD locations: select or generate one MMD image instance $R$, scale $R$ so its height matches a predefined target height ($h$) and imposed random height perturbation and randomly rotate $R$ along with its mask $M$. Last, alpha blend and image composition were done with the blending coefficient $\alpha = sum(v_t(x,y) - v_b(x,y))/sum(v_b(x,y) - v_r(x,y))$ if $(x,y) \in M$ and $v_r(x,y)$ is the pixel value of the MMD. The pixel value of the output image is computed with $v_b(x,y) = \alpha \times v_b(x,y) + (1-\alpha) \times v_r(x,y)$, if $(x,y) \in M$. This process automatically seeds MMD shapes into fluoroscopic images with randomized contrast, size, and location, while preserving control over minimum contrast differences and masking boundaries.

## Measures for handling degradation in image quality

To ensure reliable tracking under dynamic and occasionally low-contrast imaging conditions, we implemented a multi-layered strategy that integrates software- and hardware-level controls. This framework mitigates failures arising from sudden drops in image quality, as systematically characterized in Extended Data Fig. 1, where low voltage or current, high frame rate and rapid MMD motion (>20 mm s$^{-1}$) were identified as primary contributors to degraded fluoroscopic visibility.

**Software strategies.** The tracking algorithm incorporates a filtering module designed to reject false detections caused by poor image quality. Each detection is evaluated using multiple criteria, including: (1) a minimum confidence threshold from the computer vision model, (2) geometric consistency of the detected MMD (width, length and overall dimensions), (3) temporal continuity on the basis of the distance between the current and previous positions, (4) anatomical plausibility relative to the lumen centerline and (5) the recent historical localization success rate computed over the past ten frames. These filtering steps complement the preprocessing pipeline (for example, brightness/contrast adjustment and histogram equalization) and are integrated with the adaptive Kalman filter, which interpolates missing positions during occlusions. The complete algorithmic workflow and pseudocode are provided in Extended Data Fig. 6.

**Hardware and protocol strategies.** To minimize image degradation, fluoroscopy was operated above 50 kV and 50 μA (Extended Data Fig. 1). MMD translation speed was limited to <3 mm s$^{-1}$ to reduce motion blur. If tracking was lost for extended periods, magnetic actuation was adjusted by reducing rotation frequency or repositioning the magnetic field to mitigate occlusion. During continuous acquisition, the fluoroscopy frame rate automatically adapted to object motion, switching to lower frame rates when needed to improve image quality.

**Operator intervention.** If an MMD remains undetected despite the above automated controls, the operator may manually adjust the C-arm angle to improve the imaging perspective, increase voltage and current for enhanced contrast or modify the fluoroscopic field of view. This manual adjustment pathway is also illustrated within the full tracking workflow in Extended Data Fig. 6.

The interactions between detection, filtering and trajectory reconstruction are summarized in Supplementary Fig. 2 and detailed in Extended Data Fig. 6.

## Mask generation with spline curves

The shapes of liquid MMDs and mask of tissue background were programmatically generated with the following steps (Supplementary Fig. 1). The first step was to generate $n$ approximately evenly distributed but randomly perturbed points on a circle of radius $r$ around a centre $c = (c_x, c_y)$, denoted by $P_i = (x_i, y_i)$ for $i = 0, 1, ..., n-1$. Angular sector sampling was first done by uniformly sampling angles around the centre with $\theta_i \sim \mathcal{U}(i(2\pi/n), (i+1)2\pi/n)$, after which the radial sampling was performed with $r_i \sim \mathcal{U}(l_{min}, l_{max})$, where $l_{min}, l_{max}$ are the lower and upper limits of the radius, respectively. After these two steps, each point was calculated with $x_i = c_x + r_i \cos\theta_i$, $y_i = c_y + r_i \sin\theta_i$, and all points were arrange into the array $\mathbf{P} = \{(x_i, y_i)\}_{i=0}^{n-1}$. To produce a smooth, closed curve through $P$, the first point was appended to the end and then fit a periodic B-spline with $\mathbf{P}' = [(x_0, y_0), ..., (x_{n-1}, y_{n-1}), (x_0, y_0)]$. Subsequently, a periodic, smoothing-free ($s = 0$) B-spline was computed and represented as $\mathbf{C}(u) = (X(u), Y(u)), u \in [0,1]$, such that $C(u_j) = P'_j$ for a knot vector $u_j$ of length $n+1$. With the computed B-spline curve[117], $N_{interp}$ interpolated curve points were uniformly re-sampled as $(x_k^{(new)}, y_k^{(new)}) = (X(u_k^{(new)}), Y(u_k^{(new)}))$ with $u_k^{(new)} = k/(N_{interp} - 1)$ and $k = 0, ..., N_{interp} - 1$. These points were arranged as the final contour as $\mathbf{C}_{interp} = \{(x_k^{(new)}, y_k^{(new)})\}_{k=0}^{N_{interp}-1}$.

## Dataset preparation

Synthetic data were automatically generated and used to train a model referred to as model (syn.). Real MMD data were divided into soft and liquid types, each imaged under static and locomotion conditions. Static imaging placed MMDs in phantoms or biological tissues using an X-ray cabinet system (XPERT 80, KUBTEC Scientific). Locomotion imaging recorded videos with a C-arm system (Fluoroscan InSight FD, Hologic GmbH) mounted on a robotic arm (Panda, Franka Emika GmbH). Video frames were analyzed with model (syn.) and manually checked: outputs matching expert identification were adopted as labels, while discrepancies were manually annotated if the MMD was identifiable. Labels followed a one-text-file-per-image format, with each row indicating a single object's class index and polygonal contour coordinates: class index, $x_1, y_1, x_2, y_2, ..., x_n, y_n$.

For evaluating the model performance, ROIs that centred on the MMD or excluded the MMD were extracted. The locomotion dataset was categorized according to tissue type. For soft MMDs, tissue categories included porcine brain with embedded bones, porcine brain alone, heart, liver, stomach, heart 3D vessels, in vivo rat models and in vivo rabbit models. For liquid MMDs, tissue types encompassed porcine brain (with and without embedded bones), heart, liver and stomach, as well as scenarios involving MMD swarms under bone occlusion. The datasets and model weights (diffusion model and instance segmentation model) are available at ref. 118. Custom Python code was used for labelling and analyzing the data leveraging LABELME (5.5.0), MATPLOTLIB (3.7.3), NUMPY (1.24.4), SCIPY (1.10.1), PYTORCH (1.11.0) and PANDAS (1.4.4) packages.

## CV model training and inference

When the MMD occupied a small region within a large field of view, ROIs centred on the MMD were extracted for segmentation training. ROIs were split into training and validation sets at ratios of 20:1 for synthetic data and 10:1 for real data. Data augmentation included geometric transformations (scaling and translation of 0.5, rotation of ≤90° and shear of 30°), colour augmentation in the hue, saturation, value (HSV)

space (brightness factor of 0.8) and robustness-enhancing operations such as perspective distortion (0.0002), vertical flipping (50%), random erasing (20%) and copy–paste augmentation (5%). Four model variants (2.8M, 10.1M, 22.4M and 27.6M parameters) were trained for 80 epochs with a batch size of 20 using a cosine learning rate scheduler (with an initial rate of 0.01 and a final rate of 0.0001).

For model inference, we implemented a dynamic dual-mode localization framework that alternates between processing the entire frame and focused patches to optimize both accuracy and speed. The overall workflow is illustrated in Supplementary Fig. 2 and Extended Data Fig. 6. In the 'global search mode', the frame is subdivided into overlapping patches for processing. This mode is used sparingly—for initialization or recovery for lost MMDs—to ensure comprehensive scene coverage. The 'local tracking mode' serves as the primary operational mode for real-time tracking. Once the MMD's position is identified, the model processes only a cropped ROI around the last known position, rather than the entire frame. This hybrid strategy enables real-time performance by combining the speed of the local tracking mode with the robustness of the global search mode.

### Computation of metrics

For classification, each ROI is assigned to one of two classes: MMD or non-MMD. The maximum object detection confidence score within the ROI was used as the predicted probability of an MMD being present in the image. AP was used to evaluate the performance of binary or multi-class classification models by summarizing the precision-recall curve. AP calculates the weighted mean of precision values achieved at each confidence threshold, with weights determined by the change in recall between thresholds. Specifically, the score is derived using the formula $AP = \sum_n (r_n - r_{n-1}) \cdot p_n$, where $r_n$ and $r_{n-1}$ represent consecutive recall values, and $p_n$ is the precision at threshold $n$.

Detection and segmentation performance were evaluated using mAP50(B) and mAP50(M), which measure mean AP for bounding boxes and masks, respectively, at an IoU threshold of 0.5. IoU is defined as the overlap between predicted and ground-truth regions divided by their union, with predictions considered correct when IoU was >0.5. More stringent metrics, mAP50:95(B) and mAP50:95(M), average mAP over IoU thresholds from 0.5 to 0.95 in 0.05 increments, thereby assessing robustness and spatial precision of both bounding boxes and segmentation masks.

Contrast and noise were calculated using region-based analysis, where three regions are manually defined: the object region ($M_{obj}$), background region ($M_{back}$) and noise region ($M_{noise}$). Contrast was determined using the Michelson contrast formula: $contrast = (mean(v(x_{obj}, y_{obj})) - mean(v(x_{back}, y_{back}))) / (mean(v(x_{obj}, y_{obj})) + mean(v(x_{back}, y_{back})))$, if $(x_{obj}, y_{obj}) \in M_{obj}$ and $(x_{back}, y_{back}) \in M_{back}$. For noise estimation, the median absolute deviation of pixel intensities in $M_{noise}$ was calculated as the median of absolute deviations from the median intensity, then scaled by 1.4826 to approximate the standard deviation of Gaussian noise.

### Ex vivo tissue phantom preparation

Organs were obtained as animal by-products (registration number DE 08 111 1008 21) under permits issued by the Stuttgart state authorities for food control, consumer protection and veterinary services. In compliance with permit requirements, biomaterial use was documented and all samples were pressure-sterilized after experiments. Coronary arteries for locomotion and mechanical testing were isolated from fresh porcine hearts within 48 h post-slaughter, stored at 4 °C, and sourced from Slaughterhouse Ulm (Germany) and Gourmet Compagnie GmbH (Germany). Before testing, tissues were rinsed with phosphate-buffered saline. For ex vivo experiments, phosphate-buffered saline (pH 7.4, Gibco, Thermo Fisher Scientific) was perfused through the arteries at 10–12 ml min⁻¹, and angiographic imaging was performed using Iomeron 400 contrast agent (Bracco UK Limited).

Agarose gel samples with internal lumens were fabricated using 3D-printed positive moulds (Form 3, Formlabs Inc.). Agarose powder (A9539, Sigma-Aldrich) was dissolved in deionized water at 90 °C, boiled for 5 min, poured into Petri dishes containing the moulds, and cooled at room temperature (~24 °C) for 30 min before mould removal. The resulting agarose lumens were embedded in or placed beneath tissue samples to assess X-ray imaging performance of medical devices[3].

### Setup for in vivo animal testing

This animal study was approved by the Committee on Institutional Animal Care and Use Committee of Hong Kong Huateng Biotechnology Co., Ltd. (IACUC number B202502-25) and the Institutional Animal Research Ethics Sub-Committee of City University of Hong Kong (AN-STA-00001025). New Zealand White rabbits (*Oryctolagus cuniculus*), outbred albino stock (genetic background: outbred), aged 4 months and weighing 2.5–3.0 kg at the time of experimentation, were obtained from Guangzhou Xindongxinhua Experimental Animal Breeding Farm (Guangzhou, China). Sprague Dawley rats (*Rattus norvegicus*), outbred stock (genetic background: outbred), aged 4 months and weighing approximately 700 g at the time of experimentation, were obtained from Zhuhai Bestone Biotechnology Co., Ltd. (Zhuhai, China). Both species are standard, commercially available outbred laboratory animals with no genetic modifications. All procedures were performed under anaesthesia to ensure animal welfare. The MMD device was introduced into the femoral artery of rabbits and the aorta of rats via a 4-Fr sheath (Glidesheath, Terumo), following a small incision. Real-time deployment was monitored using X-ray fluoroscopy (DSA, CGO-2100, Wandong Co. Ltd.). Upon reaching the targeted vascular region, the MMD was retrieved through magnetic actuation or by wire traction. Finally, the sheath was removed, and the surgical wound was sutured following standard protocols.

To ensure haemocompatibility and biocompatibility, the MMD surface was coated with a 1 μm layer of parylene C (SCS Labcoter 2, Specialty Coating Systems). Parylene C is Food and Drug Administration-approved for blood-contacting medical devices and has a long history of use in stents, guidewires and catheters[119–121]. Our previous work demonstrated that parylene C-coated polydimethylsiloxane films tolerate repeated large-deformation bending without delamination[9], supporting short-term mechanical and chemical stability. Long-term stability under chronic physiological conditions, however, requires further investigation.

### Histological examination

The MMD was injected into the femoral artery of the rabbit and rotated under the rotating magnetic field. The part of the artery contacting the MMD was cut out for histological examinations. Histological processing, performed by the Guangzhou Huitong Medical Code Pathology Diagnostic Center, included paraffin embedding, sectioning, hematoxylin and eosin staining, Masson's trichrome staining, white light microscopy and qualitative analysis. The histological images (Supplementary Fig. 12) showed that the vascular lumen was open with no evidence of obstruction or thrombus formation. Endothelial cells of the vessel wall were arranged regularly, with no signs of hyperplasia or detachment. The internal elastic lamina was intact, with no signs of loss or rupture. The tunica media was composed of circumferentially arranged smooth muscle cells, showing no damage. The tunica adventitia consisted of loose connective tissue and appeared undamaged. Masson's trichrome staining revealed no visible fibrous tissue proliferation in the intima, media or adventitia of the vessel wall. Furthermore, detailed examinations of biocompatibility and haemocompatibility were done in our previous studies[9].

### Reporting summary

Further information on research design is available in the Nature Portfolio Reporting Summary linked to this article.

## Data availability

The datasets used in this study—including the synthetic MMD data, real MMD data, real MMD locomotion data and other data referenced in the Methods section—are available at https://huggingface.co/data-sets/luoyeguigenno1/X-ray-Miniature-Medical-Device (ref. 118). The repository also contains the trained model weights for the diffusion and instance segmentation models. Source data are provided with this paper.

## Code availability

The source code supporting this study is available at https://hugging-face.co/datasets/luoyeguigenno1/X-ray-Miniature-Medical-Device (ref. 118), under the CC BY-NC 4.0 license.

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

## Acknowledgements

We gratefully acknowledge B. Demirhan, Z. Fouladivanda, G. Michailidis, V. Theodori and D.E.T. Şanlı for their contributions in providing clinical expert annotations. Funding: C.W., C.H., S.O.D., Z.L., T.W., H.Z. and M. Sitti received funding from the Max Planck Society, European Research Council Advanced Grant SoMMoR project (grant number 834531) and the European Research Council Proof of Concept STENTBOT project (grant number 101100727). W.K. received funding from the European Union's Horizon 2022 research and innovation program under the Marie Skłodowska-Curie Postdoctoral Fellowship (grant agreement number 101109050) and start-up funding (9610735) from the City University of Hong Kong. M. Sun received start-up funding (A-0010108-00-00) from the National University of Singapore. H.U. received the funding from the Zentrum für Radiologie Heilbronn. K.H. received funding from Shandong University. T. Wang received start-up funding from the University of Hawaiʻi at Mānoa. M. Sitti received funding from the German Research Foundation Soft Material Robotic Systems (SPP 2100) Program (grant number 2197/5-1). C.W., Z.L. and M. Sitti received funding from the Max Planck Queensland Center for the Materials Science of Extracellular Matrices.

## Author contributions

C.W., T.W., W.K. and M. Sitti proposed and planned the project. C.W., M. Sun, W.K., H.U. and T.W. collected the ex vivo raw data. W.K. carried out the in vivo experiments with assistance from T.W. C.W. developed the algorithms and analyzed the data with data annotation assistance from K.H. C.W. set-up the robotic system with assistance from T.W. and S.O.D. C.W. prepared the robots with assistance from C.H., M. Sun, T.W., H.Z. and Z.L. M. Sitti and T.W. supervised the research. C.W. wrote the paper, revised by W.K., T.W. and M. Sitti. All authors discussed the results and commented or edited the paper.

## Funding

## Competing interests

M.S. and T.W. are listed as inventors on pending US patent application 18/133104 and European patent application 22167673.7 submitted by the Max–Planck–Gesellschaft zur Forderung der Wissenschaften e.V. that covers the fundamental design, fabrication and control principles of the stent-shaped magnetic soft robots included in this work. The other authors declare no competing interests.

## Additional information

**Extended data** is available for this paper at https://doi.org/10.1038/s42256-026-01190-3.

**Correspondence and requests for materials** should be addressed to Wenbin Kang, Tianlu Wang or Metin Sitti.

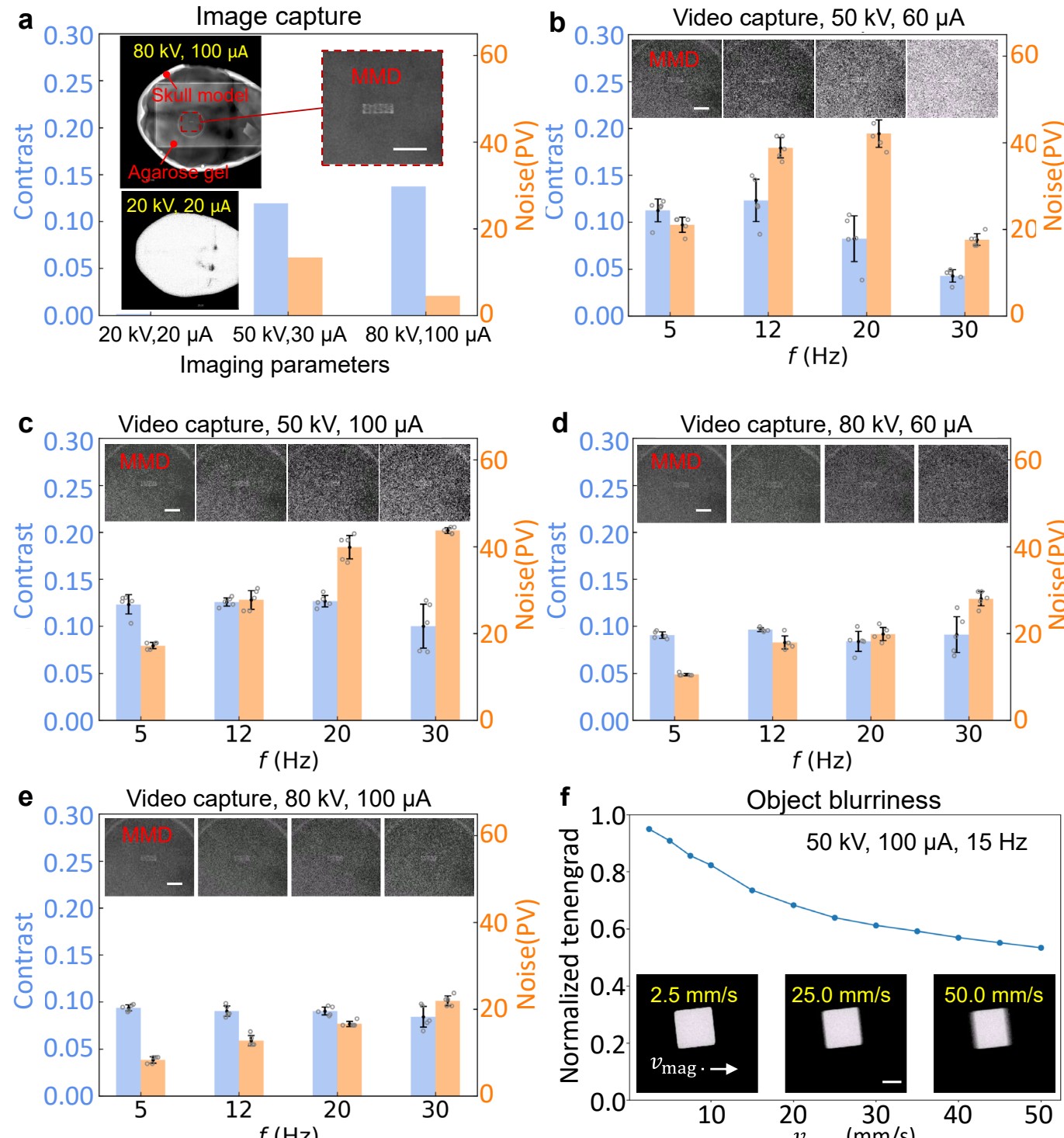

**Extended Data Fig. 1 | Effect of imaging parameters on image quality. a.** Impact of X-ray voltage and current on image contrast and noise in static capture mode. **b–e.** Influence of voltage, current, and frame capture rate ($f$) on contrast and noise in continuous capture mode. Data are presented as mean values ± standard deviation. Error bars denote standard deviation across five images. **f.**

Quantification of motion-induced blurriness as a function of magnet translation speed ($v_{mag}$) under continuous capture. Blurriness is assessed via the normalized Tenengrad sharpness metric, where lower values indicate increased motion blur. All experiments were performed using the XPERT 80 imaging system (KUBTEC Scientific). In all figures, scale bars represent 5 mm.

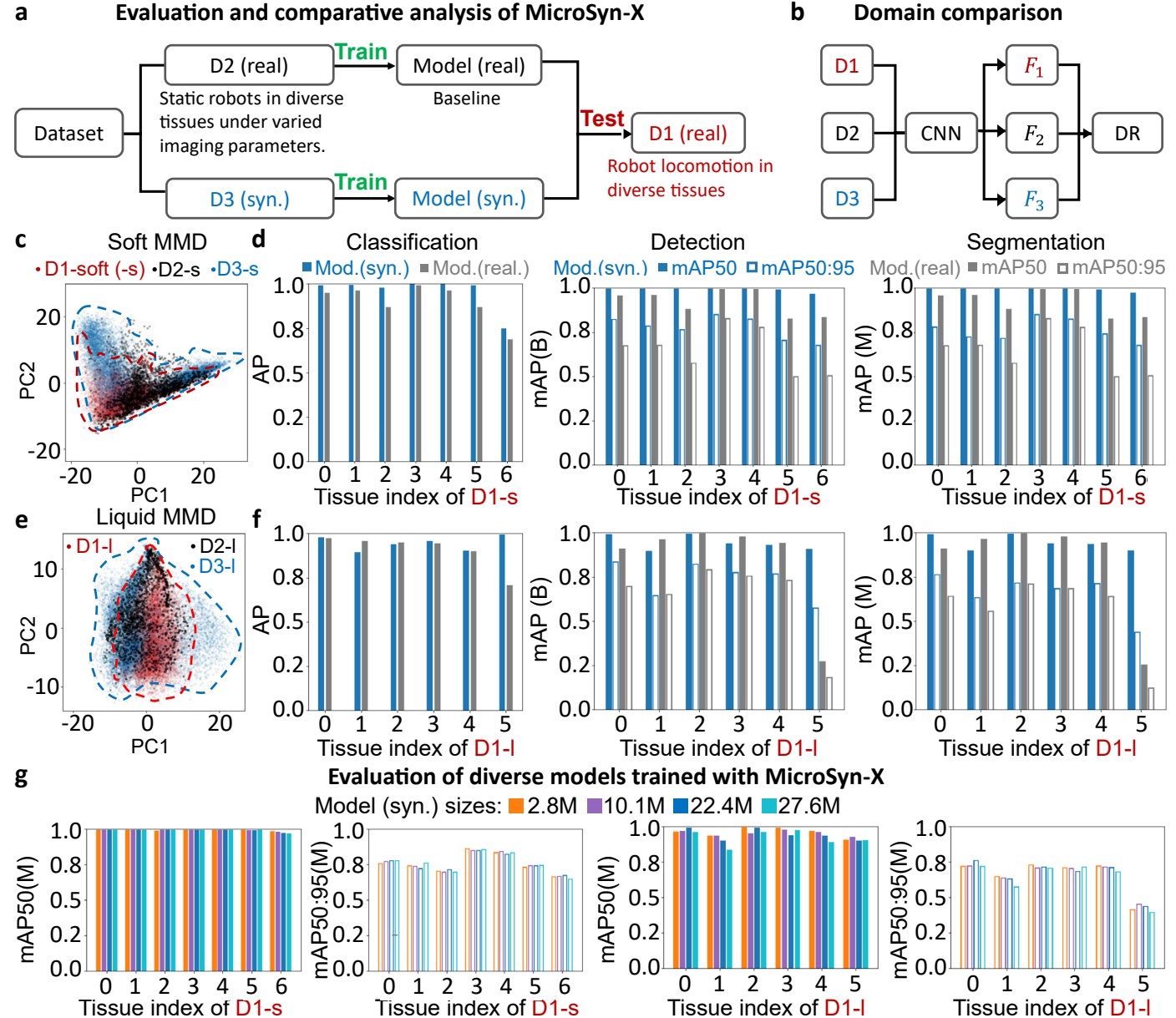

**Extended Data Fig. 2 | Evaluation of MicroSyn-X. a**. Schematic of the evaluation process. **b**. Cross-domain feature comparison using CNN-extracted features visualized by dimensionality reduction (DR). **c**. Comparison of soft MMD datasets via PCA, with dashed lines indicating data distribution boundaries. **d**. Performance of models trained on synthetic and real soft MMD data across classification, detection, and segmentation tasks, measured by AP, mAP50, and mAP50:95. mAP(B) and mAP(M) denote bounding-box and mask mAPs, respectively. Tissue index indicates datasets: soft MMDs in porcine brain with embedded bones, porcine brain, heart,

liver, stomach, heart 3D vessels, and in vivo animals. Models with a parameter size of 10.1 M is utilized. **e**. Cross-domain comparison of liquid MMD datasets. **f**. Performance comparison between models trained on synthetic and real liquid MMD data. Tissue index indicates datasets: liquid MMDs in porcine brain, heart, liver, stomach, brain or liver with embedded bones, and robot swarm under bone occlusion. **g**. Performance evaluation of models trained on synthetic data of soft and liquid MMDs, with varying model sizes (2.8 M, 10.1 M, 22.4 M, 27.6 M).

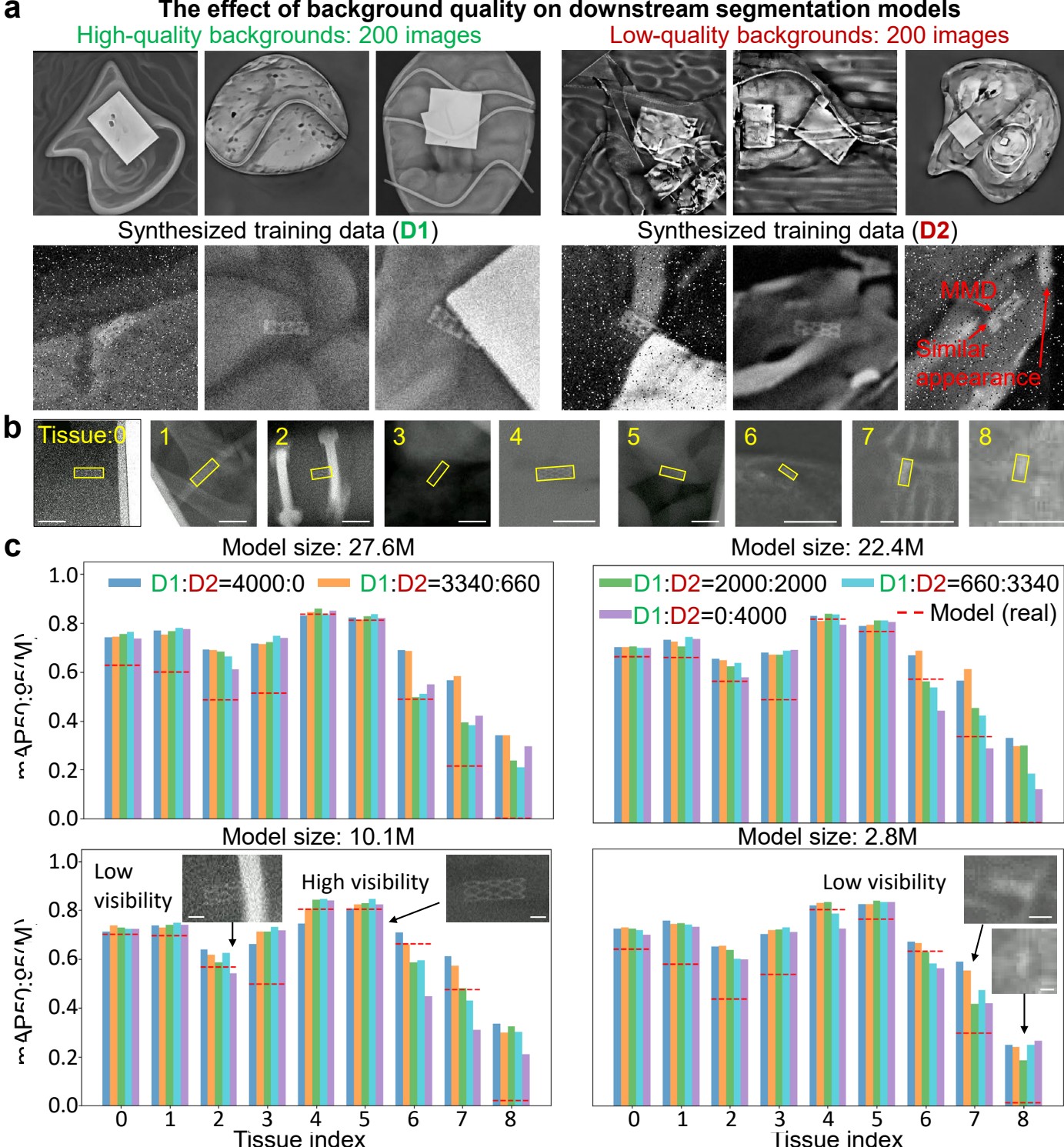

**Extended Data Fig. 3 | Effect of synthetic background quality on downstream segmentation models. a.** Dataset composition. 200 high- and 200 low-quality generated tissue backgrounds were used to construct five synthetic training sets with varying proportions of high-quality backgrounds. **b.** Real-tissue test dataset. The test images include soft MMDs in porcine brain with embedded bone, porcine brain, heart, liver, stomach, heart 3D vessels, in vivo rabbit, in vivo rat, and in vivo rat spine at low magnification. Scale bars represent 5 mm. **c.** Segmentation performance across background quality ratios and model sizes. For structurally distinct targets (for example, stents) with high visibility,

performance remains stable or improved, where low-quality backgrounds provide beneficial augmentation. In contrast, for low-contrast or ambiguous targets (for example, blurred white rectangles resembling background artefacts), inclusion of a modest fraction of low-quality backgrounds (ratio = 0.86) improves or maintains performance for larger models, but degrades performance for smaller models, evidenced by missed detections. The red dashed lines represent the result of model trained on real data. Scale bars represent 1 mm.

## Robustness evaluation: doctor clinical labels and MicroSyn-X

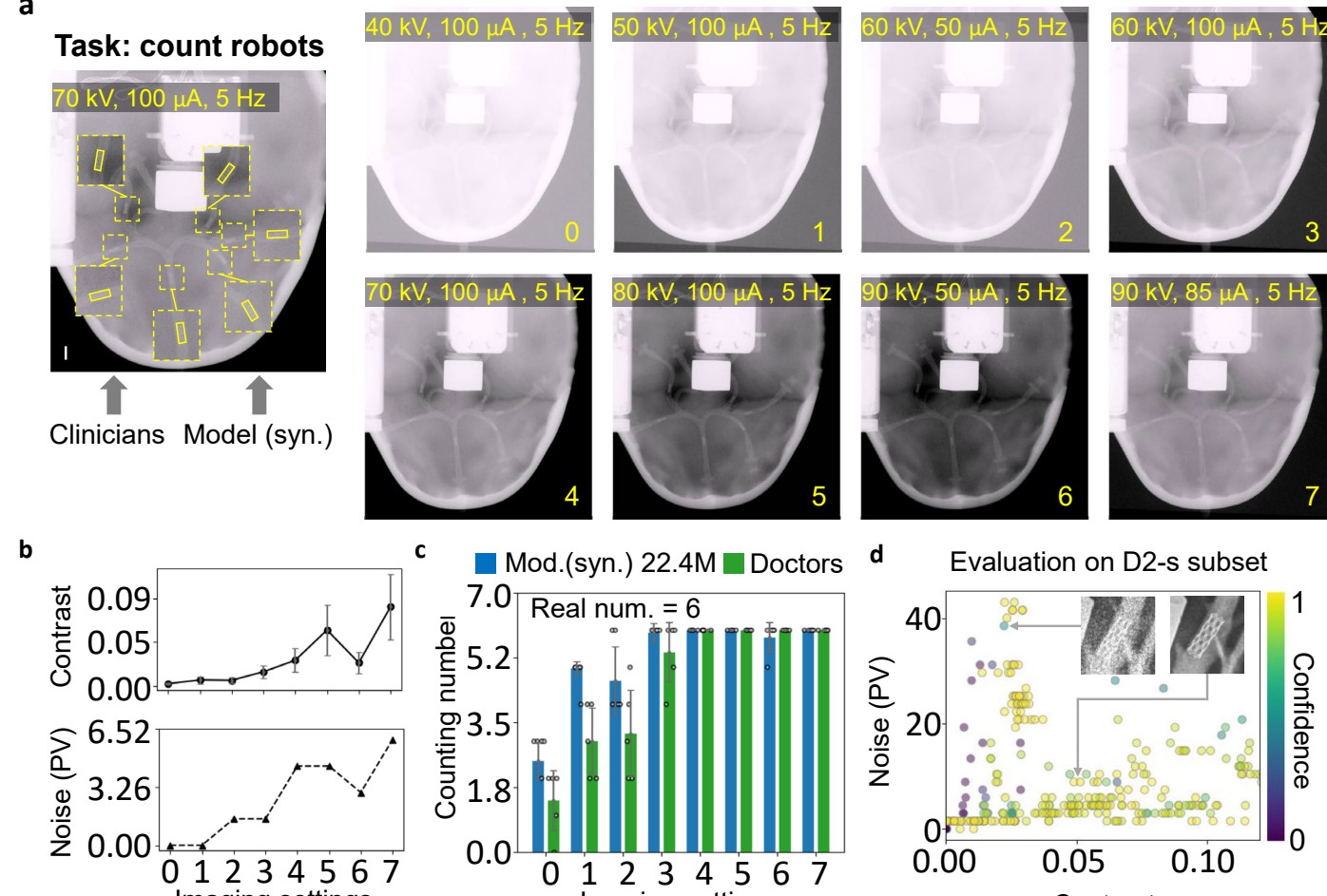

**Extended Data Fig. 4 | Robustness evaluation of MicroSyn-X. a. Evaluation task comparing clinical expert labels with MicroSyn-X.** Videos of six static robots inside a skull model were recorded across varying imaging parameters for quantifying detectable robots. The numbers represent the indexes of imaging setting. **b.** The imaging effect of MMDs under different imaging settings. Error bars represent standard deviation across six different MMDs. **c.** Expert-label comparison. Model-derived robot counts and expert annotations are evaluated across eight imaging parameters. Error bars represent standard deviation across five clinicians and model detection confidences. **d.** Detection confidence distribution relative to contrast and noise levels. Effective detections require a segmentation mask IoU > 0.5 with ground truth. A subset of dataset D2-s was manually annotated, with defined contrast and noise computation regions defined. In all figures, data are presented as mean values ± standard deviation.

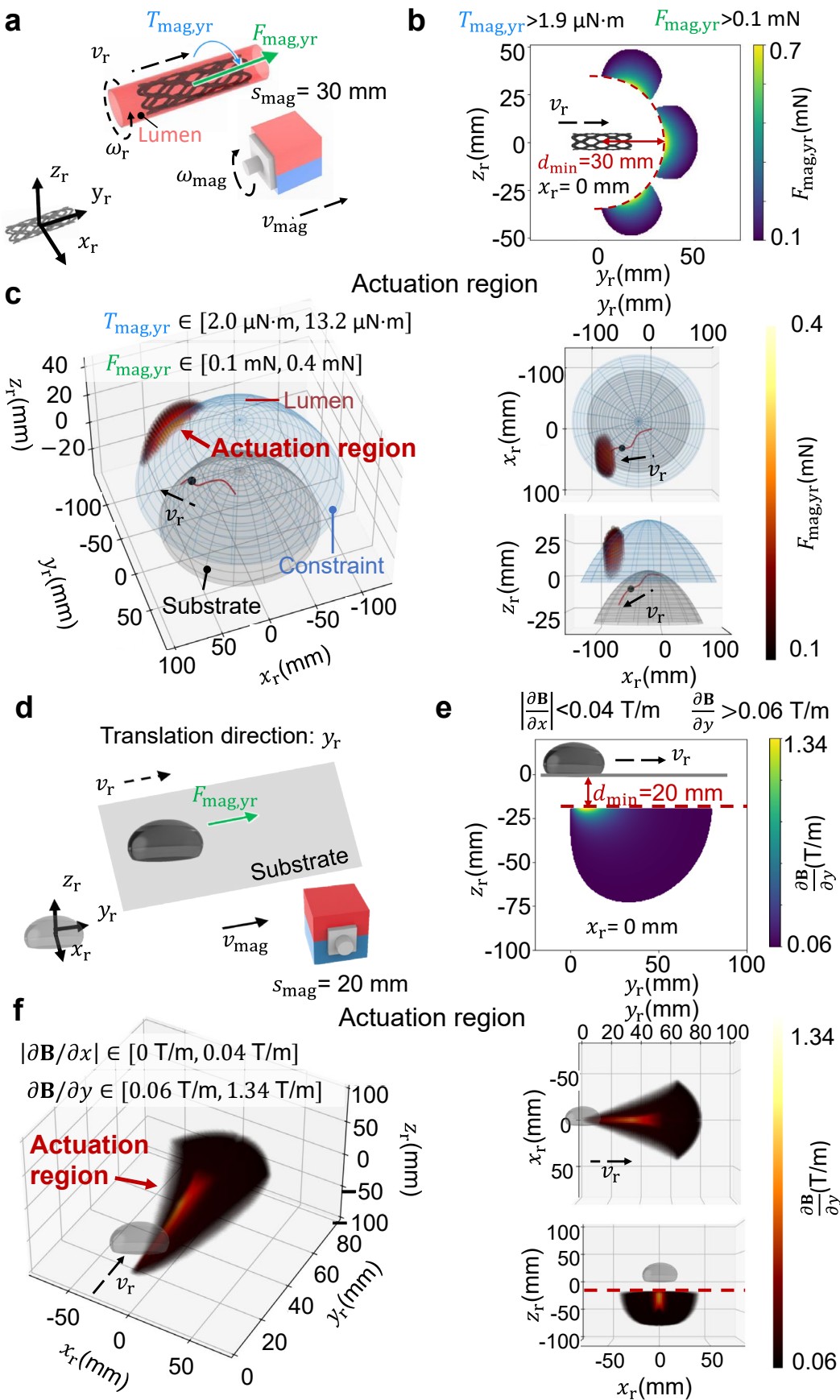

**Extended Data Fig. 5 | See next page for caption.**

**Extended Data Fig. 5 | Actuation regions of soft and liquid MMDs. a**. Actuation of the soft MMD inside a lumen, driven by magnetic torque and force. **b**. Actuation region of the soft MMD in the plane of $x_r = 0$. When the rotating permanent magnet (PM) is positioned within this region, the magnetic torque and force exceed the thresholds to enable MMD motion. **c**. Three-dimensional actuation region of the soft MMD, considering geometric constraints to prevent potential collisions. **d**. Actuation of the soft MMD on a substrate, where motion is achieved by utilizing the magnetic field gradient. **e**. Actuation region of the liquid MMD in the plane of $x_r = 0$. When the PM is located within this region, the magnetic field gradient along the desired direction exceeds the threshold to induce MMD movement. **f**. Three-dimensional actuation region of the liquid MMD. A larger PM can be employed to expand the actuation region.

---

**Algorithm 1** Object Tracking

---

 1: **Initialize:** $t \leftarrow 0$ ▷ Frame index
 2: $\mathcal{T} \leftarrow \emptyset$ ▷ Set of active tracks
 3: $\mathcal{K} \leftarrow \emptyset$ ▷ Set of Kalman filters
 4: $\mathcal{C} \leftarrow \emptyset$ ▷ Set of candidate detections with counters
 5: $T_{\text{confirm}} \leftarrow 5$ ▷ Frame threshold for track confirmation
 6: **procedure** PROCESSFRAME($I_t$)
 7: **if** $t = 0$ **then**
 8: $\mathcal{D}_t \leftarrow$ global($I_t$) ▷ Initial global detection $d_i \in \mathcal{D}_t$
 9: ROI$_i \leftarrow$ createROI($d_i$)
10: $\mathbf{x}_i^0 \leftarrow [p_x, p_y, v_x, v_y]^\top$ ▷ Initial state
11: $\mathcal{K}_i \leftarrow$ KalmanInit($\mathbf{x}_i^0, \mathbf{P}_0$)
12: $\mathcal{T} \leftarrow \mathcal{T} \cup \{\tau_i\}$
13: age($\tau_i$) $\leftarrow 1$
14:
15: **else**
16: **Step 1: Prediction for existing tracks** $\tau_i \in \mathcal{T}$
17: $\hat{\mathbf{x}}_i^t \leftarrow \mathcal{K}_i.()$ ▷ Kalman prediction
18: ROI$_i \leftarrow$ updateROI($\hat{\mathbf{x}}_i^t$)
19:
20: **Step 2: Detection and association for existing tracks** $\tau_i \in \mathcal{T}$
21: $\mathcal{D}_i \leftarrow$ local($I_t$, ROI$_i$)
22: **if** $\mathcal{D}_i = \emptyset$ **then**
23: $\mathcal{D}_i \leftarrow$ global($I_t$)
24: **if** $\mathcal{D}_i = \emptyset$ **then**
25: stopActuation()
26: retractMagnet()
27: $\mathcal{D}_i \leftarrow$ global($I_t$) ▷ Re-attempt detection
28: **end if**
29: **end if**
30: $d_i^* \leftarrow (\hat{\mathbf{x}}_i^t, \mathcal{D}_i)$
31: **if** $d_i^* \neq \emptyset$ **then**
32: $\mathcal{K}_i.(\mathbf{z}_i^t)$ ▷ Update with measurement
33: age($\tau_i$) $\leftarrow$ age($\tau_i$) $+ 1$
34: **end if**
35:
36: **Step 3: New candidate detection and confirmation**
37: $\mathcal{D}_t^{\text{new}} \leftarrow$ global($I_t$) $\setminus$ {associated detections} $d_j \in \mathcal{D}_t^{\text{new}}$
38: **if** $d_j \in \mathcal{C}$ **then** ▷ Existing candidate
39: $c_j \leftarrow c_j + 1$ ▷ Increment counter
40: **if** $c_j \geq T_{\text{confirm}}$ **then**
41: Initialize new track $\tau_j$
42: Initialize new Kalman filter $\mathcal{K}_j$
43: $\mathcal{T} \leftarrow \mathcal{T} \cup \{\tau_j\}$
44: $\mathcal{C} \leftarrow \mathcal{C} \setminus \{d_j\}$ ▷ Remove from candidates
45: **end if**
46: **else**
47: $\mathcal{C} \leftarrow \mathcal{C} \cup \{d_j\}$ ▷ Add new candidate
48: $c_j \leftarrow 1$ ▷ Initialize counter
49: **end if**
50:
51: **Step 4: Cleanup stale candidates** $d_j \in \mathcal{C}$
52: **if** detection in current frame matching $d_j$ **then**
53: $c_j \leftarrow \max(0, c_j - 1)$ ▷ Decrement counter
54: **if** $c_j = 0$ **then**
55: $\mathcal{C} \leftarrow \mathcal{C} \setminus \{d_j\}$ ▷ Remove stale candidate
56: **end if**
57: **end if**
58:
59: **end if**
60: $t \leftarrow t + 1$
61: **end procedure**

---

**Extended Data Fig. 6 | Pseudocode of the object tracking algorithm.**

**Extended Data Table 1 | External sensing modalities for in vivo MMDs**

| Technology | Advantages | Limitations | Application Scenarios | Functions | Reported Accuracy | Reference |
|---|---|---|---|---|---|---|
| X-ray / Digital Subtraction Angiography (DSA) | Real-time 2D images, templates with radio-opaque devices | Ionizing radiation exposure, requires intravascular contrast agents | Neurovascular, cardiovascular vasculature | Guiding wire and catheter placement, procedural accuracy monitoring | Spatial: ~0.1 mm; Temporal: >10 fps | Frisken et al.[19] |
| Computed Tomography (CT) | Providing 3D visualization, eliminating anatomic superimposition | Ionizing radiation exposure, not real-time | Liver, kidney, lung | Guiding needle insertion, biopsy planning, lesion targeting | Spatial: ~0.4 mm | Hsieh et al.[22] |
| Computed Tomography Angiography (CTA) | Providing 3D visualization, compatible with metallic implants | Ionizing radiation exposure, not real-time | Neurovascular, cardiovascular vasculature | Providing 3D road-map for medical procedures | Spatial: ~0.5 mm | Fleischmann et al.[17] |
| Magnetic Resonance Imaging (MRI) | Excellent soft tissue imaging, no ionizing radiation | Not suitable for real-time imaging, not compatible with metallic tools | Neurovascular, cardiovascular vasculature | Tracking and guidance of devices | Spatial: 1-2 mm; Temporal: ~0.5 fps | Lim et al.[23] and Geethanath et al.[20] |
| Ultrasound Imaging (US) | Non-ionizing real-time imaging, portable for bedside and surgery | Acoustic shadowing, low-resolution imaging | Heart, liver, kidney, peripheral vessels | Guiding and tracking of minimally invasive devices | Spatial: 0.2-1 mm; Temporal: 20-50 fps | Ng et al.[26] |
| Laser Speckle Contrast Imaging (LSCI) | Real-time, high temporal resolution, safe non-contact | Limited spatial resolution, shallow penetration depth | Skin, superficial vasculature | Tracking and navigation of devices for targeted delivery | Relative error: 1% - 5% | Olmos et al.[21] |
| Opto-acoustic Imaging (OA) | High optical contrast with ultrasound resolution, deeper penetration than pure optical methods | Limited clinical adoption, image reconstruction complexity | Skin, breast, thyroid, synovial joints, vasculature | Monitoring vascular interventions, guiding biopsies, mapping sentinel lymph nodes | Spatial: 20-200 μm (scalable with depth) | Tian et al.[29] |

**Extended Data Table 2 | Intraluminal sensing modalities for in vivo MMDs**

| Technology | Advantages | Limitations | Application Scenarios | Functions | Reported Accuracy | Reference |
|---|---|---|---|---|---|---|
| Fiber Bragg Grating Sensing (FBG) | High sensitivity, immune to electromagnetic interference, multi-parameter measurements, compact and biocompatible | Mechanical hysteresis, signal drift over time | Bronchus, GI track | Real-time 3D shape reconstruction, contact force detection | Shape: <1 mm; Force: ~5 mN | Najafzadeh et al.[25] |
| Electromagnetic (EM) Tracking | Non-line-of-sight real-time 3D positioning, measures both position and orientation | Interfered by ferromagnetic materials, limited workspace | Bronchus, vasculature | Pose and trajectory tracking of robotic end-effectors | Position: 0.5-1.5 mm RMS; Orientation: 0.1°-0.3° RMS | Yaniv et al.[30] |
| Proprioceptive Force Sensing | Improved human-robot interaction | Hard to decouple distal contact force and known friction, input delay | Kidney, vasculature | Measuring interaction forces based on motor current changes | Highly system-dependent: ~2 mN; Linearity error: ~1% | Lv et al.[24] |
| Tactile Force Sensing | Direct measurement of the contact force at the device/tissue interface | Harsh sterilization, risk of electrical hazard and biocompatibility | Prostate, liver | Providing haptic feedback, collision detection and prediction | ~55 mN | Du et al.[16] |
| Intravascular Ultrasound (IVUS) | Direct visualization of vessel walls, high resolution | Motion artifacts, limited information on structures outside the vessel | Cardiac vasculature, bronchus, GI tract | Guiding catheter-based procedures (e.g., atherectomy, tissue characterization) | Axial: ~20-200 µm; Lateral: 150-400 µm | Peng et al.[27] |
| Optical Coherence Tomography (OCT) | Micron-level resolution, high sensitivity and accuracy | Limited penetration depth (1-20 mm), sensitive to motion | Cardiac vasculature, bronchus, GI tract | Optical biopsy and device guidance | Axial: 5-20 µm; Lateral: 10-90 µm | Folgar et al.[18] and Spaide et al.[28] |

# Reporting Summary

## Statistics

For all statistical analyses, confirm that the following items are present in the figure legend, table legend, main text, or Methods section.

| n/a | Confirmed | |
|---|---|---|
| ☐ | ☒ | The exact sample size (*n*) for each experimental group/condition, given as a discrete number and unit of measurement |
| ☐ | ☒ | A statement on whether measurements were taken from distinct samples or whether the same sample was measured repeatedly |
| ☐ | ☒ | The statistical test(s) used AND whether they are one- or two-sided *Only common tests should be described solely by name; describe more complex techniques in the Methods section.* |
| ☐ | ☒ | A description of all covariates tested |
| ☐ | ☒ | A description of any assumptions or corrections, such as tests of normality and adjustment for multiple comparisons |
| ☐ | ☒ | A full description of the statistical parameters including central tendency (e.g. means) or other basic estimates (e.g. regression coefficient) AND variation (e.g. standard deviation) or associated estimates of uncertainty (e.g. confidence intervals) |
| ☐ | ☒ | For null hypothesis testing, the test statistic (e.g. *F*, *t*, *r*) with confidence intervals, effect sizes, degrees of freedom and *P* value noted *Give P values as exact values whenever suitable.* |
| ☐ | ☒ | For Bayesian analysis, information on the choice of priors and Markov chain Monte Carlo settings |
| ☐ | ☒ | For hierarchical and complex designs, identification of the appropriate level for tests and full reporting of outcomes |
| ☒ | ☐ | Estimates of effect sizes (e.g. Cohen's *d*, Pearson's *r*), indicating how they were calculated |

*Our web collection on statistics for biologists contains articles on many of the points above.*

## Software and code

Policy information about availability of computer code

| | |
|---|---|
| Data collection | Custom Python code was used for labelling the data leveraging LABELME package  (5.5.0). The commercial x-ray imaging devies of XPERT 80, KUBTEC Scientific and Fluoroscan InSight FD, Hologic GmbH were used to capture the image. |
| Data analysis | Custom Python code was used for labelling and analyzing the data leveraging LABELME (5.5.0),MATPLOTLIB (3.7.3), NUMPY (1.24.4), SCIPY (1.10.1), PYTORCH( 1.11.0), PANDAS (1.4.4) packages. The IDE software VSCODE (1.107.0) was utilized for coding. code weblink: https://huggingface.co/datasets/luoyeguigenno1/X-ray-Miniature-Medical-Device |

For manuscripts utilizing custom algorithms or software that are central to the research but not yet described in published literature, software must be made available to editors and reviewers. We strongly encourage code deposition in a community repository (e.g. GitHub). See the Nature Portfolio guidelines for submitting code & software for further information.

## Data

Policy information about availability of data

All manuscripts must include a data availability statement. This statement should provide the following information, where applicable:
- Accession codes, unique identifiers, or web links for publicly available datasets
- A description of any restrictions on data availability
- For clinical datasets or third party data, please ensure that the statement adheres to our policy

The datasets used in this study—including the synthetic MMD data, real MMD data, real MMD locomotion data, and other data referenced in the Methods section—are available at https://huggingface.co/datasets/luoyeguigenno1/X-ray-Miniature-Medical-Device. The repository also contains the trained model weights for the diffusion and instance segmentation models.

## Research involving human participants, their data, or biological material

Policy information about studies with human participants or human data. See also policy information about sex, gender (identity/presentation), and sexual orientation and race, ethnicity and racism.

| | |
|---|---|
| Reporting on sex and gender | This study did not involve human participants; therefore, sex and gender reporting are not applicable. |
| Reporting on race, ethnicity, or other socially relevant groupings | No data on race, ethnicity, or other socially relevant groupings were collected in this study. |
| Population characteristics | No human participants were involved in this study; therefore, population characteristics are not applicable. |
| Recruitment | This study did not involve human participants; therefore, recruitment is not applicable. |
| Ethics oversight | This study did not involve human participants; therefore, ethics oversight is not applicable. |

Note that full information on the approval of the study protocol must also be provided in the manuscript.

# Field-specific reporting

Please select the one below that is the best fit for your research. If you are not sure, read the appropriate sections before making your selection.

☒ Life sciences   ☐ Behavioural & social sciences   ☐ Ecological, evolutionary & environmental sciences

For a reference copy of the document with all sections, see nature.com/documents/nr-reporting-summary-flat.pdf

# Life sciences study design

All studies must disclose on these points even when the disclosure is negative.

| | |
|---|---|
| Sample size | No formal sample size calculation was performed. The study used two 4-month-old New Zealand rabbits and two 4-month-old Sprague Dawley rats, chosen as representative healthy animals to evaluate robot performance. Given that the experiments focused on technical feasibility rather than statistical inference, this sample size was considered sufficient to assess the system's operation and functionality. |
| Data exclusions | No data was excluded |
| Replication | The robot experiments can be replicated inside the arteries of rabbits and mice. The replication was verified in 2 ways: 2 robots deployed to the same animal or the same robot was deployed to 2 animals. |
| Randomization | Sample allocation was not randomized, as all experiments were conducted to assess robot performance rather than to compare treatment groups. Each animal served solely as a test subject, and covariates such as age, species, and health status were controlled by selecting healthy, 4-month-old animals, making randomization unnecessary. |
| Blinding | Blinding was not possible because the in vivo experiments were conducted solely to evaluate robot performance. Theoretically, robot performance is independent of individual animals. In healthy animals, blood vessel morphology, blood flow, and physiological features are consistent, making blinding unnecessary for this study. |

# Behavioural & social sciences study design

All studies must disclose on these points even when the disclosure is negative.

| | |
|---|---|
| Study description | *Briefly describe the study type including whether data are quantitative, qualitative, or mixed-methods (e.g. qualitative cross-sectional,* |

| Study description | quantitative experimental, mixed-methods case study). |
|---|---|
| Research sample | State the research sample (e.g. Harvard university undergraduates, villagers in rural India) and provide relevant demographic information (e.g. age, sex) and indicate whether the sample is representative. Provide a rationale for the study sample chosen. For studies involving existing datasets, please describe the dataset and source. |
| Sampling strategy | Describe the sampling procedure (e.g. random, snowball, stratified, convenience). Describe the statistical methods that were used to predetermine sample size OR if no sample-size calculation was performed, describe how sample sizes were chosen and provide a rationale for why these sample sizes are sufficient. For qualitative data, please indicate whether data saturation was considered, and what criteria were used to decide that no further sampling was needed. |
| Data collection | Provide details about the data collection procedure, including the instruments or devices used to record the data (e.g. pen and paper, computer, eye tracker, video or audio equipment) whether anyone was present besides the participant(s) and the researcher, and whether the researcher was blind to experimental condition and/or the study hypothesis during data collection. |
| Timing | Indicate the start and stop dates of data collection. If there is a gap between collection periods, state the dates for each sample cohort. |
| Data exclusions | If no data were excluded from the analyses, state so OR if data were excluded, provide the exact number of exclusions and the rationale behind them, indicating whether exclusion criteria were pre-established. |
| Non-participation | State how many participants dropped out/declined participation and the reason(s) given OR provide response rate OR state that no participants dropped out/declined participation. |
| Randomization | If participants were not allocated into experimental groups, state so OR describe how participants were allocated to groups, and if allocation was not random, describe how covariates were controlled. |

# Ecological, evolutionary & environmental sciences study design

All studies must disclose on these points even when the disclosure is negative.

| Study description | Briefly describe the study. For quantitative data include treatment factors and interactions, design structure (e.g. factorial, nested, hierarchical), nature and number of experimental units and replicates. |
|---|---|
| Research sample | Describe the research sample (e.g. a group of tagged Passer domesticus, all Stenocereus thurberi within Organ Pipe Cactus National Monument), and provide a rationale for the sample choice. When relevant, describe the organism taxa, source, sex, age range and any manipulations. State what population the sample is meant to represent when applicable. For studies involving existing datasets, describe the data and its source. |
| Sampling strategy | Note the sampling procedure. Describe the statistical methods that were used to predetermine sample size OR if no sample-size calculation was performed, describe how sample sizes were chosen and provide a rationale for why these sample sizes are sufficient. |
| Data collection | Describe the data collection procedure, including who recorded the data and how. |
| Timing and spatial scale | Indicate the start and stop dates of data collection, noting the frequency and periodicity of sampling and providing a rationale for these choices. If there is a gap between collection periods, state the dates for each sample cohort. Specify the spatial scale from which the data are taken |
| Data exclusions | If no data were excluded from the analyses, state so OR if data were excluded, describe the exclusions and the rationale behind them, indicating whether exclusion criteria were pre-established. |
| Reproducibility | Describe the measures taken to verify the reproducibility of experimental findings. For each experiment, note whether any attempts to repeat the experiment failed OR state that all attempts to repeat the experiment were successful. |
| Randomization | Describe how samples/organisms/participants were allocated into groups. If allocation was not random, describe how covariates were controlled. If this is not relevant to your study, explain why. |
| Blinding | Describe the extent of blinding used during data acquisition and analysis. If blinding was not possible, describe why OR explain why blinding was not relevant to your study. |

Did the study involve field work?  ☐ Yes   ☐ No

# Field work, collection and transport

| Field conditions | Describe the study conditions for field work, providing relevant parameters (e.g. temperature, rainfall). |
|---|---|
| Location | State the location of the sampling or experiment, providing relevant parameters (e.g. latitude and longitude, elevation, water depth). |
| Access & import/export | Describe the efforts you have made to access habitats and to collect and import/export your samples in a responsible manner and in |

| Access & import/export | compliance with local, national and international laws, noting any permits that were obtained (give the name of the issuing authority, the date of issue, and any identifying information). |
| Disturbance | Describe any disturbance caused by the study and how it was minimized. |

# Reporting for specific materials, systems and methods

We require information from authors about some types of materials, experimental systems and methods used in many studies. Here, indicate whether each material, system or method listed is relevant to your study. If you are not sure if a list item applies to your research, read the appropriate section before selecting a response.

## Materials & experimental systems

| n/a | Involved in the study |
|---|---|
| ☒ | Antibodies |
| ☒ | Eukaryotic cell lines |
| ☒ | Palaeontology and archaeology |
| ☐ | ☒ Animals and other organisms |
| ☒ | Clinical data |
| ☒ | Dual use research of concern |
| ☒ | Plants |

## Methods

| n/a | Involved in the study |
|---|---|
| ☒ | ChIP-seq |
| ☒ | Flow cytometry |
| ☒ | MRI-based neuroimaging |

## Antibodies

| Antibodies used | Describe all antibodies used in the study; as applicable, provide supplier name, catalog number, clone name, and lot number. |
| Validation | Describe the validation of each primary antibody for the species and application, noting any validation statements on the manufacturer's website, relevant citations, antibody profiles in online databases, or data provided in the manuscript. |

## Eukaryotic cell lines

Policy information about cell lines and Sex and Gender in Research

| Cell line source(s) | State the source of each cell line used and the sex of all primary cell lines and cells derived from human participants or vertebrate models. |
| Authentication | Describe the authentication procedures for each cell line used OR declare that none of the cell lines used were authenticated. |
| Mycoplasma contamination | Confirm that all cell lines tested negative for mycoplasma contamination OR describe the results of the testing for mycoplasma contamination OR declare that the cell lines were not tested for mycoplasma contamination. |
| Commonly misidentified lines (See ICLAC register) | Name any commonly misidentified cell lines used in the study and provide a rationale for their use. |

## Palaeontology and Archaeology

| Specimen provenance | Provide provenance information for specimens and describe permits that were obtained for the work (including the name of the issuing authority, the date of issue, and any identifying information). Permits should encompass collection and, where applicable, export. |
| Specimen deposition | Indicate where the specimens have been deposited to permit free access by other researchers. |
| Dating methods | If new dates are provided, describe how they were obtained (e.g. collection, storage, sample pretreatment and measurement), where they were obtained (i.e. lab name), the calibration program and the protocol for quality assurance OR state that no new dates are provided. |

☐ Tick this box to confirm that the raw and calibrated dates are available in the paper or in Supplementary Information.

| Ethics oversight | Identify the organization(s) that approved or provided guidance on the study protocol, OR state that no ethical approval or guidance was required and explain why not. |

Note that full information on the approval of the study protocol must also be provided in the manuscript.

# Animals and other research organisms

Policy information about [studies involving animals](); [ARRIVE guidelines]() recommended for reporting animal research, and [Sex and Gender in Research]()

| | |
|---|---|
| Laboratory animals | New Zealand White rabbits (Oryctolagus cuniculus), outbred albino stock (genetic background: outbred), aged 4 months and weighing 2.5–3.0 kg at the time of experimentation, were obtained from Guangzhou Xindongxinhua Experimental Animal Breeding Farm (Guangzhou, China). Sprague Dawley rats (Rattus norvegicus), outbred stock (genetic background: outbred), aged 4 months and weighing approximately 700 g at the time of experimentation, were obtained from Zhuhai Bestone Biotechnology Co., Ltd. (Zhuhai, China). Both species are standard, commercially available outbred laboratory animals with no genetic modifications. |
| Wild animals | The study did not involve wild animals. |
| Reporting on sex | sex is not considered in the experiment since the arteries were used, which is independent of sex. |
| Field-collected samples | The study did not involve field-collected samples |
| Ethics oversight | The animal study was approved by the Committee on Institutional Animal Care and Use Committee of Hong Kong Huateng Biotechnology Co., Ltd. (IACUC No. B202502-25) and the Institutional Animal Research Ethics Sub-Committee of City University of Hong Kong (AN-STA-00001025). |

Note that full information on the approval of the study protocol must also be provided in the manuscript.

# Clinical data

Policy information about [clinical studies]()
All manuscripts should comply with the ICMJE [guidelines for publication of clinical research]() and a completed [CONSORT checklist]() must be included with all submissions.

| | |
|---|---|
| Clinical trial registration | *Provide the trial registration number from ClinicalTrials.gov or an equivalent agency.* |
| Study protocol | *Note where the full trial protocol can be accessed OR if not available, explain why.* |
| Data collection | *Describe the settings and locales of data collection, noting the time periods of recruitment and data collection.* |
| Outcomes | *Describe how you pre-defined primary and secondary outcome measures and how you assessed these measures.* |

# Dual use research of concern

Policy information about [dual use research of concern]()

## Hazards

Could the accidental, deliberate or reckless misuse of agents or technologies generated in the work, or the application of information presented in the manuscript, pose a threat to:

No | Yes
- [ ] | [ ] Public health
- [ ] | [ ] National security
- [ ] | [ ] Crops and/or livestock
- [ ] | [ ] Ecosystems
- [ ] | [ ] Any other significant area

## Experiments of concern

Does the work involve any of these experiments of concern:

No | Yes
| | |
| ☐ | ☐ | Demonstrate how to render a vaccine ineffective |
| ☐ | ☐ | Confer resistance to therapeutically useful antibiotics or antiviral agents |
| ☐ | ☐ | Enhance the virulence of a pathogen or render a nonpathogen virulent |
| ☐ | ☐ | Increase transmissibility of a pathogen |
| ☐ | ☐ | Alter the host range of a pathogen |
| ☐ | ☐ | Enable evasion of diagnostic/detection modalities |
| ☐ | ☐ | Enable the weaponization of a biological agent or toxin |
| ☐ | ☐ | Any other potentially harmful combination of experiments and agents |

## Plants

**Seed stocks**

no plant is used

**Novel plant genotypes**

no plant is used

**Authentication**

no plant is used

## ChIP-seq

### Data deposition

☐ Confirm that both raw and final processed data have been deposited in a public database such as GEO.

☐ Confirm that you have deposited or provided access to graph files (e.g. BED files) for the called peaks.

**Data access links**
*May remain private before publication.*

*For "Initial submission" or "Revised version" documents, provide reviewer access links. For your "Final submission" document, provide a link to the deposited data.*

**Files in database submission**

*Provide a list of all files available in the database submission.*

**Genome browser session**
(e.g. UCSC)

*Provide a link to an anonymized genome browser session for "Initial submission" and "Revised version" documents only, to enable peer review. Write "no longer applicable" for "Final submission" documents.*

### Methodology

**Replicates**

*Describe the experimental replicates, specifying number, type and replicate agreement.*

**Sequencing depth**

*Describe the sequencing depth for each experiment, providing the total number of reads, uniquely mapped reads, length of reads and whether they were paired- or single-end.*

**Antibodies**

*Describe the antibodies used for the ChIP-seq experiments; as applicable, provide supplier name, catalog number, clone name, and lot number.*

**Peak calling parameters**

*Specify the command line program and parameters used for read mapping and peak calling, including the ChIP, control and index files used.*

**Data quality**

*Describe the methods used to ensure data quality in full detail, including how many peaks are at FDR 5% and above 5-fold enrichment.*

**Software**

*Describe the software used to collect and analyze the ChIP-seq data. For custom code that has been deposited into a community repository, provide accession details.*

# Flow Cytometry

## Plots

Confirm that:

- [ ] The axis labels state the marker and fluorochrome used (e.g. CD4-FITC).
- [ ] The axis scales are clearly visible. Include numbers along axes only for bottom left plot of group (a 'group' is an analysis of identical markers).
- [ ] All plots are contour plots with outliers or pseudocolor plots.
- [ ] A numerical value for number of cells or percentage (with statistics) is provided.

## Methodology

| | |
|---|---|
| Sample preparation | *Describe the sample preparation, detailing the biological source of the cells and any tissue processing steps used.* |
| Instrument | *Identify the instrument used for data collection, specifying make and model number.* |
| Software | *Describe the software used to collect and analyze the flow cytometry data. For custom code that has been deposited into a community repository, provide accession details.* |
| Cell population abundance | *Describe the abundance of the relevant cell populations within post-sort fractions, providing details on the purity of the samples and how it was determined.* |
| Gating strategy | *Describe the gating strategy used for all relevant experiments, specifying the preliminary FSC/SSC gates of the starting cell population, indicating where boundaries between "positive" and "negative" staining cell populations are defined.* |

- [ ] Tick this box to confirm that a figure exemplifying the gating strategy is provided in the Supplementary Information.

# Magnetic resonance imaging

## Experimental design

| | |
|---|---|
| Design type | *Indicate task or resting state; event-related or block design.* |
| Design specifications | *Specify the number of blocks, trials or experimental units per session and/or subject, and specify the length of each trial or block (if trials are blocked) and interval between trials.* |
| Behavioral performance measures | *State number and/or type of variables recorded (e.g. correct button press, response time) and what statistics were used to establish that the subjects were performing the task as expected (e.g. mean, range, and/or standard deviation across subjects).* |

## Acquisition

| | |
|---|---|
| Imaging type(s) | *Specify: functional, structural, diffusion, perfusion.* |
| Field strength | *Specify in Tesla* |
| Sequence & imaging parameters | *Specify the pulse sequence type (gradient echo, spin echo, etc.), imaging type (EPI, spiral, etc.), field of view, matrix size, slice thickness, orientation and TE/TR/flip angle.* |
| Area of acquisition | *State whether a whole brain scan was used OR define the area of acquisition, describing how the region was determined.* |

Diffusion MRI [ ] Used [ ] Not used

## Preprocessing

| | |
|---|---|
| Preprocessing software | *Provide detail on software version and revision number and on specific parameters (model/functions, brain extraction, segmentation, smoothing kernel size, etc.).* |
| Normalization | *If data were normalized/standardized, describe the approach(es): specify linear or non-linear and define image types used for transformation OR indicate that data were not normalized and explain rationale for lack of normalization.* |
| Normalization template | *Describe the template used for normalization/transformation, specifying subject space or group standardized space (e.g. original Talairach, MNI305, ICBM152) OR indicate that the data were not normalized.* |
| Noise and artifact removal | *Describe your procedure(s) for artifact and structured noise removal, specifying motion parameters, tissue signals and physiological signals (heart rate, respiration).* |

| Volume censoring | *Define your software and/or method and criteria for volume censoring, and state the extent of such censoring.* |

## Statistical modeling & inference

| Model type and settings | *Specify type (mass univariate, multivariate, RSA, predictive, etc.) and describe essential details of the model at the first and second levels (e.g. fixed, random or mixed effects; drift or auto-correlation).* |

| Effect(s) tested | *Define precise effect in terms of the task or stimulus conditions instead of psychological concepts and indicate whether ANOVA or factorial designs were used.* |

Specify type of analysis: ☐ Whole brain ☐ ROI-based ☐ Both

| Statistic type for inference | *Specify voxel-wise or cluster-wise and report all relevant parameters for cluster-wise methods.* |

(See Eklund et al. 2016)

| Correction | *Describe the type of correction and how it is obtained for multiple comparisons (e.g. FWE, FDR, permutation or Monte Carlo).* |

## Models & analysis

| n/a | Involved in the study |
|---|---|
| ☐ | ☐ Functional and/or effective connectivity |
| ☐ | ☐ Graph analysis |
| ☐ | ☐ Multivariate modeling or predictive analysis |

| Functional and/or effective connectivity | *Report the measures of dependence used and the model details (e.g. Pearson correlation, partial correlation, mutual information).* |

| Graph analysis | *Report the dependent variable and connectivity measure, specifying weighted graph or binarized graph, subject- or group-level, and the global and/or node summaries used (e.g. clustering coefficient, efficiency, etc.).* |

| Multivariate modeling and predictive analysis | *Specify independent variables, features extraction and dimension reduction, model, training and evaluation metrics.* |

