## [Peer Review file · Nature Machine Intelligence]

Synthetic X-Ray Driven Tracking and Control of Miniature Medical Devices

Corresponding Author: Professor Metin Sitti

Version 0:

Reviewer comments:

Reviewer #1

(Remarks to the Author)

This paper introduces MicroSyn-X, a framework for training computer vision models to detect and segment miniature medical devices (MMDs) under X-ray imaging. The authors address the critical challenge of data scarcity in medical imaging by developing an end-to-end pipeline that synthesizes high-fidelity, auto-labeled X-ray images. The framework combines a diffusion model for realistic background generation with a programmatic overlay of MMDs, incorporating domain randomization to ensure model robustness. The work is well-executed, the results are compelling, and the contribution of an open-source dataset is significant for the field.

The paper addresses a critical bottleneck in the clinical translation of miniature medical devices and, in my opinion, meets the standards of Nature Machine Intelligence. However, for the sake of clarity and to further strengthen the paper's claims, a few points should be addressed before publication.

First, the central premise of the paper is the elimination of laborious manual annotation. The paper effectively explains how masks for the synthetic data are programmatically generated, which clearly supports this premise. However, the role and creation of the text prompts that guide the diffusion model should be detailed more explicitly. The generation of synthetic backgrounds relies on these prompts to define scene content, and this process could be interpreted as a form of manual input. The manuscript does not specify the number of different prompts used or the method by which they were created and programmatically varied. For instance, were hundreds of unique prompts manually typed, or was a small set of terms programmatically combined and randomized to create diversity? As this can be considered a manual step in the workflow, a brief discussion on the scale and effort of this "prompt engineering" phase, and how it compares to the labor of traditional annotation, would be valuable. This clarification would further solidify the paper's primary claim of significantly improved efficiency.

Second, the framework's use of a diffusion model to generate a diverse set of synthetic tissue backgrounds is a key component of its success. However, the manuscript does not specify whether there is a quality control or filtering step applied to the images after they are generated. This is an important detail because if a manual, or even a complex programmatic, filtering process is required to discard unrealistic or low-quality outputs, it would constitute another form of human-in-the-loop intervention or an additional layer of algorithmic complexity not fully described. It would strengthen the paper to clarify whether all generated backgrounds (without any filtering) were used to create the final synthetic dataset, or if a selection process was employed. If there was a filtering step, the authors should describe the criteria used—for example, manual inspection, a quantitative metric or another automated method. Providing this information would offer a more complete picture of the data generation pipeline's autonomy and the true level of automation achieved.

Third, the manuscript details a patch-based training strategy, where large images are subdivided to focus the model's attention on small regions of interest. This is a sound and well-justified approach for training. However, the manuscript is less explicit about how detection is performed during real-time deployment. It would be beneficial for the authors to clarify whether the deployed model processes the entire video frame, or if a similar patch-based subdivision is used during inference. This clarification is important for fully understanding the real-time performance metrics.

(Remarks on code availability)

Reviewer #2

(Remarks to the Author)

The authors propose a framework that integrates synthetic X-ray data generation and vision-based robotic control of miniature magnetic devices. The overall idea is novel and technically valuable for improving localization in low-contrast clinical environments. However, several important aspects should be clarified or expanded to make this work more convincing from a robotic system perspective.

1. The magnetic manipulation system is well described, but the paper lacks information about its effective workspace. Even a simple quantitative range of field strength or torque near the robot would help define the physical boundary where tracking and actuation remain reliable.
2. The vision-based tracking is claimed to operate in real time. The authors should report the latency between X-ray acquisition, model inference, and actuation command. Quantifying this delay is essential to verify closed-loop stability during dynamic locomotion.
3. Robot locomotion results are shown only qualitatively. The authors should include metrics such as locomotion distance, actuation frequency, or success rate under occlusion to demonstrate control repeatability.
4. The domain randomization effectively expands dataset diversity, but the generated shapes of liquid robots appear purely geometric. Discussion on incorporating physics-based deformation models or real motion datasets would improve realism and generalization.
5. It would be essential to discuss how this approach compares with other guidance modalities such as ultrasound or electromagnetic tracking, especially regarding localization accuracy and clinical safety.

Minor comments

- It would be helpful to include approximate actuation force or torque values in Figures 5–7 to give readers a clearer sense of the physical scale of the robot's motion.
- Please clarify whether the parylene-C coating maintains its mechanical and chemical stability after repeated magnetic actuation cycles. This information would support the long-term applicability of the system.
- Providing basic metadata for the open-source dataset, such as robot diameter, material composition, and magnetization, would make the resource more accessible and useful for other researchers.
- It may also be worth briefly mentioning transformer-based or temporal models as potential future directions for improving tracking performance in continuous video sequences.

(Remarks on code availability)

Reviewer #3

(Remarks to the Author)

This paper presented a deep learning framework for training and guiding miniature medical devices. The authors introduced the first X-ray MMD dataset for medical robotic system, which is quite impressive. However, in its current form, the work appears incremental, primarily applying machine learning to a new dataset. If the main contribution is the model or framework, the manuscript lacks essential evaluation, such as cross-validation and comparisons with strong baselines, to support its claims. While the work holds significant potential and is of high interest, addressing the comments below will enhance the rigor of the claims:

1. Since the framework is used for real-time clinical guidance, how fast could deep learning framework react to the real-time image? When the image quality suddenly become bad, any alternative solutions to prevent errors?
2. The authors used synthesized and auto-labeled X-ray images for model training. However, if the training dataset can be auto-labeled, why not use it for testing directly?
3. The authors applied stable diffusion for tissue generation. However, stable diffusion is known to suffer structural hallucinations, such as detail duplication/missing, and generate samples that lie outside the training distribution. How does the author prevent these hallucinations?
4. Generative models may fail to accurately reproduce pathological findings (such as tumors, fractures, or inflammations), and may synthesize non-existent pseudo-features or simply blurred details. This can mislead downstream diagnostic models learning incorrect guidance. Have the authors compared the framework on true dataset versus synthesized dataset?
5. In Fig4, when evaluating the system robustness, the authors collected eight data indexes for comparing model versus human experts. What are these data indexes stand for? Are the testing data also synthetic or real? For robustness test, it is suggested to evaluate on tissue X-rays on different human subjects. Are there significant subject variations when it comes to unseen new subjects for the model?
6. After the model training, what is the objective for navigation and guiding? When there are multiple paths to guide the robot, which path will the framework choose? It is recommended to add these decision rule details in the figure and discussions.

(Remarks on code availability)

I reviewed the code briefly but didn't try running it. It seems clear and solid.

Version 1:

Reviewer comments:

Reviewer #1

(Remarks to the Author)

The authors have addressed most of my previous concerns. I have only one minor point that should be clarified before publication:

The manuscript now explains that programmatic prompt generation involves (i) a one-time curation step and (ii) subsequent programmatic generation. For the one-time curation, the authors note that 10–20 terms were initially defined. To ensure transparency and reproducibility, the complete list of curated terms should be provided. Are these terms already included in the code repository? If not, they should be added to the supplementary materials.

Aside from this point, I have no further comments.

(Remarks on code availability)

Reviewer #2

(Remarks to the Author)

The authors have adequately addressed the questions raised in the previous round of review. This work is expected to provide an important foundation for future medical applications utilizing magnetic robots.

(Remarks on code availability)

Reviewer #3

(Remarks to the Author)

The authors have addressed all my comments. It is therefore recommended for publication as is.

(Remarks on code availability)

Response to Reviewers

We want to thank the editor and reviewers for their insightful comments, which allowed us to improve the quality of the current manuscript. We have addressed all these comments point-by-point in the following response. The comments we have received are colored blue, the responses are placed below, and the revisions to the original files are highlighted in yellow.

Editorial comments:

- Punctuation is not allowed in titles.

We have removed the punctuation with the new title “Synthetic X-Ray Driven Tracking and Control of Miniature Medical Devices” X-ray is a word itself.

- The paper's abstract (200 words maximum; without references) should serve both as a general introduction to the topic, and as a brief, non-technical summary of your main results and their implications. It should start by outlining the background to your work (why the topic is important) and the main question you have addressed (the specific problem that initiated your research), before going on to describe your new observations, main conclusions and their general implications, introduced by the phrase "Here we show" or equivalent. Because we hope that a broad range of researchers will be interested in your work, the abstract should be as accessible as possible, explaining essential but specialised terms concisely. We have supplemented more introductory contents in the abstract

“The clinical translation of miniature medical devices (MMDs) for minimally invasive surgery promises transformative advances in biomedical engineering, offering enhanced precision, reduced patient trauma, and faster recovery times. However, their effective deployment in complex anatomies under real-time X-ray guidance—a widely used surgical imaging modality—presents challenges like low imaging quality and difficulties of spatial robot control.”

Word counts of the abstract: 196.

- We prefer to avoid the use of terms like "new" and "novel" or “first”, as in practice it is difficult to truly ascribe something as completely new. As such, it can detract from the achievements of the work by generating discussions about its "newness" instead of its unique scientific aspects. Please remove such words from the Abstract and text.

We all removed all relevant terms.

- The length of the main article (without abstract or methods section) should be below 4000 words.

The main article has been reduced to 3997 words.

- The maximum number of display items (figures and tables) is 6. Please move additional items to Extended Data (see below for more information on this section).

We have reduced the number of main figures to 6.

- We recommend that authors remove cartoons and icons from figures, e.g. Fig 1.

We appreciate the suggestion to remove cartoons and icons from the figures. The schematic elements in Figure 1 are important for helping readers grasp the framework quickly and intuitively, thereby improving the readability of the manuscript. To address the concern, we have replaced the illustrative icon in the image synthesis framework with a more technical representation.

Additionally, all icons and cartoon elements in Figure 5 have been removed.

All papers in Nature Machine Intelligence include a detailed Data availability statement and Code availability statement. Please see examples from papers published in our journal. The data and code should be available to reviewers during peer review.

- Data availability: see our policy here <https://www.nature.com/natmachintell/editorial->

policies/reporting-standards#availability-of-data

In particular, please clearly list **all datasets that have been used or generated in this work**, providing links and references.

- Code should be available in a public repository and include a link to the References. Please also generate a DOI (see <https://guides.github.com/activities/citable-code/>) which makes your code more easily citable and discoverable by others and which makes the article more reproducible as a specific version of your code is linked to the paper. PLEASE ADD THIS DOI TO THE REFERENCE LIST AND CITE THE REFERENCE NUMBER IN THE CODE AVAILABILITY STATEMENT.

To reduce uncertainty for people who might want to use your code, it would be helpful if you could add a License to your repository that either permits any kind of re-use and usage (e.g. the MIT license), or the conditions for re-distribution (e.g. GPLv3). GitHub makes it easy to find and apply a license to a repository, see here: <https://help.github.com/en/github/building-a-strong-community/adding-a-license-to-a-repository>

Example of GitHub doi and ref in a paper published in Nature Machine Intelligence:

<https://www.nature.com/articles/s42256-022-00531-2#code-availability>

Thank you for the feedback. We have now:

- Added a **Data Availability** section and included the corresponding access links.
- Added a **Code Availability** section, created the DOI, and cited it accordingly.
- Confirmed that the **CC BY-NC 4.0 license** has been applied.

Please let us know if any further adjustments are required.

Reviewer #1 (Remarks to the Author):

This paper introduces MicroSyn-X, a framework for training computer vision models to detect and segment miniature medical devices (MMDs) under X-ray imaging. The authors address the critical challenge of data scarcity in medical imaging by developing an end-to-end pipeline that synthesizes high-fidelity, auto-labeled X-ray images. The framework combines a diffusion model for realistic background generation with a programmatic overlay of MMDs, incorporating domain randomization to ensure model robustness. The work is well-executed, the results are compelling, and the contribution of an open-source dataset is significant for the field.

The paper addresses a critical bottleneck in the clinical translation of miniature medical devices and, in my opinion, meets the standards of Nature Machine Intelligence. However,

for the sake of clarity and to further strengthen the paper's claims, a few points should be addressed before publication.

Comment 1: First, the central premise of the paper is the elimination of laborious manual annotation. The paper effectively explains how masks for the synthetic data are programmatically generated, which clearly supports this premise. However, the role and creation of the text prompts that guide the diffusion model should be detailed more explicitly. The generation of synthetic backgrounds relies on these prompts to define scene content, and this process could be interpreted as a form of manual input. The manuscript does not specify the number of different prompts used or the method by which they were created and programmatically varied. For instance, were hundreds of unique prompts manually typed, or was a small set of terms programmatically combined and randomized to create diversity? As this can be considered a manual step in the workflow, a brief discussion on the scale and effort of this "prompt engineering" phase, and how it compares to the labor of traditional annotation, would be valuable. This clarification would further solidify the paper's primary claim of significantly improved efficiency.

Response:

We thank the reviewer for this astute observation. We have now revised the main text to clarify that this step is highly automated (**programmatically combined and randomized**) and requires minimal, one-time effort, preserving the core efficiency advantage over manual annotation.

Our Method: Programmatic Prompt Generation

Our approach was based on programmatic combination and randomization, not manual per-prompt creation. The process is a one-time setup:

- **One-time curation**: We defined a small, fixed vocabulary of anatomical terms for each tissue category (e.g., "brain," "skull," "lumen," "vessel").
- **Programmatic generation**: During synthetic data creation, prompts are assembled automatically by randomly sampling and combining these predefined terms (e.g., "porcine brain within the skull," "lumen inside heart "). This does not require manual intervention for individual images.

Efficiency Comparison

This method is fundamentally more efficient than piece-wise manual work:

- **Our method**: A one-time setup of a small vocabulary (e.g., 10-20 terms) can programmatically generate a large number of unique prompts and images. The human effort is constant and minimal, regardless of dataset size.
- **Manual annotation**: Manual annotation or per-prompt typing requires labor that scales linearly with the dataset (e.g., 10,000 images require 10,000 manual actions).

Future Enhancement

We agree that prompt design is an important area. As a future direction, one could leverage large language models (LLMs) to generate even more nuanced and varied prompts dynamically, further automating this step [1].

The relevant contents have been added to the section “Diffusion model training and inference” in Methods.

References:

[1] Bozkurt, A. & Sharma, R. C. Generative AI and prompt engineering: The art of whispering to let the genie out of the algorithmic world. *Asian Journal of Distance Education* 18, i-vii (2023).

Comment 2: Second, the framework's use of a diffusion model to generate a diverse set of synthetic tissue backgrounds is a key component of its success. However, the manuscript does not specify whether there is a quality control or filtering step applied to the images after they are generated. This is an important detail because if a manual, or even a complex programmatic, filtering process is required to discard unrealistic or low-quality outputs, it would constitute another form of human-in-the-loop intervention or an additional layer of algorithmic complexity not fully described. It would strengthen the paper to clarify whether all generated backgrounds (without any filtering) were used to create the final synthetic dataset, or if a selection process was employed. If there was a filtering step, the authors should describe the criteria used—for example, manual inspection, a quantitative metric or another automated method. Providing this information would offer a more complete picture of the data generation pipeline's autonomy and the true level of automation achieved.

Thank you for raising this important point.

- We clarify that our framework does **not employ a post-hoc filtering** step on the generated backgrounds.
- We implement a proactive, **two-phase quality control** strategy during the diffusion model training phase (select the model with good generation performance) and model inference phase (tuning discussion steps and prompt guidance weight).
- We also systematically investigate the **influence of background quality on downstream model** performance and determine the conditions under which they are harmful or beneficial as a form of data augmentation.
- We prioritize the utilization of large downstream models due to its robustness to noise.

Our approach is detailed below and summarized in the provided supplementary figures.

1. Background quality control

1.1. During Model Training:

We periodically saved candidate models during training and evaluated them using a multi-faceted protocol:

- quantitative fidelity (structural similarity index measure): We quantified structural similarity to real tissue images (**Fig. 3a, b**).
- feature distribution: we ensured generated images expanded upon the feature distribution of real data without diverging from it (**Fig. 3c**).
- Qualitative and human-centric assessment: an operator selected the final model based on the generation of realistic, randomized textures with appropriate illumination and contrast, avoiding surreal artifacts.

1. **Fig. 3. Domain randomization of synthetic tissue images and open-sourced MMD X-ray dataset.** **a.** Real tissue images under X-ray imaging. **b.** Generation of precisely matched synthetic tissues with randomized and enhanced textures. SSIM represents the structural similarity index measure. **c.** Image generation with mask-guided conditioning and prompts. Randomization of masks and prompts significantly expands the dataset. **d.** Domain comparison of real and synthetic tissues. Features from 1,140 real and 24,803 synthetic tissue images, extracted using Inception V3, are visualized via principle component analysis (PCA). PC represents principal component.

1.2 During Model Inference:

We select generation parameters to minimize artifacts. The low-quality generation cases are categorized as:

- Artifacts: surreal textures, blurriness, high-frequency noise, grid-like or checkerboard patterns, and overly noisy images.

- Inconsistent physics: inconsistent illumination/exposure, incorrect contrast,
- Lack of realistic textures: smoothness in areas that should have texture.

These artifacts are illustrated in **Supplementary Fig. 14 a**.

Supplementary Fig. 14. Quality assessment and parameter sensitivity in diffusion model inference. a. Characteristic artifacts in low-quality generations, categorized as: generation artifacts (surreal textures, blurriness, high-frequency noise, grid/checkerboard patterns), physical inconsistencies (non-uniform illumination, exposure mismatches, implausible contrast), and texture deficiencies (excessive smoothness in structurally detailed regions).

We tuned the number of diffusion steps and the guidance scale (ρ) to find a stable to minimize the occurrence of low-quality generation, as indicated by the green markers in **Supplementary Fig. 14 b**.

Supplementary Figure 14. Quality assessment and parameter sensitivity in diffusion model inference. b. Trade-offs between number of diffusion steps and classifier-free guidance scale: low diffusion step counts combined with high prompt guidance weights are prone to produce unstable or distorted outputs. Parameters selected for final inference are indicated by green markers.

These contents have been added to the “Diffusion model training and inference” section in Methods.

2. Investigation of downstream model robustness to background quality

We created five synthetic datasets by blending 200 high-quality and 200 low-quality backgrounds in varying ratios (1.0, 0.86, 0.5, 0.15, 0.0). We then trained 20 instance segmentation models of different model sizes (2.8M, 10.1M, 22.4M, 27.6M) on these datasets and evaluated them on a held-out test set of real X-ray images (**Extended Data Fig. 3**).

Extended Data Fig. 3. Effect of synthetic background quality on downstream segmentation models. a. Dataset composition. 200 high- and 200 low-quality generated tissue backgrounds were used to construct five synthetic training sets with varying proportions of high-quality backgrounds. **b.** Real-tissue test dataset. The test images include soft MMDs in porcine brain with embedded bone, porcine brain, heart, liver, stomach, heart 3D vessels, in vivo rabbit, in vivo rat, and in vivo rat spine at low magnification. Scale bars represent 5 mm.

Key Findings:

- For MMDs with high visibility and distinct features (e.g., Stents): Model performance is consistent or better across all background mixtures. In this case, low-quality backgrounds act as a beneficial data augmentation, improving robustness.
- For low-contrast or ambiguous targets (e.g., blurred white rectangles resembling background artifacts), the effect is model-capacity dependent:

- Large Models (22.4M, 27.6M): A modest fraction (0.14) of low-quality backgrounds improved or maintained performance, acting as a regularizer that forces the model to learn more robust features, while the performance degrades with a majority of low-quality data.
- Small Models (2.8M, 10.1M): Performance degraded with the inclusion of low-quality backgrounds, as their limited capacity made them susceptible to learning spurious correlations.

Extended Data Fig. 3. Effect of synthetic background quality on downstream segmentation models. c. Segmentation performance across background quality ratios and model sizes. For structurally distinct targets (e.g., stents) with high visibility, performance remains stable or improved, where low-quality backgrounds provide beneficial augmentation. In contrast, for low-contrast or ambiguous targets (e.g., blurred white rectangles resembling background artifacts), inclusion of a modest fraction of low-quality backgrounds (ratio = 0.86) improves or maintains performance for larger models, but degrades performance for smaller models, evidenced by missed detections. The red dashed lines represent the result of model trained on real data. Scale bars represent 1 mm.

Therefore, our framework adopts the following strategies to ensure the downstream model performance. We achieve this through:

- Proactive quality control during diffusion model training and inference.
- For challenging imaging scenes, utilizing large models for their demonstrated robustness to background noise, effectively turning a potential weakness into a form of data augmentation.

We could further a learned filter or classifier to automatically score and select the most beneficial "bad" backgrounds for augmentation, improving the downstream model performance.

The following paragraph has been added to the main text.

“We also investigate the impact of synthetic background quality on downstream CV models (Extended Data Fig. 3). For MMDs with distinct features like stents, performance is largely unaffected by background quality, whereas for MMDs with ambiguous features, the effect is model-dependent: smaller models degrade with low-quality data, while larger models can utilize it as effective regularization. To tackle this issue, we adopt a two-phase quality control strategy during tissue generation (diffusion model selection and artifact minimization) and prioritize the utilization of large downstream models with its robustness to noise (“Diffusion model training and inference” in Methods). A classifier can be developed to automatically select backgrounds for CV model training as a future step.”

Comment 3: Third, the manuscript details a patch-based training strategy, where large images are subdivided to focus the model's attention on small regions of interest. This is a sound and well-justified approach for training. However, the manuscript is less explicit about how detection is performed during real-time deployment. It would be beneficial for the authors to clarify whether the deployed model processes the entire video frame, or if a similar patch-based subdivision is used during inference. This clarification is important for fully understanding the real-time performance metrics.

We thank the reviewer for this critical question. For real-time deployment, we use a dynamic, **dual-mode** approach that switches between processing the **entire frame** and **focused patches** to balance accuracy and speed.

The specific workflow, detailed in Supplementary Figure 2 and Extended Data Fig. 6, is as follows:

- **Global Search Mode (Patch-based)**: Used sparingly for initialization or recovery if the robot is lost. The full video frame is subdivided into overlapping patches for processing.
 - Latency: 333.8 ± 4.9 ms (for 25 patches using a 22.4M parameter model).
- **Local Tracking Mode (Full-frame ROI)**: This is the primary operational mode for real-time tracking. Once the robot is located, the model processes only a high-resolution, cropped Region of Interest (ROI) around the last known position, not the entire frame.
 - Latency: 21.6 ± 1.6 ms per object (from reading raw data to output).
 - Hardware: NVIDIA RTX Titan, Intel Xeon 5220 @2.2GHz, 64GB RAM.

This hybrid strategy is key to our system's real-time performance. The fast local mode maintains a high tracking rate, while the global mode ensures robustness. The switching logic between these modes is shown in **Extended Data Fig. 6** and the section “Computer vision model training and inference” in Methods.

Supplementary Fig. 2. Schematic of the object localization algorithm. a. Schematics of the global mode. The input image is partitioned into overlapping patches, where patch size is adaptively determined based on image scale (pixels/mm). Each patch is processed independently by the model, and the resulting detections are aggregated and merged to produce the final localization output. **b.** Schematics of the local mode. Leveraging prior localization results, the algorithm focuses computation on regions of interest (ROIs), feeding only these refined subregions into the model to enhance the efficiency. In all figures, scale bars represent 5 mm.

Algorithm 1 Object Tracking

```
1: Initialize:  $t \leftarrow 0$  ▷ Frame index
2:  $\mathcal{T} \leftarrow \emptyset$  ▷ Set of active tracks
3:  $\mathcal{K} \leftarrow \emptyset$  ▷ Set of Kalman filters
4:  $\mathcal{C} \leftarrow \emptyset$  ▷ Set of candidate detections with counters
5:  $T_{\text{confirm}} \leftarrow 5$  ▷ Frame threshold for track confirmation
6: procedure PROCESSFRAME( $I_t$ )
7:   if  $t = 0$  then
8:      $\mathcal{D}_t \leftarrow \text{global}(I_t)$  ▷ Initial global detection  $d_i \in \mathcal{D}_t$ 
9:      $\text{ROI}_i \leftarrow \text{createROI}(d_i)$ 
10:     $\mathbf{x}_i^0 \leftarrow [p_x, p_y, v_x, v_y]^\top$  ▷ Initial state
11:     $\mathcal{K}_i \leftarrow \text{KalmanInit}(\mathbf{x}_i^0, \mathbf{P}_0)$ 
12:     $\mathcal{T} \leftarrow \mathcal{T} \cup \{\tau_i\}$ 
13:     $\text{age}(\tau_i) \leftarrow 1$ 
14:   else
15:     Step 1: Prediction for existing tracks  $\tau_i \in \mathcal{T}$ 
16:      $\hat{\mathbf{x}}_i^t \leftarrow \mathcal{K}_i(\cdot)$  ▷ Kalman prediction
17:      $\text{ROI}_i \leftarrow \text{updateROI}(\hat{\mathbf{x}}_i^t)$ 
18:     Step 2: Detection and association for existing tracks  $\tau_i \in \mathcal{T}$ 
19:      $\mathcal{D}_i \leftarrow \text{local}(I_t, \text{ROI}_i)$ 
20:     if  $\mathcal{D}_i = \emptyset$  then
21:        $\mathcal{D}_i \leftarrow \text{global}(I_t)$ 
22:       if  $\mathcal{D}_i = \emptyset$  then
23:          $\text{stopActuation}()$ 
24:          $\text{retractMagnet}()$ 
25:          $\mathcal{D}_i \leftarrow \text{global}(I_t)$  ▷ Re-attempt detection
26:       end if
27:     end if
28:      $d_i^* \leftarrow (\hat{\mathbf{x}}_i^t, \mathcal{D}_i)$ 
29:     if  $d_i^* \neq \emptyset$  then
30:        $\mathcal{K}_i(\mathbf{z}_i^t)$  ▷ Update with measurement
31:        $\text{age}(\tau_i) \leftarrow \text{age}(\tau_i) + 1$ 
32:     end if
33:     Step 3: New candidate detection and confirmation
34:      $\mathcal{D}_t^{\text{new}} \leftarrow \text{global}(I_t) \setminus \{\text{associated detections}\}$   $d_j \in \mathcal{D}_t^{\text{new}}$ 
35:     if  $d_j \in \mathcal{C}$  then ▷ Existing candidate
36:        $c_j \leftarrow c_j + 1$  ▷ Increment counter
37:       if  $c_j \geq T_{\text{confirm}}$  then
38:         Initialize new track  $\tau_j$ 
39:         Initialize new Kalman filter  $\mathcal{K}_j$ 
40:          $\mathcal{T} \leftarrow \mathcal{T} \cup \{\tau_j\}$ 
41:          $\mathcal{C} \leftarrow \mathcal{C} \setminus \{d_j\}$  ▷ Remove from candidates
42:       end if
43:     else
44:        $\mathcal{C} \leftarrow \mathcal{C} \cup \{d_j\}$  ▷ Add new candidate
45:        $c_j \leftarrow 1$  ▷ Initialize counter
46:     end if
47:     Step 4: Cleanup stale candidates  $d_j \in \mathcal{C}$ 
48:     if detection in current frame matching  $d_j$  then
49:        $c_j \leftarrow \max(0, c_j - 1)$  ▷ Decrement counter
50:       if  $c_j = 0$  then
51:          $\mathcal{C} \leftarrow \mathcal{C} \setminus \{d_j\}$  ▷ Remove stale candidate
52:       end if
53:     end if
54:   end if
55:    $t \leftarrow t + 1$ 
56: end procedure
```

Extended Data Fig. 6. Pseudocode of the object tracking algorithm.

Reviewer #2 (Remarks to the Author):

The authors propose a framework that integrates synthetic X-ray data generation and vision-based robotic control of miniature magnetic devices. The overall idea is novel and technically valuable for improving localization in low-contrast clinical environments. However, several important aspects should be clarified or expanded to make this work more convincing from a robotic system perspective.

1. The magnetic manipulation system is well described, but the paper lacks information about its effective workspace. Even a simple quantitative range of field strength or torque near the robot would help define the physical boundary where tracking and actuation remain reliable. We thank the reviewer for this valuable suggestion. We have now supplemented the manuscript with Extended Data Fig. 5 to quantitatively define the effective workspace of our magnetic manipulation system.

The key concept we introduce is the "**actuation region**," which is the 3D spatial region where the permanent magnet (PM) must be positioned to enable MMD movement. This region is defined by specific magnetic field thresholds required for effective movement, which we determined experimentally:

- For soft MMDs inside lumens: actuation requires the magnetic torque and force to exceed a minimum threshold to overcome the resistance.
- For liquid MMDs on surfaces: actuation requires the magnetic field gradient along the desired direction of motion to exceed a threshold, while gradients in other directions remain below this level to prevent undesired movement.

These experimentally-derived thresholds [1] are used to calculate and visualize the 3D action regions in **Extended Data Fig. 5**. This visualization provides a clear, quantitative boundary for the physical workspace, where the actuation parameters are also presented.

References:

[1] Wang, C., Wang, T., Li, M., Zhang, R., Ugurlu, H., & Sitti, M. (2024). Heterogeneous multiple soft millirobots in three-dimensional lumens. *Science Advances*, 10(45), eadq1951.

Extended Data Fig. 5. Actuation regions of soft and liquid MMDs. **a.** Actuation of the soft MMD inside a lumen, driven by magnetic torque and force. **b.** Actuation region of the soft MMD in the plane of $x_r = 0$ mm. When the rotating permanent magnet (PM) is positioned within this region, the magnetic torque and force exceed the thresholds to enable MMD motion. **c.** Three-dimensional actuation region of the soft MMD, considering geometric constraints to prevent potential collisions. **d.** Actuation of the soft MMD on a substrate, where motion is

achieved by utilizing the magnetic field gradient. **e.** Actuation region of the liquid MMD in the plane of $x_r = 0$. When the PM is located within this region, the magnetic field gradient along the desired direction exceeds the threshold to induce MMD movement. **f.** Three-dimensional actuation region of the liquid MMD. A larger PM can be employed to expand the actuation region.

2. The vision-based tracking is claimed to operate in real time. The authors should report the latency between X-ray acquisition, model inference, and actuation command. Quantifying this delay is essential to verify closed-loop stability during dynamic locomotion.

We thank the reviewer for this critical question regarding the real-time performance. We have quantified the latency for each step, which confirms the system's stability for dynamic locomotion (MMD speed <1.5 mm/s). The end-to-end process and its associated timings are as follows:

Image acquisition	Image processing	Actuation Command
High-resolution mode: 66.7 ms	Local mode: 21.6 ± 1.6 ms	From user command to arm execution: <112 ms
Standard-resolution mode: 33.3 ms	Global re-initialization: 333.8 ± 4.9 ms	

- Image acquisition:

X-ray Capture: The C-arm operates at an adaptive frame rate of

- 0-15 fps under the continuous high-resolution imaging mode (66.7 ms/frame minimum interval);
- 0-30 fps under the continuous standard-resolution imaging mode (33.3 ms/frame minimum interval), according to the C-arm machine datasheet (Fluoroscan InSight FD, Hologic GmbH).

- Image processing:

We employ a dual-mode algorithm for object localization, as shown in **Supplementary Fig. 2** and **Extended Data Fig. 6**, including

- Local tracking mode: focused tracking on regions of interest (primary operational mode). 21.6 ± 1.6 ms per object (model size: 22.4M) from reading the raw data to output the results.
- Global re-initialization: comprehensive search across the entire image (used sparingly for initialization). 333.8 ± 4.9 ms (model size: 22.4M, 25 patches).

Hardware: NVIDIA RTX Titan, Intel Xeon 5220 @2.2GHz, 64GB RAM.

This processing speed is fully compatible with the X-ray image capture rate of up to 30 frames per second (fps).

- Actuation Command:

- User-in-the-loop requirement: for clinical safety, the standard workflow requires the operator to approve the movement command. This is a mandated step for clinical deployment.
- System Response: after approval, the latency from command dispatch to the robotic arm executing the command is <112 ms.

The MMD's locomotion speed is below 1.5 mm/s, as shown in the supplementary tables. Each subsequent command is issued after the robot advances approximately 1–2 mm. This latency is acceptable for dynamic locomotion.

The relevant contents have been added to the section “Latency of the teleoperated robotic system” in Methods.

Supplementary Fig. 2. Schematic of the object localization algorithm. a. Schematics of the global mode. The input image is partitioned into overlapping patches, where patch size is adaptively determined based on image scale (pixels/mm). Each patch is processed independently by the model, and the resulting detections are aggregated and merged to produce the final localization output. **b.** Schematics of the local mode. Leveraging prior localization results, the algorithm focuses computation on regions of interest (ROIs), feeding only these refined subregions into the model to enhance the efficiency. In all figures, scale bars represent 5 mm.

Algorithm 1 Object Tracking

```
1: Initialize:  $t \leftarrow 0$  ▷ Frame index
2:  $\mathcal{T} \leftarrow \emptyset$  ▷ Set of active tracks
3:  $\mathcal{K} \leftarrow \emptyset$  ▷ Set of Kalman filters
4:  $\mathcal{C} \leftarrow \emptyset$  ▷ Set of candidate detections with counters
5:  $T_{\text{confirm}} \leftarrow 5$  ▷ Frame threshold for track confirmation
6: procedure PROCESSFRAME( $I_t$ )
7:   if  $t = 0$  then
8:      $\mathcal{D}_t \leftarrow \text{global}(I_t)$  ▷ Initial global detection  $d_i \in \mathcal{D}_t$ 
9:      $\text{ROI}_i \leftarrow \text{createROI}(d_i)$ 
10:     $\mathbf{x}_i^0 \leftarrow [p_x, p_y, v_x, v_y]^\top$  ▷ Initial state
11:     $\mathcal{K}_i \leftarrow \text{KalmanInit}(\mathbf{x}_i^0, \mathbf{P}_0)$ 
12:     $\mathcal{T} \leftarrow \mathcal{T} \cup \{\tau_i\}$ 
13:     $\text{age}(\tau_i) \leftarrow 1$ 
14:   else
15:     Step 1: Prediction for existing tracks  $\tau_i \in \mathcal{T}$ 
16:      $\hat{\mathbf{x}}_i^t \leftarrow \mathcal{K}_i(\cdot)$  ▷ Kalman prediction
17:      $\text{ROI}_i \leftarrow \text{updateROI}(\hat{\mathbf{x}}_i^t)$ 
18:     Step 2: Detection and association for existing tracks  $\tau_i \in \mathcal{T}$ 
19:      $\mathcal{D}_i \leftarrow \text{local}(I_t, \text{ROI}_i)$ 
20:     if  $\mathcal{D}_i = \emptyset$  then
21:        $\mathcal{D}_i \leftarrow \text{global}(I_t)$ 
22:       if  $\mathcal{D}_i = \emptyset$  then
23:          $\text{stopActuation}()$ 
24:          $\text{retractMagnet}()$ 
25:          $\mathcal{D}_i \leftarrow \text{global}(I_t)$  ▷ Re-attempt detection
26:       end if
27:     end if
28:      $d_i^* \leftarrow (\hat{\mathbf{x}}_i^t, \mathcal{D}_i)$ 
29:     if  $d_i^* \neq \emptyset$  then
30:        $\mathcal{K}_i(\mathbf{z}_i^t)$  ▷ Update with measurement
31:        $\text{age}(\tau_i) \leftarrow \text{age}(\tau_i) + 1$ 
32:     end if
33:     Step 3: New candidate detection and confirmation
34:      $\mathcal{D}_t^{\text{new}} \leftarrow \text{global}(I_t) \setminus \{\text{associated detections}\}$   $d_j \in \mathcal{D}_t^{\text{new}}$ 
35:     if  $d_j \in \mathcal{C}$  then ▷ Existing candidate
36:        $c_j \leftarrow c_j + 1$  ▷ Increment counter
37:       if  $c_j \geq T_{\text{confirm}}$  then
38:         Initialize new track  $\tau_j$ 
39:         Initialize new Kalman filter  $\mathcal{K}_j$ 
40:          $\mathcal{T} \leftarrow \mathcal{T} \cup \{\tau_j\}$ 
41:          $\mathcal{C} \leftarrow \mathcal{C} \setminus \{d_j\}$  ▷ Remove from candidates
42:       end if
43:     else
44:        $\mathcal{C} \leftarrow \mathcal{C} \cup \{d_j\}$  ▷ Add new candidate
45:        $c_j \leftarrow 1$  ▷ Initialize counter
46:     end if
47:     Step 4: Cleanup stale candidates  $d_j \in \mathcal{C}$ 
48:     if detection in current frame matching  $d_j$  then
49:        $c_j \leftarrow \max(0, c_j - 1)$  ▷ Decrement counter
50:       if  $c_j = 0$  then
51:          $\mathcal{C} \leftarrow \mathcal{C} \setminus \{d_j\}$  ▷ Remove stale candidate
52:       end if
53:     end if
54:   end if
55:    $t \leftarrow t + 1$ 
56: end procedure
```

Extended Data Fig. 6. Pseudocode of the object tracking algorithm.

3. Robot locomotion results are shown only qualitatively. The authors should include metrics such as locomotion distance, actuation frequency, or success rate under occlusion to demonstrate control repeatability.

We have added two tables to quantitatively present the locomotion results, including tissue type, imaging settings, locomotion distance, mean robot speed (mm/s), magnet rotation rate (Hz), and localization success rate. **The localization success ratio (LSR)**, defined as the ratio of successfully localized frames to the total number of video frames, is used to quantify performance under different levels of occlusion and imaging conditions. These quantitative results are also demonstrated in the supplementary videos.

Tissue	Imaging parameters (kV/ μ A)	Locomotion distance (mm)	Mean robot Speed (mm/s)	Magnet rotation rate (Hz)	Localization success rate
Bone (15-25 mm), porcine liver	61/97	79.12	0.41	0.3-0.7	98.4%
Porcine heart	60/56	68.63	0.42	0.3-0.7	57.4%
	66/97	51.66	0.68	0.9-1.3	23.4%
Porcine liver	53/64	48.85	0.40	0.3-0.7	98.8%
Porcine stomach	51/97	60.03	0.33	0.3-0.7	99.6%
Bone (5-15 mm), porcine brain, skull model	56/97	67.52	0.28	0-0.7	78.4%
	56/97	42.55	0.36	0-0.7	71.9%
	56/97	32.93	0.24	0-0.7	73.7%
	56/97	22.41	0.20	0-0.7	81.9%
Porcine heart artery, skull model	73/97	83.23	0.30	0-0.7	80.0%
Porcine brain, skull model	54/98	31.56	0.21	0.3-0.7	95.1%
	57/97	74.84	0.43	0.3-0.7	98.6%
	57/97	75.95	0.38	0.3-0.7	99.1%
	57/97	53.18	0.29	0.3-0.7	95.5%
Rabbit femoral arteries in vivo	57/98	94.91	0.53	0-1.0	93.8%
	57/98	49.39	0.38	0-1.0	94.5%
	57/98	47.38	0.44	0-1.0	91.7%
Rat aorta in vivo	58/86	76.52	2.59	0	42.4%
	58/86	26.66	1.71	0	78.3%
Rat lilac artery in vivo	58/86	9.45	0.21	0-1.0	83.7%

Supplementary table 1. The locomotion data of soft MMDs.

Tissue	Imaging parameters (kV/ μ A)	Locomotion distance (mm)	Mean robot speed (mm/s)	Magnet rotation rate (Hz)	Localization success rate
Porcine stomach	51/97	29.47	0.28	0-0.3	65.4%
Porcine brain, skull model	51/60	28.92	0.38	0-0.3	97.4%
Porcine liver	54/97	33.32	0.39	0-0.3	98.9%
Porcine heart	55/68	20.95	0.17	0-0.3	52.6%
Bone (15-25 mm), porcine liver	60/98	25.25	0.20	0-0.3	46.6%
Bone (5-15 mm), porcine brain, skull model	54/97	204.95	0.55	0	96.6%
	54/97	30.62	0.73	0	74.6%

Supplementary table 2. The locomotion data of liquid MMDs.

4. The domain randomization effectively expands dataset diversity, but the generated shapes of liquid robots appear purely geometric. Discussion on incorporating physics-based deformation models or real motion datasets would improve realism and generalization. We thank the reviewer for this insightful suggestion. We agree that enhancing the physical realism of synthetic data, particularly for deforming objects like liquid robots, is a promising direction for future work, as we mentioned in the main text:

“While mathematically generated spline curves introduce shape diversity, their simplified appearances lead to lower data coverage (Supplementary Fig. 5) and slightly reduce mAP50 for easy detections. However, this diversity enhances performance in complex tasks like tracking swarms under bone occlusions, particularly during dynamic shape transitions (splitting and merging). Future improvements could focus on refining MMD fidelity (Supplementary Fig. 7) and incorporating physics-based deformation models”

We have supplemented the initial investigation on integrating real data into model training and expanded upon in the discussion.

1. Investigation on synergizing synthetic and real data

To evaluate the potential of incorporating real motion data, we conducted an experiment where a model pre-trained on our synthetic data was fine-tuned with a small amount of real data. The key findings are:

- Initial challenge: naive fine-tuning on real data led to an imbalanced performance improvement; it boosted accuracy for well-represented classes but degraded performance for others.

- Solution: we implemented a balanced fine-tuning strategy that uses both synthetic and real data simultaneously. This approach successfully preserved the model's generalizable knowledge while adapting it to the real domain.
- Conclusion: Incorporating real images would improve downstream CV model performance. Synthetic and real data are complementary, not substitutive, with synthetic data providing scalability and diversity, and real data offering targeted realism.

2. Future Directions for Enhanced Realism

- Physics-Informed Synthesis: future directions of our framework could integrate physics-based deformation models to generate more physically plausible robot shapes and motions, thereby increasing the fidelity of the synthetic data.
- Temporal Modeling: the next step is to develop models that process video sequences. For such models, physics-based simulations could generate continuous, realistic robot trajectories, further closing the sim-to-real gap for dynamic tracking tasks.

We have expanded the discussion regarding this part “The proposed system can be improved in multiple aspects. First, more advanced generative models and **physics-based deformation models** can be adopted to produce more realistic X-ray images that closely mimic real-world anatomical and device-specific features [1]. Moreover, integrating domain knowledge, such as biomechanical models of tissue deformation, could generate time-resolved datasets reflecting physiological motion.”

Supplementary Fig. 7. Model performance after fine-tuning with real liquid MMD data.

Different numbers of real images are incorporated for fine-tuning. Simple fine-tuning using only real data leads to imbalanced performance—improving classes well represented in the real dataset while degrading underrepresented ones. In contrast, balanced fine-tuning with both synthetic and real data preserves previously learned knowledge while enabling adaptation to the real domain. N_{real} and N_{syn} denote the numbers of real and synthetic images, respectively. The model with 10.1M parameters is used in this analysis. The green and red dashed lines represent the results of model trained on synthetic images of D1-l and real images of D2-l, respectively.

References:

[1] Li, J., Zhang, C., Zhu, W. & Ren, Y. A comprehensive survey of image generation models based on deep learning. *Annals of Data Science* **12**, 141-170 (2025).

5. It would be essential to discuss how this approach compares with other guidance modalities such as ultrasound or electromagnetic tracking, especially regarding localization accuracy and clinical safety.

Thanks for the constructive suggestion.

In this paper, we focus on integrating synthetic data with X-ray imaging modalities, this approach can also potentially be applied to other imaging modalities, like the ultrasound imaging [1].

We have supplemented Extended Data Table 1 and 2 to summarize the current imaging and sensing modalities.

Extended Data Table 1. External sensing modalities for in vivo MMDs.

Technology	Advantages	Limitations	Application Scenarios	Functions	Reported Accuracy	Reference
X-ray Digital Subtraction Angiography (DSA)	Real-time 2D images, templates with radio-opaque devices	Ionizing radiation exposure, requires intravascular contrast agents	Neurovascular, cardiovascular vasculature	Guiding wire and catheter placement, procedural accuracy monitoring	Spatial: ~0.1 mm; Temporal: >10 fps	Frisken et al. [2]
Computed Tomography (CT)	Providing 3D visualization, eliminating anatomic superimposition	Ionizing radiation exposure, not real-time	Liver, kidney, lung	Guiding needle insertion, biopsy planning, lesion targeting	Spatial: ~0.4 mm	Hsieh et al. [3]
Computed Tomography Angiography (CTA)	Providing 3D visualization, compatible with metallic implants	Ionizing radiation exposure, not real-time	Neurovascular, cardiovascular vasculature	Providing 3D road-map for medical procedures	Spatial: ~0.5 mm	Fleischmann et al. [4]

Magnetic Resonance Imaging (MRI)	Excellent soft tissue imaging, no ionizing radiation	Not suitable for real-time imaging, not compatible with metallic tools	Neurovascular, cardiovascular vasculature	Tracking and guidance of devices	Spatial: 1-2 mm; Temporal: ~0.5 fps	Lim et al. [5] and Geethanath et al. [6]
Ultrasound Imaging (US)	Non-ionizing real-time imaging, portable for bedside and surgery	Acoustic shadowing, low-resolution imaging	Heart, liver, kidney, peripheral vessels	Guiding and tracking of minimally invasive devices	Spatial: 0.2-1 mm; Temporal: 20-50 fps	Ng et al. [7]
Laser Speckle Contrast Imaging (LSCI)	Real-time, high temporal resolution, safe non-contact	Limited spatial resolution, shallow penetration depth	Skin, superficial vasculature	Tracking and navigation of devices for targeted delivery	Relative error: 1% - 5%	Olmos et al. [8]
Opto-acoustic Imaging (OA)	High optical contrast with ultrasound resolution, deeper penetration than pure optical methods	Limited clinical adoption, image reconstruction complexity	Skin, breast, thyroid, synovial joints, vasculature	Monitoring vascular interventions, guiding biopsies, mapping sentinel lymph nodes	Spatial: 20-200 μm (scalable with depth)	Tian et al. [9]

Extended Data Table 2. Intraluminal sensing modalities for in vivo MMDs.

Technology	Advantages	Limitations	Application Scenarios	Functions	Reported Accuracy	Reference
Fiber Bragg Grating Sensing (FBG)	High sensitivity, immune to electromagnetic interference, multi-parameter measurements, compact and biocompatible	Mechanical hysteresis, signal drift over time	Bronchus, GI track	Real-time 3D shape reconstruction, contact force detection	Shape: <1 mm; Force: ~5 mN	Najafzadeh et al. [10]
Electromagnetic (EM) Tracking	Non-line-of-sight real-time 3D positioning, measures both position and orientation	Interfered by ferromagnetic materials, limited workspace	Bronchus, vasculature	Pose and trajectory tracking of robotic end-effectors	Position: 0.5-1.5 mm RMS; Orientation: 0.1°-0.3° RMS	Yaniv et al. [11]
Proprioceptive Force Sensing	Improved human-robot interaction	Hard to decouple distal contact force and	Kidney, vasculature	Measuring interaction forces based on motor	Highly system-dependent: ~2 mN; Linearity error: ~1%	Lv et al. [12]

		known friction, input delay		current changes		
Tactile Force Sensing	Direct measurement of the contact force at the device/tissue interface	Harsh sterilization, risk of electrical hazard and biocompatibility	Prostate, liver	Providing haptic feedback, collision detection and prediction	~55 mN	Du et al. [13]
Intravascular Ultrasound (IVUS)	Direct visualization of vessel walls, high resolution	Motion artifacts, limited information on structures outside the vessel	Cardiac vasculature, bronchus, GI tract	Guiding catheter-based procedures (e.g., atherectomy, tissue characterization)	Axial: ~20-200 μm ; Lateral: 150-400 μm	Peng et al. [14]
Optical Coherence Tomography (OCT)	Micron-level resolution, high sensitivity and accuracy	Limited penetration depth (1-20 mm), sensitive to motion	Cardiac vasculature, bronchus, GI tract	Optical biopsy and device guidance	Axial: 5-20 μm ; Lateral: 10-90 μm	Folgar et al. [15] and Spaide et al. [16]

References

- [1] Wang, C., Wang, T. & Sitti, M. Synthetic Data-Assisted Miniature Medical Robot Navigation via Ultrasound Imaging. *IEEE/ASME Transactions on Mechatronics* (2025).
- [2] Frisken, S., Haouchine, N., Du, R. & Golby, A. J. Using temporal and structural data to reconstruct 3D cerebral vasculature from a pair of 2D digital subtraction angiography sequences. *Computerized Medical Imaging and Graphics* 99, 102076 (2022).
- [3] Hsieh, J. & Flohr, T. Computed tomography recent history and future perspectives. *Journal of Medical Imaging* 8, 052109-052109 (2021).
- [4] Fleischmann, D., Chin, A. S., Molvin, L., Wang, J. & Hallett, R. Computed Tomography Angiography: A Review and Technical Update. *Radiologic Clinics* 54, 1-12 (2016). <https://doi.org/10.1016/j.rcl.2015.09.002>
- [5] Lim, W. H., Park, J. S., Park, J. & Choi, S. H. Assessing the reproducibility of high temporal and spatial resolution dynamic contrast-enhanced magnetic resonance imaging in patients with gliomas. *Scientific Reports* 11, 23217 (2021). <https://doi.org/10.1038/s41598-021-02450-5>
- [6] Geethanath, S. & Vaughan Jr, J. T. Accessible magnetic resonance imaging: A review. *Journal of Magnetic Resonance Imaging* 49, e65-e77 (2019). <https://doi.org/https://doi.org/10.1002/jmri.26638>
- [7] Ng, A. & Swanevelder, J. Resolution in ultrasound imaging. *Continuing Education in Anaesthesia, Critical Care & Pain* 11, 186-192 (2011).
- [8] González Olmos, A., Zilpelwar, S., Sunil, S., Boas, D. A. & Postnov, D. D. Optimizing the precision of laser speckle contrast imaging. *Scientific Reports* 13, 17970 (2023). <https://doi.org/10.1038/s41598-023-45303-z>
- [9] Tian, C. et al. Impact of System Factors on the Performance of Photoacoustic Tomography Scanners. *Physical Review Applied* 13, 014001 (2020). <https://doi.org/10.1103/PhysRevApplied.13.014001>
- [10] Najafzadeh, A. et al. Application of Fibre Bragg Grating Sensors in Strain Monitoring and Fracture Recovery of Human Femur Bone. *Bioengineering* 7 (2020). <https://pmc.ncbi.nlm.nih.gov/articles/PMC7552668/>.
- [11] Yaniv, Z., Wilson, E., Lindisch, D. & Cleary, K. Electromagnetic tracking in the clinical environment. *Medical physics* 36, 876-892 (2009).

- [12] Lv, C., Wang, S. & Shi, C. A High-Precision and Miniature Fiber Bragg Grating-Based Force Sensor for Tissue Palpation During Minimally Invasive Surgery. *Annals of Biomedical Engineering* 48, 669-681 (2020). <https://doi.org/10.1007/s10439-019-02388-w>
- [13] Du, L. et al. An implantable, wireless, battery-free system for tactile pressure sensing. *Microsystems & Nanoengineering* 9, 130 (2023). <https://doi.org/10.1038/s41378-023-00602-3>
- [14] Peng, C., Wu, H., Kim, S., Dai, X. & Jiang, X. Recent Advances in Transducers for Intravascular Ultrasound (IVUS) Imaging. *Sensors* 21 (2021).
- [15] Folgar, F. A., Yuan, E. L., Farsiu, S. & Toth, C. A. Lateral and axial measurement differences between spectral-domain optical coherence tomography systems. *Journal of biomedical optics* 19, 016014-016014 (2014).
- [16] Spaide, R. F. et al. Lateral resolution of a commercial optical coherence tomography instrument. *Translational Vision Science & Technology* 11, 28-28 (2022).

Minor comments

– It would be helpful to include approximate actuation force or torque values in Figures 5–7 to give readers a clearer sense of the physical scale of the robot’s motion.

Thanks for the helpful advice to improve the clarity. Details regarding the magnetic field, robot dimensions, and locomotion have been added to Figures 4–6 and their captions. For example, the caption of Figure 4 now includes the following information.

“The mean MMD translation speed and locomotion distance are denoted by v_r and l_r , respectively. LSR represents the localization success rate. Using a rotating 30 mm N45 cubic magnet, the magnetic torque and force range from 2.0 $\mu\text{N}\cdot\text{m}$ to 13.2 $\mu\text{N}\cdot\text{m}$ and from 0.1 mN to 0.4 mN, respectively. Soft MMD dimensions: 1.5 mm in diameter and 5.0 mm in length.”

– Please clarify whether the parylene-C coating maintains its mechanical and chemical stability after repeated magnetic actuation cycles. This information would support the long-term applicability of the system.

Thanks for the valuable comment.

- Parylene C is an FDA-approved, biocompatible, and hemocompatible material with a proven safety record spanning over four decades in biomedical applications, including blood-contacting implants such as stents, guidewires, and catheters [1–3].
- Our previous studies showed that Parylene C-coated PDMS thin films withstand repeated large-deformation bending without delamination or wear [4], confirming their short-term mechanical and chemical stability for dynamic interventional use.
- However, while short-term performance is well established, long-term stability under chronic physiological conditions remains to be fully assessed. Continuous mechanical stress, enzymatic activity, and blood exposure in vivo may gradually degrade the coating or alter its properties. Thus, Parylene C is well suited for short-term actuation-based applications, but its suitability for long-term implantation requires further investigation into aging, fatigue resistance, and sustained hemocompatibility.

This information has been added to the manuscript:

“Parylene C is an FDA-approved, biocompatible, and hemocompatible material with a proven safety record spanning over four decades in biomedical applications, including blood-

contacting implants such as stents, guidewires, and catheters. Our previous studies showed that Parylene C-coated PDMS thin films withstand repeated large-deformation bending without delamination or wear, confirming their short-term mechanical and chemical stability for dynamic interventional use. However, while short-term performance is well established, long-term stability under chronic physiological conditions requires further investigation into aging, fatigue resistance, and sustained hemocompatibility.”

References:

- [1] Cobo, A. M. (2017). Parylene-Based Implantable Interfaces for Biomedical Applications (Doctoral dissertation, University of Southern California).
- [2] Sonmezoglu, S., Fineman, J. R., Maltepe, E., & Maharbiz, M. M. (2021). Monitoring deep-tissue oxygenation with a millimeter-scale ultrasonic implant. *Nature Biotechnology*, 39(7), 855-864.
- [3] Guo, H., Bai, W., Ouyang, W., Liu, Y., Wu, C., Xu, Y., & Rogers, J. A. (2022). Wireless implantable optical probe for continuous monitoring of oxygen saturation in flaps and organ grafts. *Nature communications*, 13(1), 3009.
- [4] Wang, T., Ugurlu, H., Yan, Y., Li, M., Li, M., Wild, A. M., Yildiz, E., Schneider, M., Sheehan, D., Hu, W., & Sitti, M. (2022). Adaptive wireless millirobotic locomotion into distal vasculature. *Nature communications*, 13(1), 4465.

– Providing basic metadata for the open-source dataset, such as robot diameter, material composition, and magnetization, would make the resource more accessible and useful for other researchers.

Thanks for the helpful advice. We have supplemented MMD specifications to the dataset and supplementary information.

Supplementary Fig. 15. Specifications of soft and liquid miniature medical devices.

– It may also be worth briefly mentioning transformer-based or temporal models as potential future directions for improving tracking performance in continuous video sequences.

Thanks for the suggestion. We have supplemented the relevant information in the discussion section.

“Second, other downstream CV models, such as transformer-based models [1], can be utilized to enhance the tracking performance or enable new capabilities. Temporal models for video-based tracking could shift from frame-wise detection to continuous localization, improving efficiency in dynamic fluoroscopic sequences”

References:

[1] Min, Z., Lai, J. & Ren, H. Innovating robot-assisted surgery through large vision models. *Nature Reviews Electrical Engineering*, 1-14 (2025).

Reviewer #3 (Remarks to the Author):

This paper presented a deep learning framework for training and guiding miniature medical devices. The authors introduced the first X-ray MMD dataset for medical robotic system, which is quite impressive. However, in its current form, the work appears incremental, primarily applying machine learning to a new dataset. If the main contribution is the model or framework, the manuscript lacks essential evaluation, such as cross-validation and comparisons with strong baselines, to support its claims. While the work holds significant potential and is of high interest, addressing the comments below will enhance the rigor of the claims:

We sincerely thank the reviewer for their positive feedback on the novelty of our dataset and their insightful comments.

- We have improved the manuscript to make the **contributions** articulated more clearly, which is not merely a new model applied to a new dataset.
 - It is an end-to-end framework for solving the critical **data scarcity** and sim-to-real gap in X-ray-guided **Miniature Medical Devices (MMDs)**.
 - It translates to a robotic system for **real-world MMD deployment** in real tissues (e.g., guiding the soft MMD from the rat aorta to the femoral artery).
- We agree that evaluation is crucial, for which we have done extensive evaluation and supplemented more experiments in the section **“Evaluation of MicroSyn-X”**.
 - **It is compared with baselines of conventional CV model training and clinical experts, and validated across ex vivo tissues, in vivo experiments, multiple CV models, and various imaging conditions.**

1. Clarification of Main Contributions:

We have revised the introduction and conclusion to state our contributions:

- **A synthetic data generation pipeline for miniature medical devices (MMD)**. We present a framework to generate large, diverse, and annotated X-ray datasets for MMDs at low cost, overcoming the fundamental barrier of data scarcity and eliminating manual labelling. This is not limited to a single device, adaptable to new MMDs and anatomical scenes.
- **Bridging the sim-to-real gap for MMD tracking**. We demonstrate that a computer vision (CV) model trained on our synthetic data performs comparably or better than a baseline CV model trained on expensive real data (Extended Data Fig. 2. and Supplementary Fig. 6). This validates the quality and realism of our generated data.
- **System-level integration (CV model and robotic system for MMD deployment) and the validation on ex vivo and in vivo environments** (which has not been shown in the community before to the best of our knowledge). We move beyond a pure computer vision task by integrating the trained model into a full robotic system, demonstrating real-time tracking and control of MMDs in real tissues under X-ray imaging. This system-level implementation validates the entire framework's effectiveness for a potential clinical application.

The reviewer insightfully notes that if the contribution were a new model, strong baselines would be needed.

- However, our contribution is the data generation framework that makes the state-of-the-art downstream CV models viable in this domain (MMD under X-ray imaging).
- Our goal is not to beat current CV models with a novel architecture, but to show that our synthetic data enables these models to solve a problem where no data previously existed.

The contributions were summarized in the discussion section:

“The proposed framework advances MMD localization and deployment under X-ray fluoroscopy by overcoming key limitations of existing methods. MicroSyn-X expands medical data synthesis to MMD-specific conditions, generating high-fidelity X-ray images that incorporate realistic noise, occlusion, and low-contrast scenarios. It bridges the synthetic-to-real gap, demonstrating the feasibility of training models exclusively on synthetic data to perform robustly in clinical environments. With this framework, inexpensive high-quality data with large volume, expanded distribution, and accurate labelling is obtained to train generalizable downstream models. Furthermore, the robotic system serves as a functional platform for translating these advancements into clinical applications. This work facilitates clinical translation of MMDs in minimally invasive procedures, targeted therapies, and diagnostics.”

2. Extensive Evaluation and Comparisons with the Baseline.

We agree that rigorous evaluation is key. We have done extensive experiments to prove the efficacy of our synthetic data pipeline. The following baselines and evaluations are included in the paper, and the main text has been adjusted as follows to improve the clarity of the evaluation section:

“The evaluation aims to assess its ability to bridge the synthetic-to-real gap and robustness under unpredictable imaging conditions and anatomical variability. It is compared with baselines of conventional CV model training and clinical experts, and validated across ex vivo tissues, in vivo experiments, multiple CV models, and various imaging conditions.”

2.1 Baseline 1: Conventional CV model training process

We trained a CV model on a set of high-cost, manually annotated real X-ray data. This represents the current standard CV model training process.

As shown in **Extended Data Fig. 2**, the model trained on synthetic data achieves comparable or better performance to the real-data model on diverse test sets of real tissues.

Extended Data Fig. 2. Evaluation of MicroSyn-X. **a.** Schematic of the evaluation process. **b.** Cross-domain feature comparison using CNN-extracted features visualized by dimensionality reduction (DR). **c.** Comparison of soft MMD datasets via PCA, with dashed lines indicating data distribution boundaries. **d.** Performance of models trained on synthetic and real soft MMD data across classification, detection, and segmentation tasks, measured by AP, mAP50, and mAP50:95. mAP(B) and mAP(M) denote bounding-box and mask mAPs, respectively. Tissue index indicates datasets: soft MMDs in porcine brain with embedded bones, porcine brain, heart, liver, stomach, heart 3D vessels, and in vivo animals. Models with a parameter size of

10.1M is utilized. **e.** Cross-domain comparison of liquid MMD datasets. **f.** Performance comparison between models trained on synthetic and real liquid MMD data. Tissue index indicates datasets: liquid MMDs in porcine brain, heart, liver, stomach, brain or liver with embedded bones, and robot swarm under bone occlusion.

Supplementary Fig. 6. Comparison between models trained with synthetic and real data.

a. Performance of models trained on synthetic and real soft MMD data, measured by mAP50(M), and mAP50:95(M). Tissue index indicates datasets: soft MMDs in porcine brain with embedded bones, porcine brain, heart, liver, stomach, heart 3D vessels, and in vivo animals. **b.** Performance of models trained on synthetic and real liquid MMD data. Tissue index indicates datasets: liquid MMDs in porcine brain, heart, liver, stomach, brain or liver with embedded bones, and robot swarm under bone occlusion.

2.2 Baseline 2: Comparison with clinical experts

As shown in **Extended Data Fig. 4**, to evaluate the clinical relevance of MicroSyn-X, we compared its performance with expert annotations under challenging imaging conditions. Six soft robots were placed within a 3D lumen phantom under a skull model and imaged across varying X-ray voltages and currents. Both clinical experts and the model were tasked with counting visible robots. Quantitative analysis reveals that the model outputs match expert consensus (**Extended Data Fig. 4c**), achieving high agreement even in low-contrast and high-noise environments.

Extended Data Fig. 4. Robustness evaluation of MicroSyn-X. a. Evaluation task comparing clinical expert labels with MicroSyn-X. Videos of six static robots inside a skull model were recorded across varying imaging parameters for quantifying detectable robots. The numbers represent the indexes of imaging setting. **b.** The imaging effect of MMDs under different imaging settings. Error bars represent standard deviation across different MMDs. **c.** Expert-label comparison. Model-derived robot counts and expert annotations are evaluated across eight imaging parameters. Error bars represent standard deviation across five clinicians and model detection confidences. **d.** Detection confidence distribution relative to contrast and noise levels. Effective detections require a segmentation mask IoU > 0.5 with ground truth. A subset of dataset D2-s was manually annotated, with defined contrast and noise computation regions defined.

2.3 Validation on different CV models and MMDs.

As shown in **Extended Data Fig. 2**, CV models of different sizes trained on synthetic data of soft and liquid MMDs achieves good performance, with the average mAP50(M) and mAP50:95(M) exceeding 0.9 and 0.7, respectively.

Extended Data Fig. 2. Evaluation of MicroSyn-X. g. Performance evaluation of models trained on synthetic data of soft and liquid MMDs, with varying model sizes (2.8M, 10.1M, 22.4M, 27.6M).

2.5 Evaluation on real ex-vivo tissues and live animals.

Crucially, the CV model is integrated into a robotic system for real-time tracking and control of MMDs in real tissues, demonstrating good generalization when tested on new backgrounds or MMD appearance.

Beyond quantitative detection metrics, the most significant evaluation is the successful in-vivo deployment of the CV Model. The CV model trained exclusively on our synthetic data can reliably track a robot in a living animal and facilitate deploying the MMD to a target site is the ultimate validation of our framework's utility and performance.

1. Since the framework is used for real-time clinical guidance, how fast could deep learning framework react to the real-time image? When the image quality suddenly become bad, any alternative solutions to prevent errors?

Thanks for the comment.

- The tracking algorithm achieves real-time performance (21.6 ms latency).
- The localization robustness under poor imaging conditions is achieved via software filtering, hardware adjustments, and human oversight.

Detailed explanation:

1. Real-time performance capability:

We employ a dual-mode algorithm for object localization, as shown in **Supplementary Fig. 2**, including

- Local tracking mode: focused tracking on regions of interest (primary operational mode). 21.6 ± 1.6 ms per object (model size: 22.4M) from reading the raw data to output the results.

- Global re-initialization: Comprehensive search across the entire image (used sparingly for initialization). 333.8 ± 4.9 ms (model size: 22.4M, 25 patches).

Hardware: NVIDIA RTX Titan, Intel Xeon 5220 @2.2GHz, 64GB RAM.

This processing speed is fully compatible with the X-ray image capture rate of up to 15 frames per second (fps). The speed can be further improved by future algorithm optimizations and more powerful hardware.

Supplementary Fig. 2. Schematic of the object localization algorithm. a. Schematics of the global mode. The input image is partitioned into overlapping patches, where patch size is adaptively determined based on image scale (pixels/mm). Each patch is processed independently by the model, and the resulting detections are aggregated and merged to produce the final localization output. b. Schematics of the local mode. Leveraging prior localization results, the algorithm focuses computation on regions of interest (ROIs), feeding only these refined subregions into the model to enhance the efficiency. In all figures, scale bars represent 5 mm.

2. Solutions to handle potential sudden drops in image quality

We have implemented a multi-layered strategy, encompassing both software and hardware solutions, to prevent errors and ensure robust tracking when image quality degrades.

- Potential cases of low image quality: we have systematically investigated the factors leading to poor image quality (low voltage/current, high frame rate, fast object movement >20 mm/s), as quantified in **Extended Data Fig. 1**.
- Software solutions: the algorithm includes a filtering step that uses multiple criteria to validate detections and exclude false localizations caused by poor image quality. These criteria include:

- Detection confidence: a minimum confidence threshold from the CV model.
- MMD dimensions: checking for consistency in the detected object's width, length, and size.
- Temporal consistency: evaluating the distance of the current position to the last position.
- Anatomical context: considering the distance to the lumen centerline.
- Historical success rate: monitoring the localization success rate over the recent 10 frames.
- Hardware and protocol solutions:
 - Imaging controls: the C-arm fluoroscopy settings (voltage and current) are maintained above a minimum threshold (50 kV, 50 μ A) to ensure basic quality.
 - Speed limit: the MMD translation speed is controlled under 3 mm/s to minimize motion blur.
 - Actuation: if a MMD is lost for an extended period, the magnet rotates at a lower frequency or move away to avoid occluding MMDs. Under the continuous capture mode of C-arm fluoroscopy, the frame rate is adapted based on object movement. In this way, a low-frame-rate capture mode is triggered for a better image quality.
- Human intervention:
 - if the MMD is lost, the operator can adjust the C-arm angle to obtain a better imaging perspective, increase the machine's voltage and current, and modify the imaging field of view.

This approach to handling image quality is integrated into the tracking algorithm, and the pseudocode is provided in **Extended Data Fig. 6**.

These figures have been added to the main text

“In detection-based MMD tracking, the CV model localizes the MMD in individual frames (**Supplementary Fig. 2**), and the tracking algorithm links these detections into trajectories. To handle the dynamic, low-contrast, and noisy imaging environment with frequent occlusions, the system mitigates false positives, missed detections, and abrupt appearance changes through several strategies. Each frame is preprocessed (e.g., brightness/contrast adjustment, histogram equalization) to enhance MMD visibility. Detection outputs are filtered by confidence scores, geometric consistency, spatial plausibility, and temporal persistence, and the adaptive Kalman filter (AKF) interpolates missing data during occlusions (**Extended Data Fig. 6**; “Measures for handling degradation in image quality” in Methods). “

Extended Data Fig. 1. Effect of imaging parameters on image quality. **a.** Impact of X-ray voltage and current on image contrast and noise in static capture mode. **b–e.** Influence of voltage, current, and frame capture rate (f) on contrast and noise in continuous capture mode. **f.** Quantification of motion-induced blurriness as a function of magnet translation speed (v_{mag}) under continuous capture. Blurriness is assessed via the normalized Tenengrad sharpness metric, where lower values indicate increased motion blur. All experiments were performed using the XPERT 80 imaging system (KUBTEC Scientific). In all figures, scale bars represent 5 mm.

Algorithm 1 Object Tracking

```
1: Initialize:  $t \leftarrow 0$  ▷ Frame index
2:  $\mathcal{T} \leftarrow \emptyset$  ▷ Set of active tracks
3:  $\mathcal{K} \leftarrow \emptyset$  ▷ Set of Kalman filters
4:  $\mathcal{C} \leftarrow \emptyset$  ▷ Set of candidate detections with counters
5:  $T_{\text{confirm}} \leftarrow 5$  ▷ Frame threshold for track confirmation
6: procedure PROCESSFRAME( $I_t$ )
7:   if  $t = 0$  then
8:      $\mathcal{D}_t \leftarrow \text{global}(I_t)$  ▷ Initial global detection  $d_i \in \mathcal{D}_t$ 
9:      $\text{ROI}_i \leftarrow \text{createROI}(d_i)$ 
10:     $\mathbf{x}_i^0 \leftarrow [p_x, p_y, v_x, v_y]^\top$  ▷ Initial state
11:     $\mathcal{K}_i \leftarrow \text{KalmanInit}(\mathbf{x}_i^0, \mathbf{P}_0)$ 
12:     $\mathcal{T} \leftarrow \mathcal{T} \cup \{\tau_i\}$ 
13:     $\text{age}(\tau_i) \leftarrow 1$ 
14:   else
15:     Step 1: Prediction for existing tracks  $\tau_i \in \mathcal{T}$ 
16:      $\hat{\mathbf{x}}_i^t \leftarrow \mathcal{K}_i(\cdot)$  ▷ Kalman prediction
17:      $\text{ROI}_i \leftarrow \text{updateROI}(\hat{\mathbf{x}}_i^t)$ 
18:     Step 2: Detection and association for existing tracks  $\tau_i \in \mathcal{T}$ 
19:      $\mathcal{D}_i \leftarrow \text{local}(I_t, \text{ROI}_i)$ 
20:     if  $\mathcal{D}_i = \emptyset$  then
21:        $\mathcal{D}_i \leftarrow \text{global}(I_t)$ 
22:       if  $\mathcal{D}_i = \emptyset$  then
23:          $\text{stopActuation}()$ 
24:          $\text{retractMagnet}()$ 
25:          $\mathcal{D}_i \leftarrow \text{global}(I_t)$  ▷ Re-attempt detection
26:       end if
27:     end if
28:      $d_i^* \leftarrow (\hat{\mathbf{x}}_i^t, \mathcal{D}_i)$ 
29:     if  $d_i^* \neq \emptyset$  then
30:        $\mathcal{K}_i(\mathbf{z}_i^t)$  ▷ Update with measurement
31:        $\text{age}(\tau_i) \leftarrow \text{age}(\tau_i) + 1$ 
32:     end if
33:     Step 3: New candidate detection and confirmation
34:      $\mathcal{D}_t^{\text{new}} \leftarrow \text{global}(I_t) \setminus \{\text{associated detections}\}$   $d_j \in \mathcal{D}_t^{\text{new}}$ 
35:     if  $d_j \in \mathcal{C}$  then ▷ Existing candidate
36:        $c_j \leftarrow c_j + 1$  ▷ Increment counter
37:       if  $c_j \geq T_{\text{confirm}}$  then
38:         Initialize new track  $\tau_j$ 
39:         Initialize new Kalman filter  $\mathcal{K}_j$ 
40:          $\mathcal{T} \leftarrow \mathcal{T} \cup \{\tau_j\}$ 
41:          $\mathcal{C} \leftarrow \mathcal{C} \setminus \{d_j\}$  ▷ Remove from candidates
42:       end if
43:     else
44:        $\mathcal{C} \leftarrow \mathcal{C} \cup \{d_j\}$  ▷ Add new candidate
45:        $c_j \leftarrow 1$  ▷ Initialize counter
46:     end if
47:     Step 4: Cleanup stale candidates  $d_j \in \mathcal{C}$ 
48:     if detection in current frame matching  $d_j$  then
49:        $c_j \leftarrow \max(0, c_j - 1)$  ▷ Decrement counter
50:       if  $c_j = 0$  then
51:          $\mathcal{C} \leftarrow \mathcal{C} \setminus \{d_j\}$  ▷ Remove stale candidate
52:       end if
53:     end if
54:   end if
55:    $t \leftarrow t + 1$ 
56: end procedure
```

Extended Data Fig. 6. Pseudocode of the object tracking algorithm.

2. The authors used synthesized and auto-labeled X-ray images for model training. However, if the training dataset can be auto-labeled, why not use it for testing directly?

We thank the reviewer for this important question. The **automatic labeling is an inherent part of the synthetic data generation pipeline** (The generation of auto-labeled synthetic data pipeline can be referred to Fig. 2); it is **not** a separate step that can be applied to arbitrary images, such as MMDs in real tissues.

To clarify our testing strategy:

- The goal is generalization to the real world: the ultimate objective of our work is to develop a model that performs reliably on real clinical x-ray images. Evaluating on a held-out set of auto-labeled synthetic data would not measure the critical synthetic-to-real transfer performance, which is the central challenge our work aims to address.
- Real data is the true benchmark: the most rigorous and clinically relevant evaluation is to test the model on real, manually annotated X-ray images that it has never seen during training. This is the only way to validate that our synthetic data generation pipeline successfully produces a model capable of operating in the target domain.

3. The authors applied stable diffusion for tissue generation. However, stable diffusion is known to suffer structural hallucinations, such as detail duplication/missing, and generate samples that lie outside the training distribution. How does the author prevent these hallucinations?

We thank the reviewer for this critical question.

- Our objective is to **generate diverse tissue backgrounds** to train a robust model for localizing MMDs
- The MMD's own image quality is kept controlled and realistic as there **a separate stage to integrate MMDs to the background** (Fig. 2c).

We acknowledge that structural hallucinations are a known challenge in diffusion models. Rather than seeking to eliminate them entirely, our strategy is twofold:

- **Background quality control**: proactively control and minimize severe artifacts that harm downstream CV model training,
- **Investigation of downstream model robustness to background quality**: systematically evaluate how different levels of background quality affect the downstream task, leveraging mild artifacts as a form of beneficial data augmentation.

Our approach is detailed below and summarized in the provided supplementary figures.

1. Background quality control

1.1. During model training:

We periodically saved candidate models during training and evaluated them using a multi-faceted protocol:

- quantitative fidelity (structural similarity index measure): we quantified structural similarity to real tissue images (**Fig. 3a, b**).
- feature distribution: we ensured generated images expanded upon the feature distribution of real data without diverging from it (**Fig. 3c**).
- Qualitative and human-centric assessment: an operator selected the final model based on the generation of realistic, randomized textures with appropriate illumination and contrast, avoiding surreal artifacts.

Fig. 3. Domain randomization of synthetic tissue images and open-sourced microrobot X-ray dataset. a. Real tissue images under X-ray imaging. **b.** Generation of precisely matched synthetic tissues with randomized and enhanced textures. SSIM represents the structural similarity index measure. **c.** Image generation with mask-guided conditioning and prompts. Randomization of masks and prompts significantly expands the dataset. **d.** Domain comparison of real and synthetic tissues. Features from 1,140 real and 24,803 synthetic tissue images, extracted using Inception V3, are visualized via principle component analysis (PCA). PC represents principal component.

1.3 During model inference:

We select generation parameters to minimize artifacts. The low-quality generation cases are categorized as:

- Artifacts: surreal textures, blurriness, high-frequency noise, grid-like or checkerboard patterns, and overly noisy images.
- Inconsistent physics: inconsistent illumination/exposure, incorrect contrast,
- Lack of realistic textures: smoothness in areas that should have texture.

These artifacts are illustrated in **Supplementary Fig. 14**.

Supplementary Figure 14. Quality assessment and parameter sensitivity in diffusion model inference. **a.** Characteristic artifacts in low-quality generations, categorized as: generation artifacts (surreal textures, blurriness, high-frequency noise, grid/checkerboard patterns), physical inconsistencies (non-uniform illumination, exposure mismatches, implausible contrast), and texture deficiencies (excessive smoothness in structurally detailed regions).

We tuned the number of diffusion steps and the guidance scale (ρ) to find a stable to minimize the occurrence of low-quality generation, as indicated by the green markers in Supplementary Fig. 3b.

Supplementary Fig. 14. Quality assessment and parameter sensitivity in diffusion model inference. **b.** Trade-offs between number of diffusion steps and classifier-free guidance scale: low diffusion step counts combined with high prompt guidance weights are prone to produce

unstable or distorted outputs. Parameters selected for final inference are indicated by green markers.

These contents have been added to the “**Diffusion model training and inference**” section in Methods.

2. Investigation of downstream model robustness to background quality

We created five synthetic datasets by blending 200 high-quality and 200 low-quality backgrounds in varying ratios (1.0, 0.86, 0.5, 0.15, 0.0). We then trained 20 instance segmentation models of different model sizes (2.8M, 10.1M, 22.4M, 27.6M) on these datasets and evaluated them on a held-out test set of real X-ray images (**Extended Data Fig. 2**).

Extended Data Fig. 3. Effect of synthetic background quality on downstream segmentation models. **a.** Dataset composition. 200 high- and 200 low-quality generated tissue backgrounds were used to construct five synthetic training sets with varying proportions of high-quality backgrounds (ratios: 1.0, 0.86, 0.5, 0.15, 0.0). **b.** Real-tissue test dataset. The test images include soft MMDs in porcine brain with embedded bone, porcine brain, heart, liver, stomach, heart 3D vessels, in vivo rabbit, in vivo rat, and in vivo rat spine at low magnification. Scale bars represent 5 mm.

Key Findings:

- For MMDs with high visibility and distinct features (e.g., stents): model performance is consistent or better across all background mixtures. In this case, low-quality backgrounds act as a beneficial data augmentation, improving robustness.
- For low-contrast or ambiguous targets (e.g., blurred white rectangles resembling background artifacts), the effect is model-capacity dependent:
 - Large Models (22.4M, 27.6M): A modest fraction (0.14) of low-quality backgrounds improved or maintained performance, acting as a regularizer that

forces the model to learn more robust features, while the performance degrades with a majority of low-quality data.

- Small Models (2.8M, 10.1M): Performance degraded with the inclusion of low-quality backgrounds, as their limited capacity made them susceptible to learning spurious correlations.

Extended Data Fig. 3. Effect of synthetic background quality on downstream segmentation models. c. Segmentation performance across background quality ratios and model sizes. For structurally distinct targets (e.g., stents) with high visibility, performance remains stable or improved regardless of background quality, where low-quality backgrounds provide beneficial augmentation. In contrast, for low-contrast or ambiguous targets (e.g., blurred white rectangles resembling background artifacts), inclusion of a modest fraction of low-quality backgrounds (ratio = 0.86) improves or maintains performance for larger models, but degrades performance for smaller models, evidenced by missed detections. The red dashed lines represent the result of model trained on real data. Scale bars represent 1 mm.

All the relevant data and model weights are available in the open-sourced dataset.

Therefore, our framework adopts the following strategies to ensure the downstream model performance. We achieve this through:

- Proactive quality control during diffusion model training and inference.
- For challenging imaging scenes, utilizing large models for their demonstrated robustness to background noise, effectively turning a potential weakness into a form of data augmentation.

We could further develop a learned filter or classifier to automatically score and select the most beneficial "bad" backgrounds for augmentation, improving the downstream model performance.

The following paragraph has been added to the main text.

“We also investigate the impact of synthetic background quality on downstream CV models (Extended Data Fig. 3). For MMDs with distinct features like stents, performance is largely unaffected by background quality, whereas for MMDs with ambiguous features, the effect is model-dependent: smaller models degrade with low-quality data, while larger models can utilize it as effective regularization. To tackle this issue, we adopt a two-phase quality control strategy during tissue generation (diffusion model selection and artifact minimization) and prioritize the utilization of large downstream models with its robustness to noise (“Diffusion model training and inference” in Methods). A classifier can be developed to automatically select backgrounds for CV model training as a future step.”

4. Generative models may fail to accurately reproduce pathological findings (such as tumors, fractures, or inflammations), and may synthesize non-existent pseudo-features or simply blurred details. This can mislead downstream diagnostic models learning incorrect guidance. Have the authors compared the framework on true dataset versus synthesized dataset?

We thank the reviewer for this important question. We clarify that **our framework targets MMD localization**, not disease diagnosis, which fundamentally changes the requirements for our synthetic data.

- Objective clarification: Our goal is to **generate diverse anatomical backgrounds for training robust MMD detectors**. The MMDs themselves are integrated as separate components (**Fig. 2c**).
- Handling of artifacts (see response to Comment #3): We proactively manage generative artifacts through systematic **quality control**. We also systematically evaluate how different levels of background quality affect the downstream task, leveraging mild artifacts as a form of beneficial data augmentation
- **Comparison with CV models trained on real images**: we trained a CV model on a set of high-cost, manually annotated real X-ray data. This represents the current standard CV model training process. As shown in Extended Data Fig. 2 (original Fig. 4), the model trained on synthetic data achieves comparable or better performance to the real-data model on diverse test sets of real images.

Fig. 2. Workflow of MicroSyn-X. **b.** Controlled tissue generation process. Stable Diffusion creates high-fidelity tissue images from user-defined masks and prompts. **c.** Integration of medical devices with tissues. Captured or generated robot images are seamlessly integrated into the background with flexible parameters, ensuring pixel-accurate labeling.

Extended Data Fig. 2. Evaluation of MicroSyn-X. **a.** Schematic of the evaluation process. **b.** Cross-domain feature comparison using CNN-extracted features visualized by dimensionality reduction (DR). **c.** Comparison of soft MMD datasets via PCA, with dashed lines indicating data distribution boundaries. **d.** Performance of models trained on synthetic and real soft MMD data across classification, detection, and segmentation tasks, measured by AP, mAP50, and

mAP50:95. mAP(B) and mAP(M) denote bounding-box and mask mAPs, respectively. Tissue index indicates datasets: soft MMDs in porcine brain with embedded bones, porcine brain, heart, liver, stomach, heart 3D vessels, and in vivo animals. Models with a parameter size of 22.4M is utilized. e. Cross-domain comparison of liquid MMD datasets. f. Performance comparison between models trained on synthetic and real liquid MMD data. Tissue index indicates datasets: liquid MMDs in porcine brain, heart, liver, stomach, brain or liver with embedded bones, and robot swarm under bone occlusion.

Supplementary Fig. 6. Comparison between models trained with synthetic and real data. a. Performance of models trained on synthetic and real soft MMD data, measured by mAP50(M), and mAP50:95(M). Tissue index indicates datasets: soft MMDs in porcine brain with embedded bones, porcine brain, heart, liver, stomach, heart 3D vessels, and in vivo animals. b. Performance of models trained on synthetic and real liquid MMD data. Tissue index indicates

datasets: liquid MMDs in porcine brain, heart, liver, stomach, brain or liver with embedded bones, and robot swarm under bone occlusion.

5. In Fig4, when evaluating the system robustness, the authors collected eight data indexes for comparing model versus human experts. What are these data indexes stand for? Are the testing data also synthetic or real? For robustness test, it is suggested to evaluate on tissue X-rays on different human subjects. Are there significant subject variations when it comes to unseen new subjects for the model?

We thank the reviewer for these critical questions regarding our evaluation's robustness. Please find our point-by-point response below.

1. Testing Data

All testing data presented consists of real X-ray images, including both ex vivo tissues and in vivo animal experiments. No synthetic data was used for testing, as the core objective was to validate the model's performance in real-world scenarios.

2. Explanation of the eight data indexes

The eight data indexes represent distinct imaging parameter settings (variations in X-ray voltage and current) that directly impact the quality of the MMD visualization. We have adjusted the figure to make it clearer.

Extended Data Fig. 4. Robustness evaluation of MicroSyn-X. **a.** Evaluation task comparing clinical expert labels with MicroSyn-X. Videos of six static robots inside a skull model were recorded across varying imaging parameters for quantifying detectable robots. The numbers represent the indexes of imaging setting. **b.** The imaging effect of MMDs under different imaging settings. Error bars represent standard deviation across different MMDs. **c.** Expert-label comparison. Model-derived robot counts and expert annotations are evaluated across eight imaging parameters. Error bars represent standard deviation across five clinicians and model detection confidences. **d.** Detection confidence distribution relative to contrast and noise levels. Effective detections require a segmentation mask IoU > 0.5 with ground truth. A subset of dataset D2-s was manually annotated, with defined contrast and noise computation regions defined.

3. Evaluation of subject variation and robustness

We have evaluated the model's performance across variations in tissue subjects, MMD subjects, and imaging conditions:

- **Tissue Variation**: the model was tested on a wide range of tissues (both ex vivo and in vivo) that were completely unseen during the training of both the computer vision and generative models. This includes significant differences in anatomical structures, tissue density, and background texture.
- **MMD Variation**: the MMDs themselves undergo substantial variations, including:
 - Soft MMD: Changes in size, dimension, bending, and frequent occlusion.
 - Liquid MMD: Extreme shape transformations, such as splitting and merging.
- **Imaging effect variation**: there are large variations in contrast, noise, brightness, and blurriness.

4. Testing on human tissues.

We agree that testing on human subjects is the ultimate goal, extensive animal experiments are currently the established standard for this development stage due to ethical and regulatory constraints. The anatomical similarities under X-ray imaging between the animal models used and humans provide a good approach to testing the proposed framework. Human trials are planned as the next stage of this research.

6. After the model training, what is the objective for navigation and guiding? When there are multiple paths to guide the robot, which path will the framework choose? It is recommended to add these decision rule details in the figure and discussions.

We thank the reviewer for this helpful comment. The objective it for **controlling miniature medical devices for medical interventions in tissues.**

1. Navigation objective

The primary objective after successful model training is to leverage the CV model for precisely controlling the MMD, enabling it to:

- Navigate to a target location within the anatomy, where the MMD needs to traverse physiological barriers, such as blood vessel bifurcations.
- Perform a desired diagnostic or therapeutic function, such as drug delivery, upon arrival.

This objective is now more clearly stated in the main text.

“Specifically, small-scale devices actuated by external fields can **navigate through enclosed spaces** challenging for conventional tethered tools, and offer functionalities like drug delivery and physiological property sensing.”

2. Path selection and decision rules

The reviewer correctly identifies a critical aspect of navigation. In our current experimental setup within the blood vessel network, the MMD path is largely anatomically deterministic. The vascular network presents a bifurcation structure rather than a complex web of equivalent paths, simplifying the decision-making process to following the correct branch at each bifurcation.

For the actuation, the path for the external magnet (which actuates the robot) is not chosen arbitrarily. It is calculated by a dedicated planning algorithm to maximize magnetic actuation and ensure path continuity and safety. Please refer to our prior work [1].

To avoid the confusion of robot arm and the miniature robot, we replace “soft robot”, “liquid robot” with “soft MMD” and “liquid MMD” throughout the manuscript,

References:

[1] Wang, C., Wang, T., Li, M., Zhang, R., Ugurlu, H., & Sitti, M. (2024). Heterogeneous multiple soft millirobots in three-dimensional lumens. *Science Advances*, 10(45), eadq1951.

Reviewer #3 (Remarks on code availability):

I reviewed the code briefly but didn't try running it. It seems clear and solid.

We appreciate the reviewer's effort in examining our code.

Response to Reviewers

We want to thank the editor and reviewers for their insightful comments, which allowed us to improve the quality of the current manuscript. We have addressed all these comments point-by-point in the following response. The comments we have received are colored blue, the responses are placed below, and the revisions to the original files are highlighted in yellow.

Editorial comments:

- Punctuation is not allowed in titles.

We have removed the punctuation with the new title “Synthetic X-Ray Driven Tracking and Control of Miniature Medical Devices” X-ray is a word itself.

- The paper's abstract (200 words maximum; without references) should serve both as a general introduction to the topic, and as a brief, non-technical summary of your main results and their implications. It should start by outlining the background to your work (why the topic is important) and the main question you have addressed (the specific problem that initiated your research), before going on to describe your new observations, main conclusions and their general implications, introduced by the phrase "Here we show" or equivalent. Because we hope that a broad range of researchers will be interested in your work, the abstract should be as accessible as possible, explaining essential but specialised terms concisely. We have supplemented more introductory contents in the abstract

“The clinical translation of miniature medical devices (MMDs) for minimally invasive surgery promises transformative advances in biomedical engineering, offering enhanced precision, reduced patient trauma, and faster recovery times. However, their effective deployment in complex anatomies under real-time X-ray guidance—a widely used surgical imaging modality—presents challenges like low imaging quality and difficulties of spatial robot control.”

Word counts of the abstract: 196.

- We prefer to avoid the use of terms like "new" and "novel" or “first”, as in practice it is difficult to truly ascribe something as completely new. As such, it can detract from the achievements of the work by generating discussions about its "newness" instead of its unique scientific aspects. Please remove such words from the Abstract and text.

We all removed all relevant terms.

- The length of the main article (without abstract or methods section) should be below 4000 words.

The main article has been reduced to 3997 words.

- The maximum number of display items (figures and tables) is 6. Please move additional items to Extended Data (see below for more information on this section).

We have reduced the number of main figures to 6.

- We recommend that authors remove cartoons and icons from figures, e.g. Fig 1.

We appreciate the suggestion to remove cartoons and icons from the figures. The schematic elements in Figure 1 are important for helping readers grasp the framework quickly and intuitively, thereby improving the readability of the manuscript. To address the concern, we have replaced the illustrative icon in the image synthesis framework with a more technical representation.

Additionally, all icons and cartoon elements in Figure 5 have been removed.

All papers in Nature Machine Intelligence include a detailed Data availability statement and Code availability statement. Please see examples from papers published in our journal. The data and code should be available to reviewers during peer review.

- Data availability: see our policy here <https://www.nature.com/natmachintell/editorial->

policies/reporting-standards#availability-of-data

In particular, please clearly list **all datasets that have been used or generated in this work**, providing links and references.

- Code should be available in a public repository and include a link to the References. Please also generate a DOI (see <https://guides.github.com/activities/citable-code/>) which makes your code more easily citable and discoverable by others and which makes the article more reproducible as a specific version of your code is linked to the paper. PLEASE ADD THIS DOI TO THE REFERENCE LIST AND CITE THE REFERENCE NUMBER IN THE CODE AVAILABILITY STATEMENT.

To reduce uncertainty for people who might want to use your code, it would be helpful if you could add a License to your repository that either permits any kind of re-use and usage (e.g. the MIT license), or the conditions for re-distribution (e.g. GPLv3). GitHub makes it easy to find and apply a license to a repository, see here: <https://help.github.com/en/github/building-a-strong-community/adding-a-license-to-a-repository>

Example of GitHub doi and ref in a paper published in Nature Machine Intelligence:

<https://www.nature.com/articles/s42256-022-00531-2#code-availability>

Thank you for the feedback. We have now:

- Added a **Data Availability** section and included the corresponding access links.
- Added a **Code Availability** section, created the DOI, and cited it accordingly.
- Confirmed that the **CC BY-NC 4.0 license** has been applied.

Please let us know if any further adjustments are required.

Reviewer #1 (Remarks to the Author):

This paper introduces MicroSyn-X, a framework for training computer vision models to detect and segment miniature medical devices (MMDs) under X-ray imaging. The authors address the critical challenge of data scarcity in medical imaging by developing an end-to-end pipeline that synthesizes high-fidelity, auto-labeled X-ray images. The framework combines a diffusion model for realistic background generation with a programmatic overlay of MMDs, incorporating domain randomization to ensure model robustness. The work is well-executed, the results are compelling, and the contribution of an open-source dataset is significant for the field.

The paper addresses a critical bottleneck in the clinical translation of miniature medical devices and, in my opinion, meets the standards of Nature Machine Intelligence. However,

for the sake of clarity and to further strengthen the paper's claims, a few points should be addressed before publication.

Comment 1: First, the central premise of the paper is the elimination of laborious manual annotation. The paper effectively explains how masks for the synthetic data are programmatically generated, which clearly supports this premise. However, the role and creation of the text prompts that guide the diffusion model should be detailed more explicitly. The generation of synthetic backgrounds relies on these prompts to define scene content, and this process could be interpreted as a form of manual input. The manuscript does not specify the number of different prompts used or the method by which they were created and programmatically varied. For instance, were hundreds of unique prompts manually typed, or was a small set of terms programmatically combined and randomized to create diversity? As this can be considered a manual step in the workflow, a brief discussion on the scale and effort of this "prompt engineering" phase, and how it compares to the labor of traditional annotation, would be valuable. This clarification would further solidify the paper's primary claim of significantly improved efficiency.

Response:

We thank the reviewer for this astute observation. We have now revised the main text to clarify that this step is highly automated (**programmatically combined and randomized**) and requires minimal, one-time effort, preserving the core efficiency advantage over manual annotation.

Our Method: Programmatic Prompt Generation

Our approach was based on programmatic combination and randomization, not manual per-prompt creation. The process is a one-time setup:

- **One-time curation**: We defined a small, fixed vocabulary of anatomical terms for each tissue category (e.g., "brain," "skull," "lumen," "vessel").
- **Programmatic generation**: During synthetic data creation, prompts are assembled automatically by randomly sampling and combining these predefined terms (e.g., "porcine brain within the skull," "lumen inside heart "). This does not require manual intervention for individual images.

Efficiency Comparison

This method is fundamentally more efficient than piece-wise manual work:

- **Our method**: A one-time setup of a small vocabulary (e.g., 10-20 terms) can programmatically generate a large number of unique prompts and images. The human effort is constant and minimal, regardless of dataset size.
- **Manual annotation**: Manual annotation or per-prompt typing requires labor that scales linearly with the dataset (e.g., 10,000 images require 10,000 manual actions).

Future Enhancement

We agree that prompt design is an important area. As a future direction, one could leverage large language models (LLMs) to generate even more nuanced and varied prompts dynamically, further automating this step [1].

The relevant contents have been added to the section “Diffusion model training and inference” in Methods.

References:

[1] Bozkurt, A. & Sharma, R. C. Generative AI and prompt engineering: The art of whispering to let the genie out of the algorithmic world. *Asian Journal of Distance Education* 18, i-vii (2023).

Comment 2: Second, the framework's use of a diffusion model to generate a diverse set of synthetic tissue backgrounds is a key component of its success. However, the manuscript does not specify whether there is a quality control or filtering step applied to the images after they are generated. This is an important detail because if a manual, or even a complex programmatic, filtering process is required to discard unrealistic or low-quality outputs, it would constitute another form of human-in-the-loop intervention or an additional layer of algorithmic complexity not fully described. It would strengthen the paper to clarify whether all generated backgrounds (without any filtering) were used to create the final synthetic dataset, or if a selection process was employed. If there was a filtering step, the authors should describe the criteria used—for example, manual inspection, a quantitative metric or another automated method. Providing this information would offer a more complete picture of the data generation pipeline's autonomy and the true level of automation achieved.

Thank you for raising this important point.

- We clarify that our framework does **not employ a post-hoc filtering** step on the generated backgrounds.
- We implement a proactive, **two-phase quality control** strategy during the diffusion model training phase (select the model with good generation performance) and model inference phase (tuning discussion steps and prompt guidance weight).
- We also systematically investigate the **influence of background quality on downstream model** performance and determine the conditions under which they are harmful or beneficial as a form of data augmentation.
- We prioritize the utilization of large downstream models due to its robustness to noise.

Our approach is detailed below and summarized in the provided supplementary figures.

1. Background quality control

1.1. During Model Training:

We periodically saved candidate models during training and evaluated them using a multi-faceted protocol:

- quantitative fidelity (structural similarity index measure): We quantified structural similarity to real tissue images (**Fig. 3a, b**).
- feature distribution: we ensured generated images expanded upon the feature distribution of real data without diverging from it (**Fig. 3c**).
- Qualitative and human-centric assessment: an operator selected the final model based on the generation of realistic, randomized textures with appropriate illumination and contrast, avoiding surreal artifacts.

1. **Fig. 3. Domain randomization of synthetic tissue images and open-sourced MMD X-ray dataset.** **a.** Real tissue images under X-ray imaging. **b.** Generation of precisely matched synthetic tissues with randomized and enhanced textures. SSIM represents the structural similarity index measure. **c.** Image generation with mask-guided conditioning and prompts. Randomization of masks and prompts significantly expands the dataset. **d.** Domain comparison of real and synthetic tissues. Features from 1,140 real and 24,803 synthetic tissue images, extracted using Inception V3, are visualized via principle component analysis (PCA). PC represents principal component.

1.2 During Model Inference:

We select generation parameters to minimize artifacts. The low-quality generation cases are categorized as:

- Artifacts: surreal textures, blurriness, high-frequency noise, grid-like or checkerboard patterns, and overly noisy images.

- Inconsistent physics: inconsistent illumination/exposure, incorrect contrast,
- Lack of realistic textures: smoothness in areas that should have texture.

These artifacts are illustrated in **Supplementary Fig. 14 a**.

Supplementary Fig. 14. Quality assessment and parameter sensitivity in diffusion model inference. a. Characteristic artifacts in low-quality generations, categorized as: generation artifacts (surreal textures, blurriness, high-frequency noise, grid/checkerboard patterns), physical inconsistencies (non-uniform illumination, exposure mismatches, implausible contrast), and texture deficiencies (excessive smoothness in structurally detailed regions).

We tuned the number of diffusion steps and the guidance scale (ρ) to find a stable to minimize the occurrence of low-quality generation, as indicated by the green markers in **Supplementary Fig. 14 b**.

Supplementary Figure 14. Quality assessment and parameter sensitivity in diffusion model inference. b. Trade-offs between number of diffusion steps and classifier-free guidance scale: low diffusion step counts combined with high prompt guidance weights are prone to produce unstable or distorted outputs. Parameters selected for final inference are indicated by green markers.

These contents have been added to the “Diffusion model training and inference” section in Methods.

2. Investigation of downstream model robustness to background quality

We created five synthetic datasets by blending 200 high-quality and 200 low-quality backgrounds in varying ratios (1.0, 0.86, 0.5, 0.15, 0.0). We then trained 20 instance segmentation models of different model sizes (2.8M, 10.1M, 22.4M, 27.6M) on these datasets and evaluated them on a held-out test set of real X-ray images (**Extended Data Fig. 3**).

Extended Data Fig. 3. Effect of synthetic background quality on downstream segmentation models. a. Dataset composition. 200 high- and 200 low-quality generated tissue backgrounds were used to construct five synthetic training sets with varying proportions of high-quality backgrounds. **b.** Real-tissue test dataset. The test images include soft MMDs in porcine brain with embedded bone, porcine brain, heart, liver, stomach, heart 3D vessels, in vivo rabbit, in vivo rat, and in vivo rat spine at low magnification. Scale bars represent 5 mm.

Key Findings:

- For MMDs with high visibility and distinct features (e.g., Stents): Model performance is consistent or better across all background mixtures. In this case, low-quality backgrounds act as a beneficial data augmentation, improving robustness.
- For low-contrast or ambiguous targets (e.g., blurred white rectangles resembling background artifacts), the effect is model-capacity dependent:

- Large Models (22.4M, 27.6M): A modest fraction (0.14) of low-quality backgrounds improved or maintained performance, acting as a regularizer that forces the model to learn more robust features, while the performance degrades with a majority of low-quality data.
- Small Models (2.8M, 10.1M): Performance degraded with the inclusion of low-quality backgrounds, as their limited capacity made them susceptible to learning spurious correlations.

Extended Data Fig. 3. Effect of synthetic background quality on downstream segmentation models. **c.** Segmentation performance across background quality ratios and model sizes. For structurally distinct targets (e.g., stents) with high visibility, performance remains stable or improved, where low-quality backgrounds provide beneficial augmentation. In contrast, for low-contrast or ambiguous targets (e.g., blurred white rectangles resembling background artifacts), inclusion of a modest fraction of low-quality backgrounds (ratio = 0.86) improves or maintains performance for larger models, but degrades performance for smaller models, evidenced by missed detections. The red dashed lines represent the result of model trained on real data. Scale bars represent 1 mm.

Therefore, our framework adopts the following strategies to ensure the downstream model performance. We achieve this through:

- Proactive quality control during diffusion model training and inference.
- For challenging imaging scenes, utilizing large models for their demonstrated robustness to background noise, effectively turning a potential weakness into a form of data augmentation.

We could further a learned filter or classifier to automatically score and select the most beneficial "bad" backgrounds for augmentation, improving the downstream model performance.

The following paragraph has been added to the main text.

“We also investigate the impact of synthetic background quality on downstream CV models (Extended Data Fig. 3). For MMDs with distinct features like stents, performance is largely unaffected by background quality, whereas for MMDs with ambiguous features, the effect is model-dependent: smaller models degrade with low-quality data, while larger models can utilize it as effective regularization. To tackle this issue, we adopt a two-phase quality control strategy during tissue generation (diffusion model selection and artifact minimization) and prioritize the utilization of large downstream models with its robustness to noise (“Diffusion model training and inference” in Methods). A classifier can be developed to automatically select backgrounds for CV model training as a future step.”

Comment 3: Third, the manuscript details a patch-based training strategy, where large images are subdivided to focus the model's attention on small regions of interest. This is a sound and well-justified approach for training. However, the manuscript is less explicit about how detection is performed during real-time deployment. It would be beneficial for the authors to clarify whether the deployed model processes the entire video frame, or if a similar patch-based subdivision is used during inference. This clarification is important for fully understanding the real-time performance metrics.

We thank the reviewer for this critical question. For real-time deployment, we use a dynamic, **dual-mode** approach that switches between processing the **entire frame** and **focused patches** to balance accuracy and speed.

The specific workflow, detailed in Supplementary Figure 2 and Extended Data Fig. 6, is as follows:

- **Global Search Mode (Patch-based)**: Used sparingly for initialization or recovery if the robot is lost. The full video frame is subdivided into overlapping patches for processing.
 - Latency: 333.8 ± 4.9 ms (for 25 patches using a 22.4M parameter model).
- **Local Tracking Mode (Full-frame ROI)**: This is the primary operational mode for real-time tracking. Once the robot is located, the model processes only a high-resolution, cropped Region of Interest (ROI) around the last known position, not the entire frame.
 - Latency: 21.6 ± 1.6 ms per object (from reading raw data to output).
 - Hardware: NVIDIA RTX Titan, Intel Xeon 5220 @2.2GHz, 64GB RAM.

This hybrid strategy is key to our system's real-time performance. The fast local mode maintains a high tracking rate, while the global mode ensures robustness. The switching logic between these modes is shown in **Extended Data Fig. 6** and the section “Computer vision model training and inference” in Methods.

Supplementary Fig. 2. Schematic of the object localization algorithm. a. Schematics of the global mode. The input image is partitioned into overlapping patches, where patch size is adaptively determined based on image scale (pixels/mm). Each patch is processed independently by the model, and the resulting detections are aggregated and merged to produce the final localization output. **b.** Schematics of the local mode. Leveraging prior localization results, the algorithm focuses computation on regions of interest (ROIs), feeding only these refined subregions into the model to enhance the efficiency. In all figures, scale bars represent 5 mm.

Algorithm 1 Object Tracking

```
1: Initialize:  $t \leftarrow 0$  ▷ Frame index
2:  $\mathcal{T} \leftarrow \emptyset$  ▷ Set of active tracks
3:  $\mathcal{K} \leftarrow \emptyset$  ▷ Set of Kalman filters
4:  $\mathcal{C} \leftarrow \emptyset$  ▷ Set of candidate detections with counters
5:  $T_{\text{confirm}} \leftarrow 5$  ▷ Frame threshold for track confirmation
6: procedure PROCESSFRAME( $I_t$ )
7:   if  $t = 0$  then
8:      $\mathcal{D}_t \leftarrow \text{global}(I_t)$  ▷ Initial global detection  $d_i \in \mathcal{D}_t$ 
9:      $\text{ROI}_i \leftarrow \text{createROI}(d_i)$ 
10:     $\mathbf{x}_i^0 \leftarrow [p_x, p_y, v_x, v_y]^\top$  ▷ Initial state
11:     $\mathcal{K}_i \leftarrow \text{KalmanInit}(\mathbf{x}_i^0, \mathbf{P}_0)$ 
12:     $\mathcal{T} \leftarrow \mathcal{T} \cup \{\tau_i\}$ 
13:     $\text{age}(\tau_i) \leftarrow 1$ 
14:   else
15:     Step 1: Prediction for existing tracks  $\tau_i \in \mathcal{T}$ 
16:      $\hat{\mathbf{x}}_i^t \leftarrow \mathcal{K}_i(\cdot)$  ▷ Kalman prediction
17:      $\text{ROI}_i \leftarrow \text{updateROI}(\hat{\mathbf{x}}_i^t)$ 
18:     Step 2: Detection and association for existing tracks  $\tau_i \in \mathcal{T}$ 
19:      $\mathcal{D}_i \leftarrow \text{local}(I_t, \text{ROI}_i)$ 
20:     if  $\mathcal{D}_i = \emptyset$  then
21:        $\mathcal{D}_i \leftarrow \text{global}(I_t)$ 
22:       if  $\mathcal{D}_i = \emptyset$  then
23:          $\text{stopActuation}()$ 
24:          $\text{retractMagnet}()$ 
25:          $\mathcal{D}_i \leftarrow \text{global}(I_t)$  ▷ Re-attempt detection
26:       end if
27:     end if
28:      $d_i^* \leftarrow (\hat{\mathbf{x}}_i^t, \mathcal{D}_i)$ 
29:     if  $d_i^* \neq \emptyset$  then
30:        $\mathcal{K}_i(\mathbf{z}_i^t)$  ▷ Update with measurement
31:        $\text{age}(\tau_i) \leftarrow \text{age}(\tau_i) + 1$ 
32:     end if
33:     Step 3: New candidate detection and confirmation
34:      $\mathcal{D}_t^{\text{new}} \leftarrow \text{global}(I_t) \setminus \{\text{associated detections}\}$   $d_j \in \mathcal{D}_t^{\text{new}}$ 
35:     if  $d_j \in \mathcal{C}$  then ▷ Existing candidate
36:        $c_j \leftarrow c_j + 1$  ▷ Increment counter
37:       if  $c_j \geq T_{\text{confirm}}$  then
38:         Initialize new track  $\tau_j$ 
39:         Initialize new Kalman filter  $\mathcal{K}_j$ 
40:          $\mathcal{T} \leftarrow \mathcal{T} \cup \{\tau_j\}$ 
41:          $\mathcal{C} \leftarrow \mathcal{C} \setminus \{d_j\}$  ▷ Remove from candidates
42:       end if
43:     else
44:        $\mathcal{C} \leftarrow \mathcal{C} \cup \{d_j\}$  ▷ Add new candidate
45:        $c_j \leftarrow 1$  ▷ Initialize counter
46:     end if
47:     Step 4: Cleanup stale candidates  $d_j \in \mathcal{C}$ 
48:     if detection in current frame matching  $d_j$  then
49:        $c_j \leftarrow \max(0, c_j - 1)$  ▷ Decrement counter
50:       if  $c_j = 0$  then
51:          $\mathcal{C} \leftarrow \mathcal{C} \setminus \{d_j\}$  ▷ Remove stale candidate
52:       end if
53:     end if
54:   end if
55:    $t \leftarrow t + 1$ 
56: end procedure
```

Extended Data Fig. 6. Pseudocode of the object tracking algorithm.

Reviewer #2 (Remarks to the Author):

The authors propose a framework that integrates synthetic X-ray data generation and vision-based robotic control of miniature magnetic devices. The overall idea is novel and technically valuable for improving localization in low-contrast clinical environments. However, several important aspects should be clarified or expanded to make this work more convincing from a robotic system perspective.

1. The magnetic manipulation system is well described, but the paper lacks information about its effective workspace. Even a simple quantitative range of field strength or torque near the robot would help define the physical boundary where tracking and actuation remain reliable. We thank the reviewer for this valuable suggestion. We have now supplemented the manuscript with Extended Data Fig. 5 to quantitatively define the effective workspace of our magnetic manipulation system.

The key concept we introduce is the "**actuation region**," which is the 3D spatial region where the permanent magnet (PM) must be positioned to enable MMD movement. This region is defined by specific magnetic field thresholds required for effective movement, which we determined experimentally:

- For soft MMDs inside lumens: actuation requires the magnetic torque and force to exceed a minimum threshold to overcome the resistance.
- For liquid MMDs on surfaces: actuation requires the magnetic field gradient along the desired direction of motion to exceed a threshold, while gradients in other directions remain below this level to prevent undesired movement.

These experimentally-derived thresholds [1] are used to calculate and visualize the 3D action regions in **Extended Data Fig. 5**. This visualization provides a clear, quantitative boundary for the physical workspace, where the actuation parameters are also presented.

References:

[1] Wang, C., Wang, T., Li, M., Zhang, R., Ugurlu, H., & Sitti, M. (2024). Heterogeneous multiple soft millirobots in three-dimensional lumens. *Science Advances*, 10(45), eadq1951.

Extended Data Fig. 5. Actuation regions of soft and liquid MMDs. **a.** Actuation of the soft MMD inside a lumen, driven by magnetic torque and force. **b.** Actuation region of the soft MMD in the plane of $x_r = 0$. When the rotating permanent magnet (PM) is positioned within this region, the magnetic torque and force exceed the thresholds to enable MMD motion. **c.** Three-dimensional actuation region of the soft MMD, considering geometric constraints to prevent potential collisions. **d.** Actuation of the soft MMD on a substrate, where motion is

achieved by utilizing the magnetic field gradient. **e.** Actuation region of the liquid MMD in the plane of $x_r = 0$. When the PM is located within this region, the magnetic field gradient along the desired direction exceeds the threshold to induce MMD movement. **f.** Three-dimensional actuation region of the liquid MMD. A larger PM can be employed to expand the actuation region.

2. The vision-based tracking is claimed to operate in real time. The authors should report the latency between X-ray acquisition, model inference, and actuation command. Quantifying this delay is essential to verify closed-loop stability during dynamic locomotion.

We thank the reviewer for this critical question regarding the real-time performance. We have quantified the latency for each step, which confirms the system's stability for dynamic locomotion (MMD speed <1.5 mm/s). The end-to-end process and its associated timings are as follows:

Image acquisition	Image processing	Actuation Command
High-resolution mode: 66.7 ms	Local mode: 21.6 ± 1.6 ms	From user command to arm execution: <112 ms
Standard-resolution mode: 33.3 ms	Global re-initialization: 333.8 ± 4.9 ms	

- Image acquisition:

X-ray Capture: The C-arm operates at an adaptive frame rate of

- 0-15 fps under the continuous high-resolution imaging mode (66.7 ms/frame minimum interval);
- 0-30 fps under the continuous standard-resolution imaging mode (33.3 ms/frame minimum interval), according to the C-arm machine datasheet (Fluoroscan InSight FD, Hologic GmbH).

- Image processing:

We employ a dual-mode algorithm for object localization, as shown in **Supplementary Fig. 2** and **Extended Data Fig. 6**, including

- Local tracking mode: focused tracking on regions of interest (primary operational mode). 21.6 ± 1.6 ms per object (model size: 22.4M) from reading the raw data to output the results.
- Global re-initialization: comprehensive search across the entire image (used sparingly for initialization). 333.8 ± 4.9 ms (model size: 22.4M, 25 patches).

Hardware: NVIDIA RTX Titan, Intel Xeon 5220 @2.2GHz, 64GB RAM.

This processing speed is fully compatible with the X-ray image capture rate of up to 30 frames per second (fps).

- Actuation Command:

- User-in-the-loop requirement: for clinical safety, the standard workflow requires the operator to approve the movement command. This is a mandated step for clinical deployment.
- System Response: after approval, the latency from command dispatch to the robotic arm executing the command is <112 ms.

The MMD's locomotion speed is below 1.5 mm/s, as shown in the supplementary tables. Each subsequent command is issued after the robot advances approximately 1–2 mm. This latency is acceptable for dynamic locomotion.

The relevant contents have been added to the section “Latency of the teleoperated robotic system” in Methods.

Supplementary Fig. 2. Schematic of the object localization algorithm. a. Schematics of the global mode. The input image is partitioned into overlapping patches, where patch size is adaptively determined based on image scale (pixels/mm). Each patch is processed independently by the model, and the resulting detections are aggregated and merged to produce the final localization output. **b.** Schematics of the local mode. Leveraging prior localization results, the algorithm focuses computation on regions of interest (ROIs), feeding only these refined subregions into the model to enhance the efficiency. In all figures, scale bars represent 5 mm.

Algorithm 1 Object Tracking

```
1: Initialize:  $t \leftarrow 0$  ▷ Frame index
2:  $\mathcal{T} \leftarrow \emptyset$  ▷ Set of active tracks
3:  $\mathcal{K} \leftarrow \emptyset$  ▷ Set of Kalman filters
4:  $\mathcal{C} \leftarrow \emptyset$  ▷ Set of candidate detections with counters
5:  $T_{\text{confirm}} \leftarrow 5$  ▷ Frame threshold for track confirmation
6: procedure PROCESSFRAME( $I_t$ )
7:   if  $t = 0$  then
8:      $\mathcal{D}_t \leftarrow \text{global}(I_t)$  ▷ Initial global detection  $d_i \in \mathcal{D}_t$ 
9:      $\text{ROI}_i \leftarrow \text{createROI}(d_i)$ 
10:     $\mathbf{x}_i^0 \leftarrow [p_x, p_y, v_x, v_y]^T$  ▷ Initial state
11:     $\mathcal{K}_i \leftarrow \text{KalmanInit}(\mathbf{x}_i^0, \mathbf{P}_0)$ 
12:     $\mathcal{T} \leftarrow \mathcal{T} \cup \{\tau_i\}$ 
13:     $\text{age}(\tau_i) \leftarrow 1$ 
14:   else
15:     Step 1: Prediction for existing tracks  $\tau_i \in \mathcal{T}$ 
16:      $\hat{\mathbf{x}}_i^t \leftarrow \mathcal{K}_i(\cdot)$  ▷ Kalman prediction
17:      $\text{ROI}_i \leftarrow \text{updateROI}(\hat{\mathbf{x}}_i^t)$ 
18:     Step 2: Detection and association for existing tracks  $\tau_i \in \mathcal{T}$ 
19:      $\mathcal{D}_i \leftarrow \text{local}(I_t, \text{ROI}_i)$ 
20:     if  $\mathcal{D}_i = \emptyset$  then
21:        $\mathcal{D}_i \leftarrow \text{global}(I_t)$ 
22:       if  $\mathcal{D}_i = \emptyset$  then
23:          $\text{stopActuation}()$ 
24:          $\text{retractMagnet}()$ 
25:          $\mathcal{D}_i \leftarrow \text{global}(I_t)$  ▷ Re-attempt detection
26:       end if
27:     end if
28:      $d_i^* \leftarrow (\hat{\mathbf{x}}_i^t, \mathcal{D}_i)$ 
29:     if  $d_i^* \neq \emptyset$  then
30:        $\mathcal{K}_i(\mathbf{z}_i^t)$  ▷ Update with measurement
31:        $\text{age}(\tau_i) \leftarrow \text{age}(\tau_i) + 1$ 
32:     end if
33:     Step 3: New candidate detection and confirmation
34:      $\mathcal{D}_t^{\text{new}} \leftarrow \text{global}(I_t) \setminus \{\text{associated detections}\}$   $d_j \in \mathcal{D}_t^{\text{new}}$ 
35:     if  $d_j \in \mathcal{C}$  then ▷ Existing candidate
36:        $c_j \leftarrow c_j + 1$  ▷ Increment counter
37:       if  $c_j \geq T_{\text{confirm}}$  then
38:         Initialize new track  $\tau_j$ 
39:         Initialize new Kalman filter  $\mathcal{K}_j$ 
40:          $\mathcal{T} \leftarrow \mathcal{T} \cup \{\tau_j\}$ 
41:          $\mathcal{C} \leftarrow \mathcal{C} \setminus \{d_j\}$  ▷ Remove from candidates
42:       end if
43:     else
44:        $\mathcal{C} \leftarrow \mathcal{C} \cup \{d_j\}$  ▷ Add new candidate
45:        $c_j \leftarrow 1$  ▷ Initialize counter
46:     end if
47:     Step 4: Cleanup stale candidates  $d_j \in \mathcal{C}$ 
48:     if detection in current frame matching  $d_j$  then
49:        $c_j \leftarrow \max(0, c_j - 1)$  ▷ Decrement counter
50:       if  $c_j = 0$  then
51:          $\mathcal{C} \leftarrow \mathcal{C} \setminus \{d_j\}$  ▷ Remove stale candidate
52:       end if
53:     end if
54:   end if
55:    $t \leftarrow t + 1$ 
56: end procedure
```

Extended Data Fig. 6. Pseudocode of the object tracking algorithm.

3. Robot locomotion results are shown only qualitatively. The authors should include metrics such as locomotion distance, actuation frequency, or success rate under occlusion to demonstrate control repeatability.

We have added two tables to quantitatively present the locomotion results, including tissue type, imaging settings, locomotion distance, mean robot speed (mm/s), magnet rotation rate (Hz), and localization success rate. **The localization success ratio (LSR)**, defined as the ratio of successfully localized frames to the total number of video frames, is used to quantify performance under different levels of occlusion and imaging conditions. These quantitative results are also demonstrated in the supplementary videos.

Tissue	Imaging parameters (kV/ μ A)	Locomotion distance (mm)	Mean robot Speed (mm/s)	Magnet rotation rate (Hz)	Localization success rate
Bone (15-25 mm), porcine liver	61/97	79.12	0.41	0.3-0.7	98.4%
Porcine heart	60/56	68.63	0.42	0.3-0.7	57.4%
	66/97	51.66	0.68	0.9-1.3	23.4%
Porcine liver	53/64	48.85	0.40	0.3-0.7	98.8%
Porcine stomach	51/97	60.03	0.33	0.3-0.7	99.6%
Bone (5-15 mm), porcine brain, skull model	56/97	67.52	0.28	0-0.7	78.4%
	56/97	42.55	0.36	0-0.7	71.9%
	56/97	32.93	0.24	0-0.7	73.7%
	56/97	22.41	0.20	0-0.7	81.9%
Porcine heart artery, skull model	73/97	83.23	0.30	0-0.7	80.0%
Porcine brain, skull model	54/98	31.56	0.21	0.3-0.7	95.1%
	57/97	74.84	0.43	0.3-0.7	98.6%
	57/97	75.95	0.38	0.3-0.7	99.1%
	57/97	53.18	0.29	0.3-0.7	95.5%
Rabbit femoral arteries in vivo	57/98	94.91	0.53	0-1.0	93.8%
	57/98	49.39	0.38	0-1.0	94.5%
	57/98	47.38	0.44	0-1.0	91.7%
Rat aorta in vivo	58/86	76.52	2.59	0	42.4%
	58/86	26.66	1.71	0	78.3%
Rat lilac artery in vivo	58/86	9.45	0.21	0-1.0	83.7%

Supplementary table 1. The locomotion data of soft MMDs.

Tissue	Imaging parameters (kV/ μ A)	Locomotion distance (mm)	Mean robot speed (mm/s)	Magnet rotation rate (Hz)	Localization success rate
Porcine stomach	51/97	29.47	0.28	0-0.3	65.4%
Porcine brain, skull model	51/60	28.92	0.38	0-0.3	97.4%
Porcine liver	54/97	33.32	0.39	0-0.3	98.9%
Porcine heart	55/68	20.95	0.17	0-0.3	52.6%
Bone (15-25 mm), porcine liver	60/98	25.25	0.20	0-0.3	46.6%
Bone (5-15 mm), porcine brain, skull model	54/97	204.95	0.55	0	96.6%
	54/97	30.62	0.73	0	74.6%

Supplementary table 2. The locomotion data of liquid MMDs.

4. The domain randomization effectively expands dataset diversity, but the generated shapes of liquid robots appear purely geometric. Discussion on incorporating physics-based deformation models or real motion datasets would improve realism and generalization. We thank the reviewer for this insightful suggestion. We agree that enhancing the physical realism of synthetic data, particularly for deforming objects like liquid robots, is a promising direction for future work, as we mentioned in the main text:

“While mathematically generated spline curves introduce shape diversity, their simplified appearances lead to lower data coverage (Supplementary Fig. 5) and slightly reduce mAP50 for easy detections. However, this diversity enhances performance in complex tasks like tracking swarms under bone occlusions, particularly during dynamic shape transitions (splitting and merging). Future improvements could focus on refining MMD fidelity (Supplementary Fig. 7) and incorporating physics-based deformation models”

We have supplemented the initial investigation on integrating real data into model training and expanded upon in the discussion.

1. Investigation on synergizing synthetic and real data

To evaluate the potential of incorporating real motion data, we conducted an experiment where a model pre-trained on our synthetic data was fine-tuned with a small amount of real data. The key findings are:

- Initial challenge: naive fine-tuning on real data led to an imbalanced performance improvement; it boosted accuracy for well-represented classes but degraded performance for others.

- Solution: we implemented a balanced fine-tuning strategy that uses both synthetic and real data simultaneously. This approach successfully preserved the model's generalizable knowledge while adapting it to the real domain.
- Conclusion: Incorporating real images would improve downstream CV model performance. Synthetic and real data are complementary, not substitutive, with synthetic data providing scalability and diversity, and real data offering targeted realism.

2. Future Directions for Enhanced Realism

- Physics-Informed Synthesis: future directions of our framework could integrate physics-based deformation models to generate more physically plausible robot shapes and motions, thereby increasing the fidelity of the synthetic data.
- Temporal Modeling: the next step is to develop models that process video sequences. For such models, physics-based simulations could generate continuous, realistic robot trajectories, further closing the sim-to-real gap for dynamic tracking tasks.

We have expanded the discussion regarding this part “The proposed system can be improved in multiple aspects. First, more advanced generative models and **physics-based deformation models** can be adopted to produce more realistic X-ray images that closely mimic real-world anatomical and device-specific features [1]. Moreover, integrating domain knowledge, such as biomechanical models of tissue deformation, could generate time-resolved datasets reflecting physiological motion.”

Supplementary Fig. 7. Model performance after fine-tuning with real liquid MMD data.

Different numbers of real images are incorporated for fine-tuning. Simple fine-tuning using only real data leads to imbalanced performance—improving classes well represented in the real dataset while degrading underrepresented ones. In contrast, balanced fine-tuning with both synthetic and real data preserves previously learned knowledge while enabling adaptation to the real domain. N_{real} and N_{syn} denote the numbers of real and synthetic images, respectively. The model with 10.1M parameters is used in this analysis. The green and red dashed lines represent the results of model trained on synthetic images of D1-l and real images of D2-l, respectively.

References:

[1] Li, J., Zhang, C., Zhu, W. & Ren, Y. A comprehensive survey of image generation models based on deep learning. *Annals of Data Science* **12**, 141-170 (2025).

5. It would be essential to discuss how this approach compares with other guidance modalities such as ultrasound or electromagnetic tracking, especially regarding localization accuracy and clinical safety.

Thanks for the constructive suggestion.

In this paper, we focus on integrating synthetic data with X-ray imaging modalities, this approach can also potentially be applied to other imaging modalities, like the ultrasound imaging [1].

We have supplemented Extended Data Table 1 and 2 to summarize the current imaging and sensing modalities.

Extended Data Table 1. External sensing modalities for in vivo MMDs.

Technology	Advantages	Limitations	Application Scenarios	Functions	Reported Accuracy	Reference
X-ray Digital Subtraction Angiography (DSA)	Real-time 2D images, templates with radio-opaque devices	Ionizing radiation exposure, requires intravascular contrast agents	Neurovascular, cardiovascular vasculature	Guiding wire and catheter placement, procedural accuracy monitoring	Spatial: ~0.1 mm; Temporal: >10 fps	Frisken et al. [2]
Computed Tomography (CT)	Providing 3D visualization, eliminating anatomic superimposition	Ionizing radiation exposure, not real-time	Liver, kidney, lung	Guiding needle insertion, biopsy planning, lesion targeting	Spatial: ~0.4 mm	Hsieh et al. [3]
Computed Tomography Angiography (CTA)	Providing 3D visualization, compatible with metallic implants	Ionizing radiation exposure, not real-time	Neurovascular, cardiovascular vasculature	Providing 3D road-map for medical procedures	Spatial: ~0.5 mm	Fleischmann et al. [4]

Magnetic Resonance Imaging (MRI)	Excellent soft tissue imaging, no ionizing radiation	Not suitable for real-time imaging, not compatible with metallic tools	Neurovascular, cardiovascular vasculature	Tracking and guidance devices	Spatial: 1-2 mm; Temporal: ~0.5 fps	Lim et al. [5] and Geethanath et al. [6]
Ultrasound Imaging (US)	Non-ionizing real-time imaging, portable for bedside and surgery	Acoustic shadowing, low-resolution imaging	Heart, liver, kidney, peripheral vessels	Guiding and tracking of minimally invasive devices	Spatial: 0.2-1 mm; Temporal: 20-50 fps	Ng et al. [7]
Laser Speckle Contrast Imaging (LSCI)	Real-time, high temporal resolution, safe non-contact	Limited spatial resolution, shallow penetration depth	Skin, superficial vasculature	Tracking and navigation of devices for targeted delivery	Relative error: 1% - 5%	Olmos et al. [8]
Opto-acoustic Imaging (OA)	High optical contrast with ultrasound resolution, deeper penetration than pure optical methods	Limited clinical adoption, image reconstruction complexity	Skin, breast, thyroid, synovial joints, vasculature	Monitoring vascular interventions, guiding biopsies, mapping sentinel lymph nodes	Spatial: 20-200 μ m (scalable with depth)	Tian et al. [9]

Extended Data Table 2. Intraluminal sensing modalities for in vivo MMDs.

Technology	Advantages	Limitations	Application Scenarios	Functions	Reported Accuracy	Reference
Fiber Bragg Grating Sensing (FBG)	High sensitivity, immune to electromagnetic interference, multi-parameter measurements, compact and biocompatible	Mechanical hysteresis, signal drift over time	Bronchus, GI track	Real-time 3D shape reconstruction, contact force detection	Shape: <1 mm; Force: ~5 mN	Najafzadeh et al. [10]
Electromagnetic (EM) Tracking	Non-line-of-sight real-time 3D positioning, measures both position and orientation	Interfered by ferromagnetic materials, limited workspace	Bronchus, vasculature	Pose and trajectory tracking of robotic end-effectors	Position: 0.5-1.5 mm RMS; Orientation: 0.1°-0.3° RMS	Yaniv et al. [11]
Proprioceptive Force Sensing	Improved human-robot interaction	Hard to decouple distal contact force and	Kidney, vasculature	Measuring interaction forces based on motor	Highly system-dependent: ~2 mN; Linearity error: ~1%	Lv et al. [12]

		known friction, input delay		current changes		
Tactile Force Sensing	Direct measurement of the contact force at the device/tissue interface	Harsh sterilization, risk of electrical hazard and biocompatibility	Prostate, liver	Providing haptic feedback, collision detection and prediction	~55 mN	Du et al. [13]
Intravascular Ultrasound (IVUS)	Direct visualization of vessel walls, high resolution	Motion artifacts, limited information on structures outside the vessel	Cardiac vasculature, bronchus, GI tract	Guiding catheter-based procedures (e.g., atherectomy, tissue characterization)	Axial: ~20-200 μm ; Lateral: 150-400 μm	Peng et al. [14]
Optical Coherence Tomography (OCT)	Micron-level resolution, high sensitivity and accuracy	Limited penetration depth (1-20 mm), sensitive to motion	Cardiac vasculature, bronchus, GI tract	Optical biopsy and device guidance	Axial: 5-20 μm ; Lateral: 10-90 μm	Folgar et al. [15] and Spaide et al. [16]

References

- [1] Wang, C., Wang, T. & Sitti, M. Synthetic Data-Assisted Miniature Medical Robot Navigation via Ultrasound Imaging. *IEEE/ASME Transactions on Mechatronics* (2025).
- [2] Frisken, S., Haouchine, N., Du, R. & Golby, A. J. Using temporal and structural data to reconstruct 3D cerebral vasculature from a pair of 2D digital subtraction angiography sequences. *Computerized Medical Imaging and Graphics* 99, 102076 (2022).
- [3] Hsieh, J. & Flohr, T. Computed tomography recent history and future perspectives. *Journal of Medical Imaging* 8, 052109-052109 (2021).
- [4] Fleischmann, D., Chin, A. S., Molvin, L., Wang, J. & Hallett, R. Computed Tomography Angiography: A Review and Technical Update. *Radiologic Clinics* 54, 1-12 (2016). <https://doi.org/10.1016/j.rcl.2015.09.002>
- [5] Lim, W. H., Park, J. S., Park, J. & Choi, S. H. Assessing the reproducibility of high temporal and spatial resolution dynamic contrast-enhanced magnetic resonance imaging in patients with gliomas. *Scientific Reports* 11, 23217 (2021). <https://doi.org/10.1038/s41598-021-02450-5>
- [6] Geethanath, S. & Vaughan Jr, J. T. Accessible magnetic resonance imaging: A review. *Journal of Magnetic Resonance Imaging* 49, e65-e77 (2019). <https://doi.org/https://doi.org/10.1002/jmri.26638>
- [7] Ng, A. & Swanevelder, J. Resolution in ultrasound imaging. *Continuing Education in Anaesthesia, Critical Care & Pain* 11, 186-192 (2011).
- [8] González Olmos, A., Zilpelwar, S., Sunil, S., Boas, D. A. & Postnov, D. D. Optimizing the precision of laser speckle contrast imaging. *Scientific Reports* 13, 17970 (2023). <https://doi.org/10.1038/s41598-023-45303-z>
- [9] Tian, C. et al. Impact of System Factors on the Performance of Photoacoustic Tomography Scanners. *Physical Review Applied* 13, 014001 (2020). <https://doi.org/10.1103/PhysRevApplied.13.014001>
- [10] Najafzadeh, A. et al. Application of Fibre Bragg Grating Sensors in Strain Monitoring and Fracture Recovery of Human Femur Bone. *Bioengineering* 7 (2020). <https://pmc.ncbi.nlm.nih.gov/articles/PMC7552668/>.
- [11] Yaniv, Z., Wilson, E., Lindisch, D. & Cleary, K. Electromagnetic tracking in the clinical environment. *Medical physics* 36, 876-892 (2009).

- [12] Lv, C., Wang, S. & Shi, C. A High-Precision and Miniature Fiber Bragg Grating-Based Force Sensor for Tissue Palpation During Minimally Invasive Surgery. *Annals of Biomedical Engineering* 48, 669-681 (2020). <https://doi.org/10.1007/s10439-019-02388-w>
- [13] Du, L. et al. An implantable, wireless, battery-free system for tactile pressure sensing. *Microsystems & Nanoengineering* 9, 130 (2023). <https://doi.org/10.1038/s41378-023-00602-3>
- [14] Peng, C., Wu, H., Kim, S., Dai, X. & Jiang, X. Recent Advances in Transducers for Intravascular Ultrasound (IVUS) Imaging. *Sensors* 21 (2021).
- [15] Folgar, F. A., Yuan, E. L., Farsi, S. & Toth, C. A. Lateral and axial measurement differences between spectral-domain optical coherence tomography systems. *Journal of biomedical optics* 19, 016014-016014 (2014).
- [16] Spaide, R. F. et al. Lateral resolution of a commercial optical coherence tomography instrument. *Translational Vision Science & Technology* 11, 28-28 (2022).

Minor comments

– It would be helpful to include approximate actuation force or torque values in Figures 5–7 to give readers a clearer sense of the physical scale of the robot’s motion.

Thanks for the helpful advice to improve the clarity. Details regarding the magnetic field, robot dimensions, and locomotion have been added to Figures 4–6 and their captions. For example, the caption of Figure 4 now includes the following information.

“The mean MMD translation speed and locomotion distance are denoted by v_r and l_r , respectively. LSR represents the localization success rate. Using a rotating 30 mm N45 cubic magnet, the magnetic torque and force range from 2.0 $\mu\text{N}\cdot\text{m}$ to 13.2 $\mu\text{N}\cdot\text{m}$ and from 0.1 mN to 0.4 mN, respectively. Soft MMD dimensions: 1.5 mm in diameter and 5.0 mm in length.”

– Please clarify whether the parylene-C coating maintains its mechanical and chemical stability after repeated magnetic actuation cycles. This information would support the long-term applicability of the system.

Thanks for the valuable comment.

- Parylene C is an FDA-approved, biocompatible, and hemocompatible material with a proven safety record spanning over four decades in biomedical applications, including blood-contacting implants such as stents, guidewires, and catheters [1–3].
- Our previous studies showed that Parylene C-coated PDMS thin films withstand repeated large-deformation bending without delamination or wear [4], confirming their short-term mechanical and chemical stability for dynamic interventional use.
- However, while short-term performance is well established, long-term stability under chronic physiological conditions remains to be fully assessed. Continuous mechanical stress, enzymatic activity, and blood exposure in vivo may gradually degrade the coating or alter its properties. Thus, Parylene C is well suited for short-term actuation-based applications, but its suitability for long-term implantation requires further investigation into aging, fatigue resistance, and sustained hemocompatibility.

This information has been added to the manuscript:

“Parylene C is an FDA-approved, biocompatible, and hemocompatible material with a proven safety record spanning over four decades in biomedical applications, including blood-

contacting implants such as stents, guidewires, and catheters. Our previous studies showed that Parylene C-coated PDMS thin films withstand repeated large-deformation bending without delamination or wear, confirming their short-term mechanical and chemical stability for dynamic interventional use. However, while short-term performance is well established, long-term stability under chronic physiological conditions requires further investigation into aging, fatigue resistance, and sustained hemocompatibility.”

References:

- [1] Cobo, A. M. (2017). Parylene-Based Implantable Interfaces for Biomedical Applications (Doctoral dissertation, University of Southern California).
- [2] Sonmezoglu, S., Fineman, J. R., Maltepe, E., & Maharbiz, M. M. (2021). Monitoring deep-tissue oxygenation with a millimeter-scale ultrasonic implant. *Nature Biotechnology*, 39(7), 855-864.
- [3] Guo, H., Bai, W., Ouyang, W., Liu, Y., Wu, C., Xu, Y., & Rogers, J. A. (2022). Wireless implantable optical probe for continuous monitoring of oxygen saturation in flaps and organ grafts. *Nature communications*, 13(1), 3009.
- [4] Wang, T., Ugurlu, H., Yan, Y., Li, M., Li, M., Wild, A. M., Yildiz, E., Schneider, M., Sheehan, D., Hu, W., & Sitti, M. (2022). Adaptive wireless millirobotic locomotion into distal vasculature. *Nature communications*, 13(1), 4465.

– Providing basic metadata for the open-source dataset, such as robot diameter, material composition, and magnetization, would make the resource more accessible and useful for other researchers.

Thanks for the helpful advice. We have supplemented MMD specifications to the dataset and supplementary information.

Supplementary Fig. 15. Specifications of soft and liquid miniature medical devices.

– It may also be worth briefly mentioning transformer-based or temporal models as potential future directions for improving tracking performance in continuous video sequences.

Thanks for the suggestion. We have supplemented the relevant information in the discussion section.

“Second, other downstream CV models, **such as transformer-based models** [1], can be utilized to enhance the tracking performance or enable new capabilities. Temporal models for video-based tracking could shift from frame-wise detection to continuous localization, improving efficiency in dynamic fluoroscopic sequences”

References:

[1] Min, Z., Lai, J. & Ren, H. Innovating robot-assisted surgery through large vision models. *Nature Reviews Electrical Engineering*, 1-14 (2025).

Reviewer #3 (Remarks to the Author):

This paper presented a deep learning framework for training and guiding miniature medical devices. The authors introduced the first X-ray MMD dataset for medical robotic system, which is quite impressive. However, in its current form, the work appears incremental, primarily applying machine learning to a new dataset. If the main contribution is the model or framework, the manuscript lacks essential evaluation, such as cross-validation and comparisons with strong baselines, to support its claims. While the work holds significant potential and is of high interest, addressing the comments below will enhance the rigor of the claims:

We sincerely thank the reviewer for their positive feedback on the novelty of our dataset and their insightful comments.

- We have improved the manuscript to make the **contributions** articulated more clearly, which is not merely a new model applied to a new dataset.
 - It is an end-to-end framework for solving the critical **data scarcity** and sim-to-real gap in X-ray-guided **Miniature Medical Devices (MMDs)**.
 - It translates to a robotic system for **real-world MMD deployment** in real tissues (e.g., guiding the soft MMD from the rat aorta to the femoral artery).
- We agree that evaluation is crucial, for which we have done extensive evaluation and supplemented more experiments in the section **“Evaluation of MicroSyn-X”**.
 - **It is compared with baselines of conventional CV model training and clinical experts, and validated across ex vivo tissues, in vivo experiments, multiple CV models, and various imaging conditions.**

1. Clarification of Main Contributions:

We have revised the introduction and conclusion to state our contributions:

- **A synthetic data generation pipeline for miniature medical devices (MMD)**. We present a framework to generate large, diverse, and annotated X-ray datasets for MMDs at low cost, overcoming the fundamental barrier of data scarcity and eliminating manual labelling. This is not limited to a single device, adaptable to new MMDs and anatomical scenes.
- **Bridging the sim-to-real gap for MMD tracking**. We demonstrate that a computer vision (CV) model trained on our synthetic data performs comparably or better than a baseline CV model trained on expensive real data (Extended Data Fig. 2. and Supplementary Fig. 6). This validates the quality and realism of our generated data.
- **System-level integration (CV model and robotic system for MMD deployment) and the validation on ex vivo and in vivo environments** (which has not been shown in the community before to the best of our knowledge). We move beyond a pure computer vision task by integrating the trained model into a full robotic system, demonstrating real-time tracking and control of MMDs in real tissues under X-ray imaging. This system-level implementation validates the entire framework's effectiveness for a potential clinical application.

The reviewer insightfully notes that if the contribution were a new model, strong baselines would be needed.

- However, our contribution is the data generation framework that makes the state-of-the-art downstream CV models viable in this domain (MMD under X-ray imaging).
- Our goal is not to beat current CV models with a novel architecture, but to show that our synthetic data enables these models to solve a problem where no data previously existed.

The contributions were summarized in the discussion section:

“The proposed framework advances MMD localization and deployment under X-ray fluoroscopy by overcoming key limitations of existing methods. MicroSyn-X expands medical data synthesis to MMD-specific conditions, generating high-fidelity X-ray images that incorporate realistic noise, occlusion, and low-contrast scenarios. It bridges the synthetic-to-real gap, demonstrating the feasibility of training models exclusively on synthetic data to perform robustly in clinical environments. With this framework, inexpensive high-quality data with large volume, expanded distribution, and accurate labelling is obtained to train generalizable downstream models. Furthermore, the robotic system serves as a functional platform for translating these advancements into clinical applications. This work facilitates clinical translation of MMDs in minimally invasive procedures, targeted therapies, and diagnostics.”

2. Extensive Evaluation and Comparisons with the Baseline.

We agree that rigorous evaluation is key. We have done extensive experiments to prove the efficacy of our synthetic data pipeline. The following baselines and evaluations are included in the paper, and the main text has been adjusted as follows to improve the clarity of the evaluation section:

“The evaluation aims to assess its ability to bridge the synthetic-to-real gap and robustness under unpredictable imaging conditions and anatomical variability. It is compared with baselines of conventional CV model training and clinical experts, and validated across ex vivo tissues, in vivo experiments, multiple CV models, and various imaging conditions.”

2.1 Baseline 1: Conventional CV model training process

We trained a CV model on a set of high-cost, manually annotated real X-ray data. This represents the current standard CV model training process.

As shown in **Extended Data Fig. 2**, the model trained on synthetic data achieves comparable or better performance to the real-data model on diverse test sets of real tissues.

Extended Data Fig. 2. Evaluation of MicroSyn-X. **a.** Schematic of the evaluation process. **b.** Cross-domain feature comparison using CNN-extracted features visualized by dimensionality reduction (DR). **c.** Comparison of soft MMD datasets via PCA, with dashed lines indicating data distribution boundaries. **d.** Performance of models trained on synthetic and real soft MMD data across classification, detection, and segmentation tasks, measured by AP, mAP50, and mAP50:95. mAP(B) and mAP(M) denote bounding-box and mask mAPs, respectively. Tissue index indicates datasets: soft MMDs in porcine brain with embedded bones, porcine brain, heart, liver, stomach, heart 3D vessels, and in vivo animals. Models with a parameter size of

10.1M is utilized. **e.** Cross-domain comparison of liquid MMD datasets. **f.** Performance comparison between models trained on synthetic and real liquid MMD data. Tissue index indicates datasets: liquid MMDs in porcine brain, heart, liver, stomach, brain or liver with embedded bones, and robot swarm under bone occlusion.

Supplementary Fig. 6. Comparison between models trained with synthetic and real data.

a. Performance of models trained on synthetic and real soft MMD data, measured by mAP50(M), and mAP50:95(M). Tissue index indicates datasets: soft MMDs in porcine brain with embedded bones, porcine brain, heart, liver, stomach, heart 3D vessels, and in vivo animals. **b.** Performance of models trained on synthetic and real liquid MMD data. Tissue index indicates datasets: liquid MMDs in porcine brain, heart, liver, stomach, brain or liver with embedded bones, and robot swarm under bone occlusion.

2.2 Baseline 2: Comparison with clinical experts

As shown in **Extended Data Fig. 4**, to evaluate the clinical relevance of MicroSyn-X, we compared its performance with expert annotations under challenging imaging conditions. Six soft robots were placed within a 3D lumen phantom under a skull model and imaged across varying X-ray voltages and currents. Both clinical experts and the model were tasked with counting visible robots. Quantitative analysis reveals that the model outputs match expert consensus (**Extended Data Fig. 4c**), achieving high agreement even in low-contrast and high-noise environments.

Extended Data Fig. 4. Robustness evaluation of MicroSyn-X. a. Evaluation task comparing clinical expert labels with MicroSyn-X. Videos of six static robots inside a skull model were recorded across varying imaging parameters for quantifying detectable robots. The numbers represent the indexes of imaging setting. **b.** The imaging effect of MMDs under different imaging settings. Error bars represent standard deviation across different MMDs. **c.** Expert-label comparison. Model-derived robot counts and expert annotations are evaluated across eight imaging parameters. Error bars represent standard deviation across five clinicians and model detection confidences. **d.** Detection confidence distribution relative to contrast and noise levels. Effective detections require a segmentation mask IoU $>$ 0.5 with ground truth. A subset of dataset D2-s was manually annotated, with defined contrast and noise computation regions defined.

2.3 Validation on different CV models and MMDs.

As shown in **Extended Data Fig. 2**, CV models of different sizes trained on synthetic data of soft and liquid MMDs achieves good performance, with the average mAP50(M) and mAP50:95(M) exceeding 0.9 and 0.7, respectively.

Extended Data Fig. 2. Evaluation of MicroSyn-X. g. Performance evaluation of models trained on synthetic data of soft and liquid MMDs, with varying model sizes (2.8M, 10.1M, 22.4M, 27.6M).

2.5 Evaluation on real ex-vivo tissues and live animals.

Crucially, the CV model is integrated into a robotic system for real-time tracking and control of MMDs in real tissues, demonstrating good generalization when tested on new backgrounds or MMD appearance.

Beyond quantitative detection metrics, the most significant evaluation is the successful in-vivo deployment of the CV Model. The CV model trained exclusively on our synthetic data can reliably track a robot in a living animal and facilitate deploying the MMD to a target site is the ultimate validation of our framework's utility and performance.

1. Since the framework is used for real-time clinical guidance, how fast could deep learning framework react to the real-time image? When the image quality suddenly become bad, any alternative solutions to prevent errors?

Thanks for the comment.

- The tracking algorithm achieves real-time performance (21.6 ms latency).
- The localization robustness under poor imaging conditions is achieved via software filtering, hardware adjustments, and human oversight.

Detailed explanation:

1. Real-time performance capability:

We employ a dual-mode algorithm for object localization, as shown in **Supplementary Fig. 2**, including

- Local tracking mode: focused tracking on regions of interest (primary operational mode). 21.6 ± 1.6 ms per object (model size: 22.4M) from reading the raw data to output the results.

- Global re-initialization: Comprehensive search across the entire image (used sparingly for initialization). 333.8 ± 4.9 ms (model size: 22.4M, 25 patches).

Hardware: NVIDIA RTX Titan, Intel Xeon 5220 @2.2GHz, 64GB RAM.

This processing speed is fully compatible with the X-ray image capture rate of up to 15 frames per second (fps). The speed can be further improved by future algorithm optimizations and more powerful hardware.

Supplementary Fig. 2. Schematic of the object localization algorithm. a. Schematics of the global mode. The input image is partitioned into overlapping patches, where patch size is adaptively determined based on image scale (pixels/mm). Each patch is processed independently by the model, and the resulting detections are aggregated and merged to produce the final localization output. b. Schematics of the local mode. Leveraging prior localization results, the algorithm focuses computation on regions of interest (ROIs), feeding only these refined subregions into the model to enhance the efficiency. In all figures, scale bars represent 5 mm.

2. Solutions to handle potential sudden drops in image quality

We have implemented a multi-layered strategy, encompassing both software and hardware solutions, to prevent errors and ensure robust tracking when image quality degrades.

- Potential cases of low image quality: we have systematically investigated the factors leading to poor image quality (low voltage/current, high frame rate, fast object movement >20 mm/s), as quantified in **Extended Data Fig. 1**.
- Software solutions: the algorithm includes a filtering step that uses multiple criteria to validate detections and exclude false localizations caused by poor image quality. These criteria include:

- Detection confidence: a minimum confidence threshold from the CV model.
- MMD dimensions: checking for consistency in the detected object's width, length, and size.
- Temporal consistency: evaluating the distance of the current position to the last position.
- Anatomical context: considering the distance to the lumen centerline.
- Historical success rate: monitoring the localization success rate over the recent 10 frames.
- Hardware and protocol solutions:
 - Imaging controls: the C-arm fluoroscopy settings (voltage and current) are maintained above a minimum threshold (50 kV, 50 μ A) to ensure basic quality.
 - Speed limit: the MMD translation speed is controlled under 3 mm/s to minimize motion blur.
 - Actuation: if a MMD is lost for an extended period, the magnet rotates at a lower frequency or move away to avoid occluding MMDs. Under the continuous capture mode of C-arm fluoroscopy, the frame rate is adapted based on object movement. In this way, a low-frame-rate capture mode is triggered for a better image quality.
- Human intervention:
 - if the MMD is lost, the operator can adjust the C-arm angle to obtain a better imaging perspective, increase the machine's voltage and current, and modify the imaging field of view.

This approach to handling image quality is integrated into the tracking algorithm, and the pseudocode is provided in **Extended Data Fig. 6**.

These figures have been added to the main text

“In detection-based MMD tracking, the CV model localizes the MMD in individual frames (**Supplementary Fig. 2**), and the tracking algorithm links these detections into trajectories. To handle the dynamic, low-contrast, and noisy imaging environment with frequent occlusions, the system mitigates false positives, missed detections, and abrupt appearance changes through several strategies. Each frame is preprocessed (e.g., brightness/contrast adjustment, histogram equalization) to enhance MMD visibility. Detection outputs are filtered by confidence scores, geometric consistency, spatial plausibility, and temporal persistence, and the adaptive Kalman filter (AKF) interpolates missing data during occlusions (**Extended Data Fig. 6**; “Measures for handling degradation in image quality” in Methods). “

Extended Data Fig. 1. Effect of imaging parameters on image quality. **a.** Impact of X-ray voltage and current on image contrast and noise in static capture mode. **b–e.** Influence of voltage, current, and frame capture rate (f) on contrast and noise in continuous capture mode. **f.** Quantification of motion-induced blurriness as a function of magnet translation speed (v_{mag}) under continuous capture. Blurriness is assessed via the normalized Tenengrad sharpness metric, where lower values indicate increased motion blur. All experiments were performed using the XPERT 80 imaging system (KUBTEC Scientific). In all figures, scale bars represent 5 mm.

Algorithm 1 Object Tracking

```
1: Initialize:  $t \leftarrow 0$  ▷ Frame index
2:  $\mathcal{T} \leftarrow \emptyset$  ▷ Set of active tracks
3:  $\mathcal{K} \leftarrow \emptyset$  ▷ Set of Kalman filters
4:  $\mathcal{C} \leftarrow \emptyset$  ▷ Set of candidate detections with counters
5:  $T_{\text{confirm}} \leftarrow 5$  ▷ Frame threshold for track confirmation
6: procedure PROCESSFRAME( $I_t$ )
7:   if  $t = 0$  then
8:      $\mathcal{D}_t \leftarrow \text{global}(I_t)$  ▷ Initial global detection  $d_i \in \mathcal{D}_t$ 
9:      $\text{ROI}_i \leftarrow \text{createROI}(d_i)$ 
10:     $\mathbf{x}_i^0 \leftarrow [p_x, p_y, v_x, v_y]^\top$  ▷ Initial state
11:     $\mathcal{K}_i \leftarrow \text{KalmanInit}(\mathbf{x}_i^0, \mathbf{P}_0)$ 
12:     $\mathcal{T} \leftarrow \mathcal{T} \cup \{\tau_i\}$ 
13:     $\text{age}(\tau_i) \leftarrow 1$ 
14:   else
15:     Step 1: Prediction for existing tracks  $\tau_i \in \mathcal{T}$ 
16:      $\hat{\mathbf{x}}_i^t \leftarrow \mathcal{K}_i(\cdot)$  ▷ Kalman prediction
17:      $\text{ROI}_i \leftarrow \text{updateROI}(\hat{\mathbf{x}}_i^t)$ 
18:     Step 2: Detection and association for existing tracks  $\tau_i \in \mathcal{T}$ 
19:      $\mathcal{D}_i \leftarrow \text{local}(I_t, \text{ROI}_i)$ 
20:     if  $\mathcal{D}_i = \emptyset$  then
21:        $\mathcal{D}_i \leftarrow \text{global}(I_t)$ 
22:       if  $\mathcal{D}_i = \emptyset$  then
23:          $\text{stopActuation}()$ 
24:          $\text{retractMagnet}()$ 
25:          $\mathcal{D}_i \leftarrow \text{global}(I_t)$  ▷ Re-attempt detection
26:       end if
27:     end if
28:      $d_i^* \leftarrow (\hat{\mathbf{x}}_i^t, \mathcal{D}_i)$ 
29:     if  $d_i^* \neq \emptyset$  then
30:        $\mathcal{K}_i(\mathbf{z}_i^t)$  ▷ Update with measurement
31:        $\text{age}(\tau_i) \leftarrow \text{age}(\tau_i) + 1$ 
32:     end if
33:     Step 3: New candidate detection and confirmation
34:      $\mathcal{D}_t^{\text{new}} \leftarrow \text{global}(I_t) \setminus \{\text{associated detections}\} \quad d_j \in \mathcal{D}_t^{\text{new}}$ 
35:     if  $d_j \in \mathcal{C}$  then ▷ Existing candidate
36:        $c_j \leftarrow c_j + 1$  ▷ Increment counter
37:       if  $c_j \geq T_{\text{confirm}}$  then
38:         Initialize new track  $\tau_j$ 
39:         Initialize new Kalman filter  $\mathcal{K}_j$ 
40:          $\mathcal{T} \leftarrow \mathcal{T} \cup \{\tau_j\}$ 
41:          $\mathcal{C} \leftarrow \mathcal{C} \setminus \{d_j\}$  ▷ Remove from candidates
42:       end if
43:     else
44:        $\mathcal{C} \leftarrow \mathcal{C} \cup \{d_j\}$  ▷ Add new candidate
45:        $c_j \leftarrow 1$  ▷ Initialize counter
46:     end if
47:     Step 4: Cleanup stale candidates  $d_j \in \mathcal{C}$ 
48:     if detection in current frame matching  $d_j$  then
49:        $c_j \leftarrow \max(0, c_j - 1)$  ▷ Decrement counter
50:       if  $c_j = 0$  then
51:          $\mathcal{C} \leftarrow \mathcal{C} \setminus \{d_j\}$  ▷ Remove stale candidate
52:       end if
53:     end if
54:   end if
55:    $t \leftarrow t + 1$ 
56: end procedure
```

Extended Data Fig. 6. Pseudocode of the object tracking algorithm.

2. The authors used synthesized and auto-labeled X-ray images for model training. However, if the training dataset can be auto-labeled, why not use it for testing directly?

We thank the reviewer for this important question. The **automatic labeling is an inherent part of the synthetic data generation pipeline** (The generation of auto-labeled synthetic data pipeline can be referred to Fig. 2); it is **not** a separate step that can be applied to arbitrary images, such as MMDs in real tissues.

To clarify our testing strategy:

- The goal is generalization to the real world: the ultimate objective of our work is to develop a model that performs reliably on real clinical x-ray images. Evaluating on a held-out set of auto-labeled synthetic data would not measure the critical synthetic-to-real transfer performance, which is the central challenge our work aims to address.
- Real data is the true benchmark: the most rigorous and clinically relevant evaluation is to test the model on real, manually annotated X-ray images that it has never seen during training. This is the only way to validate that our synthetic data generation pipeline successfully produces a model capable of operating in the target domain.

3. The authors applied stable diffusion for tissue generation. However, stable diffusion is known to suffer structural hallucinations, such as detail duplication/missing, and generate samples that lie outside the training distribution. How does the author prevent these hallucinations?

We thank the reviewer for this critical question.

- Our objective is to **generate diverse tissue backgrounds** to train a robust model for localizing MMDs
- The MMD's own image quality is kept controlled and realistic as there **a separate stage to integrate MMDs to the background** (Fig. 2c).

We acknowledge that structural hallucinations are a known challenge in diffusion models. Rather than seeking to eliminate them entirely, our strategy is twofold:

- **Background quality control**: proactively control and minimize severe artifacts that harm downstream CV model training,
- **Investigation of downstream model robustness to background quality**: systematically evaluate how different levels of background quality affect the downstream task, leveraging mild artifacts as a form of beneficial data augmentation.

Our approach is detailed below and summarized in the provided supplementary figures.

1. Background quality control

1.1. During model training:

We periodically saved candidate models during training and evaluated them using a multi-faceted protocol:

- quantitative fidelity (structural similarity index measure): we quantified structural similarity to real tissue images (**Fig. 3a, b**).
- feature distribution: we ensured generated images expanded upon the feature distribution of real data without diverging from it (**Fig. 3c**).
- Qualitative and human-centric assessment: an operator selected the final model based on the generation of realistic, randomized textures with appropriate illumination and contrast, avoiding surreal artifacts.

Fig. 3. Domain randomization of synthetic tissue images and open-sourced microrobot X-ray dataset. **a.** Real tissue images under X-ray imaging. **b.** Generation of precisely matched synthetic tissues with randomized and enhanced textures. SSIM represents the structural similarity index measure. **c.** Image generation with mask-guided conditioning and prompts. Randomization of masks and prompts significantly expands the dataset. **d.** Domain comparison of real and synthetic tissues. Features from 1,140 real and 24,803 synthetic tissue images, extracted using Inception V3, are visualized via principle component analysis (PCA). PC represents principal component.

1.3 During model inference:

We select generation parameters to minimize artifacts. The low-quality generation cases are categorized as:

- Artifacts: surreal textures, blurriness, high-frequency noise, grid-like or checkerboard patterns, and overly noisy images.
- Inconsistent physics: inconsistent illumination/exposure, incorrect contrast,
- Lack of realistic textures: smoothness in areas that should have texture.

These artifacts are illustrated in **Supplementary Fig. 14**.

Supplementary Figure 14. Quality assessment and parameter sensitivity in diffusion model inference. **a.** Characteristic artifacts in low-quality generations, categorized as: generation artifacts (surreal textures, blurriness, high-frequency noise, grid/checkerboard patterns), physical inconsistencies (non-uniform illumination, exposure mismatches, implausible contrast), and texture deficiencies (excessive smoothness in structurally detailed regions).

We tuned the number of diffusion steps and the guidance scale (ρ) to find a stable to minimize the occurrence of low-quality generation, as indicated by the green markers in Supplementary Fig. 3b.

Supplementary Fig. 14. Quality assessment and parameter sensitivity in diffusion model inference. **b.** Trade-offs between number of diffusion steps and classifier-free guidance scale: low diffusion step counts combined with high prompt guidance weights are prone to produce

unstable or distorted outputs. Parameters selected for final inference are indicated by green markers.

These contents have been added to the “**Diffusion model training and inference**” section in Methods.

2. Investigation of downstream model robustness to background quality

We created five synthetic datasets by blending 200 high-quality and 200 low-quality backgrounds in varying ratios (1.0, 0.86, 0.5, 0.15, 0.0). We then trained 20 instance segmentation models of different model sizes (2.8M, 10.1M, 22.4M, 27.6M) on these datasets and evaluated them on a held-out test set of real X-ray images (**Extended Data Fig. 2**).

Extended Data Fig. 3. Effect of synthetic background quality on downstream segmentation models. **a.** Dataset composition. 200 high- and 200 low-quality generated tissue backgrounds were used to construct five synthetic training sets with varying proportions of high-quality backgrounds (ratios: 1.0, 0.86, 0.5, 0.15, 0.0). **b.** Real-tissue test dataset. The test images include soft MMDs in porcine brain with embedded bone, porcine brain, heart, liver, stomach, heart 3D vessels, in vivo rabbit, in vivo rat, and in vivo rat spine at low magnification. Scale bars represent 5 mm.

Key Findings:

- For MMDs with high visibility and distinct features (e.g., stents): model performance is consistent or better across all background mixtures. In this case, low-quality backgrounds act as a beneficial data augmentation, improving robustness.
- For low-contrast or ambiguous targets (e.g., blurred white rectangles resembling background artifacts), the effect is model-capacity dependent:
 - Large Models (22.4M, 27.6M): A modest fraction (0.14) of low-quality backgrounds improved or maintained performance, acting as a regularizer that

forces the model to learn more robust features, while the performance degrades with a majority of low-quality data.

- Small Models (2.8M, 10.1M): Performance degraded with the inclusion of low-quality backgrounds, as their limited capacity made them susceptible to learning spurious correlations.

Extended Data Fig. 3. Effect of synthetic background quality on downstream segmentation models. c. Segmentation performance across background quality ratios and model sizes. For structurally distinct targets (e.g., stents) with high visibility, performance remains stable or improved regardless of background quality, where low-quality backgrounds provide beneficial augmentation. In contrast, for low-contrast or ambiguous targets (e.g., blurred white rectangles resembling background artifacts), inclusion of a modest fraction of low-quality backgrounds (ratio = 0.86) improves or maintains performance for larger models, but degrades performance for smaller models, evidenced by missed detections. The red dashed lines represent the result of model trained on real data. Scale bars represent 1 mm.

All the relevant data and model weights are available in the open-sourced dataset.

Therefore, our framework adopts the following strategies to ensure the downstream model performance. We achieve this through:

- Proactive quality control during diffusion model training and inference.
- For challenging imaging scenes, utilizing large models for their demonstrated robustness to background noise, effectively turning a potential weakness into a form of data augmentation.

We could further develop a learned filter or classifier to automatically score and select the most beneficial "bad" backgrounds for augmentation, improving the downstream model performance.

The following paragraph has been added to the main text.

“We also investigate the impact of synthetic background quality on downstream CV models (Extended Data Fig. 3). For MMDs with distinct features like stents, performance is largely unaffected by background quality, whereas for MMDs with ambiguous features, the effect is model-dependent: smaller models degrade with low-quality data, while larger models can utilize it as effective regularization. To tackle this issue, we adopt a two-phase quality control strategy during tissue generation (diffusion model selection and artifact minimization) and prioritize the utilization of large downstream models with its robustness to noise (“Diffusion model training and inference” in Methods). A classifier can be developed to automatically select backgrounds for CV model training as a future step.”

4. Generative models may fail to accurately reproduce pathological findings (such as tumors, fractures, or inflammations), and may synthesize non-existent pseudo-features or simply blurred details. This can mislead downstream diagnostic models learning incorrect guidance. Have the authors compared the framework on true dataset versus synthesized dataset?

We thank the reviewer for this important question. We clarify that **our framework targets MMD localization**, not disease diagnosis, which fundamentally changes the requirements for our synthetic data.

- Objective clarification: Our goal is to **generate diverse anatomical backgrounds for training robust MMD detectors**. The MMDs themselves are integrated as separate components (**Fig. 2c**).
- Handling of artifacts (see response to Comment #3): We proactively manage generative artifacts through systematic **quality control**. We also systematically evaluate how different levels of background quality affect the downstream task, leveraging mild artifacts as a form of beneficial data augmentation
- **Comparison with CV models trained on real images**: we trained a CV model on a set of high-cost, manually annotated real X-ray data. This represents the current standard CV model training process. As shown in Extended Data Fig. 2 (original Fig. 4), the model trained on synthetic data achieves comparable or better performance to the real-data model on diverse test sets of real images.

Fig. 2. Workflow of MicroSyn-X. **b.** Controlled tissue generation process. Stable Diffusion creates high-fidelity tissue images from user-defined masks and prompts. **c.** Integration of medical devices with tissues. Captured or generated robot images are seamlessly integrated into the background with flexible parameters, ensuring pixel-accurate labeling.

Extended Data Fig. 2. Evaluation of MicroSyn-X. **a.** Schematic of the evaluation process. **b.** Cross-domain feature comparison using CNN-extracted features visualized by dimensionality reduction (DR). **c.** Comparison of soft MMD datasets via PCA, with dashed lines indicating data distribution boundaries. **d.** Performance of models trained on synthetic and real soft MMD data across classification, detection, and segmentation tasks, measured by AP, mAP50, and

mAP50:95. mAP(B) and mAP(M) denote bounding-box and mask mAPs, respectively. Tissue index indicates datasets: soft MMDs in porcine brain with embedded bones, porcine brain, heart, liver, stomach, heart 3D vessels, and in vivo animals. Models with a parameter size of 22.4M is utilized. e. Cross-domain comparison of liquid MMD datasets. f. Performance comparison between models trained on synthetic and real liquid MMD data. Tissue index indicates datasets: liquid MMDs in porcine brain, heart, liver, stomach, brain or liver with embedded bones, and robot swarm under bone occlusion.

Supplementary Fig. 6. Comparison between models trained with synthetic and real data. a. Performance of models trained on synthetic and real soft MMD data, measured by mAP50(M), and mAP50:95(M). Tissue index indicates datasets: soft MMDs in porcine brain with embedded bones, porcine brain, heart, liver, stomach, heart 3D vessels, and in vivo animals. b. Performance of models trained on synthetic and real liquid MMD data. Tissue index indicates

datasets: liquid MMDs in porcine brain, heart, liver, stomach, brain or liver with embedded bones, and robot swarm under bone occlusion.

5. In Fig4, when evaluating the system robustness, the authors collected eight data indexes for comparing model versus human experts. What are these data indexes stand for? Are the testing data also synthetic or real? For robustness test, it is suggested to evaluate on tissue X-rays on different human subjects. Are there significant subject variations when it comes to unseen new subjects for the model?

We thank the reviewer for these critical questions regarding our evaluation's robustness. Please find our point-by-point response below.

1. Testing Data

All testing data presented consists of real X-ray images, including both ex vivo tissues and in vivo animal experiments. No synthetic data was used for testing, as the core objective was to validate the model's performance in real-world scenarios.

2. Explanation of the eight data indexes

The eight data indexes represent distinct imaging parameter settings (variations in X-ray voltage and current) that directly impact the quality of the MMD visualization. We have adjusted the figure to make it clearer.

Extended Data Fig. 4. Robustness evaluation of MicroSyn-X. **a.** Evaluation task comparing clinical expert labels with MicroSyn-X. Videos of six static robots inside a skull model were recorded across varying imaging parameters for quantifying detectable robots. The numbers represent the indexes of imaging setting. **b.** The imaging effect of MMDs under different imaging settings. Error bars represent standard deviation across different MMDs. **c.** Expert-label comparison. Model-derived robot counts and expert annotations are evaluated across eight imaging parameters. Error bars represent standard deviation across five clinicians and model detection confidences. **d.** Detection confidence distribution relative to contrast and noise levels. Effective detections require a segmentation mask IoU > 0.5 with ground truth. A subset of dataset D2-s was manually annotated, with defined contrast and noise computation regions defined.

3. Evaluation of subject variation and robustness

We have evaluated the model's performance across variations in tissue subjects, MMD subjects, and imaging conditions:

- **Tissue Variation**: the model was tested on a wide range of tissues (both ex vivo and in vivo) that were completely unseen during the training of both the computer vision and generative models. This includes significant differences in anatomical structures, tissue density, and background texture.
- **MMD Variation**: the MMDs themselves undergo substantial variations, including:
 - Soft MMD: Changes in size, dimension, bending, and frequent occlusion.
 - Liquid MMD: Extreme shape transformations, such as splitting and merging.
- **Imaging effect variation**: there are large variations in contrast, noise, brightness, and blurriness.

4. Testing on human tissues.

We agree that testing on human subjects is the ultimate goal, extensive animal experiments are currently the established standard for this development stage due to ethical and regulatory constraints. The anatomical similarities under X-ray imaging between the animal models used and humans provide a good approach to testing the proposed framework. Human trials are planned as the next stage of this research.

6. After the model training, what is the objective for navigation and guiding? When there are multiple paths to guide the robot, which path will the framework choose? It is recommended to add these decision rule details in the figure and discussions.

We thank the reviewer for this helpful comment. The objective it for **controlling miniature medical devices for medical interventions in tissues.**

1. Navigation objective

The primary objective after successful model training is to leverage the CV model for precisely controlling the MMD, enabling it to:

- Navigate to a target location within the anatomy, where the MMD needs to traverse physiological barriers, such as blood vessel bifurcations.
- Perform a desired diagnostic or therapeutic function, such as drug delivery, upon arrival.

This objective is now more clearly stated in the main text.

“Specifically, small-scale devices actuated by external fields can **navigate through enclosed spaces** challenging for conventional tethered tools, and offer functionalities like drug delivery and physiological property sensing.”

2. Path selection and decision rules

The reviewer correctly identifies a critical aspect of navigation. In our current experimental setup within the blood vessel network, the MMD path is largely anatomically deterministic. The vascular network presents a bifurcation structure rather than a complex web of equivalent paths, simplifying the decision-making process to following the correct branch at each bifurcation.

For the actuation, the path for the external magnet (which actuates the robot) is not chosen arbitrarily. It is calculated by a dedicated planning algorithm to maximize magnetic actuation and ensure path continuity and safety. Please refer to our prior work [1].

To avoid the confusion of robot arm and the miniature robot, we replace “soft robot”, “liquid robot” with “soft MMD” and “liquid MMD” throughout the manuscript,

References:

[1] Wang, C., Wang, T., Li, M., Zhang, R., Ugurlu, H., & Sitti, M. (2024). Heterogeneous multiple soft millirobots in three-dimensional lumens. *Science Advances*, 10(45), eadq1951.

Reviewer #3 (Remarks on code availability):

I reviewed the code briefly but didn't try running it. It seems clear and solid.

We appreciate the reviewer's effort in examining our code.

Second revision 2.

Reviewer #1 (Remarks to the Author):

Comment 1: The authors have addressed most of my previous concerns. I have only one minor point that should be clarified before publication:

The manuscript now explains that programmatic prompt generation involves (i) a one-time curation step and (ii) subsequent programmatic generation. For the one-time curation, the authors note that 10–20 terms were initially defined. To ensure transparency and reproducibility, the complete list of curated terms should be provided. Are these terms already included in the code repository? If not, they should be added to the supplementary materials.

Aside from this point, I have no further comments.

Response:

We thank the reviewer for this comment. These items were provided in the code repository under the folder `model_weights/Diffusion_model_weights`.